# The Fractional Energy Balance Equation for Climate projections through 2100

Roman Procyk[1], Shaun Lovejoy[1], and Raphael Hébert[2,3]

[1]Physics Dept., McGill University, 3600 rue University, Montreal, Quebec, H3A 2T8, Canada
[2]Alfred-Wegener Institute Helmholtz Center for Polar and Marine Research, Telegrafenberg A45, 14473 Potsdam, Germany
[3]Institute of Geosciences, University of Potsdam, Karl-Liebknecht-Str. 24/25, 14476 Potsdam, Germany

**Correspondence:** Roman Procyk (roman.procyk@mail.mcgill.ca)

**Abstract.**

We produce climate projections through the $21^{st}$ century using the fractional energy balance equation (FEBE): a generalization of the standard energy balance equation (EBE). The FEBE can be derived from Budyko-Sellers models, or phenomenologically through the application of the scaling symmetry to energy storage processes, easily implemented by changing the integer order of the storage (derivative) term in the EBE to a fractional value.

The FEBE is defined by three parameters: a fundamental shape parameter, a timescale and an amplitude, corresponding to, respectively, the scaling exponent $h$, the relaxation time $\tau$, and the equilibrium climate sensitivity (ECS). Two additional parameters were needed for the forcing: an aerosol re-calibration factor $\alpha$ to account for the large aerosol uncertainty, and a volcanic intermittency correction exponent $\nu$. A Bayesian framework based on historical temperatures and natural and anthropogenic forcing series was used for parameter estimation. Significantly, the error model was not ad hoc, rather predicted by the model itself: the internal variability response to white noise internal forcing.

The 90% Credible Interval (CI) of the exponent and relaxation time were $h = [0.33, 0.44]$ (median=0.38), and $\tau = [2.4, 7.0]$ (median=4.7) years compared to the usual EBE $h = 1$, and literature values of $\tau$ typically in the range 2–8 years. Aerosol forcings were too strong, requiring a decrease by an average factor $\alpha = [0.2, 1.0]$ (median=0.6); the volcanic intermittency correction exponent was $\nu = [0.15, 0.41]$ (median=0.28) compared to standard values $\alpha = \nu = 1$. The overpowered aerosols support a revision of the global modern (2005) aerosol forcing 90% CI to a narrower range $[-1.0, -0.2] W m^{-2}$. For the IPCC forcings, the key parameter ECS = $[1.6, 2.4]K$ (median = $2.0K$) compared with the IPCC AR5 range $[1.5, 4.5]K$ (median = $3.2K$). Similarly, we found the transient climate sensitivity (TCR) = $[1.2, 1.8]K$ (median = $1.5K$) compared to the AR5 range TCR = $[1.0, 2.5]K$ (median =$1.8K$). As often seen in other observational-based studies, the FEBE values for climate sensitivities are therefore somewhat lower but still consistent with those in IPCC AR5.

Using these parameters we made projections to 2100 using both the Representative Carbon Pathways (RCP) and Shared Socioeconomic Pathways (SSP) scenarios, and compared them to the corresponding CMIP5 and CMIP6 multi-model ensembles (MMEs). The FEBE historical reconstructions (1880–2020) closely follow observations, notably during the 1998–2014 slowdown ("hiatus"). We also reproduce the internal variability with the FEBE and statistically validate this against centennial

scale temperature observations. Overall the FEBE projections were 10-15% lower but due to their smaller uncertainties, their 90% CIs lie completely within the GCM 90% CIs. This agreement means that the FEBE validates the MME and vice versa.

# 1   Introduction

The Earth is a complex, heterogenous system with turbulent atmospheric and oceanic processes operating over scales ranging from millimeters up to planetary scales. When considered by time scale, there are three main regimes: the weather,
macroweather and climate (Lovejoy and Schertzer, 2013; Lovejoy, 2013). From dissipation times up until the scale of ten days - the lifetime of planetary structures - fluctuations in the temperature and other atmospheric quantities increase with time scale: this is the weather regime. Beyond this - in macroweather - fluctuations generally decrease with scale: averaging anomalies over longer and longer times decrease their average. Eventually, this is reversed and fluctuations again tend to increase, marking the beginning of the climate regime. In the industrial epoch this occurs at a scale of $\approx 20$ years, while in the pre-
industrial epoch the transition is at centuries or millennia and the regime continues up to Milankovitch scales (Lovejoy, 2015b, 2019b).

A major challenge is to determine the Earth's decadal and centennial response to anthropogenic and natural perturbations. At the moment, projection uncertainties - famously exemplified in the range $1.5$–$4.5K$ for a $CO_2$ doubling – are so large that for many purposes, including the development of mitigation policies, the development of complementary approaches are needed.
When considering alternatives, although perturbations to the Earth system can be quite varied, when compared to the mean solar radiation, over the past and future decades, those of interest are of the order of only a few percent. This allows diverse forcings to be conveniently approximated by their equivalent radiative forcings. It also explains why - in spite of their highly nonlinear weather dynamics - that to a good approximation, General Circulation Model (GCM) macroweather and climate responses to deterministic external perturbations are typically linear (as quantified for CMIP5 models in Hébert and Lovejoy
(2018)) but with stochastic internal variability.

In order to construct macroweather and climate models, beyond linearity and stochasticity, we require additional model constraints, the classical one being energy balance. Starting with the first Energy Balance Models (EBMs) proposed by Budyko (1969) and Sellers (1969), EBMs and stochastic climate models have been extensively used for understanding the climate (North, 1975; Hasselmann, 1976; North et al., 1981; Imkeller and Von Storch, 2001; Trenberth et al., 2014; North and Kim,
2017; Proistosescu et al., 2018; Ziegler and Rehfeld, 2020). In this paper, we will only consider EBMs for the globally averaged temperature. The resulting "zero dimensional" energy balance equation (EBE) is a first order linear differential equation, it can be obtained by considering the Earth to be a uniform slab of material ("box") radiatively exchanging heat with outer space. Such box models usually involve at least two boxes and they assume Newton's law of cooling as well as ad hoc assumptions relating surface temperature gradients to the rate of heat exchange.
Energy conservation is an important symmetry principle, yet when implemented in box type models, it violates another symmetry: scale invariance. This is because box models are integer ordered differential equations whose response functions (Green's functions) are exponentials (see Ghil and Lucarini, 2020, for a discussion on the exponential decay of the Green

function in the climate context). In order to respect the scaling, these "climate response functions" (CRFs) have therefore been postulated to be scaling (power-law). However, the use of pure power law CRFs (e.g. Rypdal, 2012; Myrvoll-Nilsen et al., 2020) leads to divergences: the "runaway Green's function effect" (Hébert and Lovejoy, 2015) so that if the Earth is perturbed by even an infinitesimal step function forcing, its temperature monotonically increases without ever attaining thermodynamic equilibrium: its ECS is infinite. Whereas the classical EBMs conserve energy but violate scaling, the pure scaling CRF models are scaling but violate energy conservation. Such models can only make projections by using forcings that start and then return to zero.

Hébert et al. (2021) proposed taming the divergences by cutting off the power law CRFs at small scales. The resulting model was scaling at long times and when forced by step functions, reaches thermodynamic equilibrium. With this truncated power law CRF and using Bayesian techniques Hébert et al. (2021) were able to make climate projections through 2100 with the IPCC RCP scenario forcings that were coherent with the MME 90% CI. Furthermore, using the historical part of each GCM simulation, the corresponding GCM climate projections were accurately reproduced, meaning (in regards to the Earth's globally averaged temperature) that both models were effectively equivalent. The caveat was that the CRF model truncation was somewhat ad hoc, and therefore only useful at decadal or longer scales.

To make more realistic models, the key issue is energy storage. Storage is a consequence of imbalances in incoming short wave and outgoing long wave radiation and it must be accounted for in applications of the energy balance principle (Trenberth et al., 2009). As pointed out in Lovejoy (2019a, b) and developed in Lovejoy et al. (2021) it is sufficient that the scaling principle not be applied to the CRF, but rather to the storage term in the EBE. In lieu of the energy being stored by uniformly heating a box, energy is instead stored in a hierarchy of structures from small to large, each with time constants that are power laws of their sizes. This conceptual shift can be implemented simply by changing the integer order of the storage (derivative) term in the EBE to a fractional value: the Fractional Energy Balance Equation (FEBE). While Lovejoy et al. (2021) derived the FEBE in a phenomenological manner, Lovejoy (2021a, b) showed how it could instead be derived from the classical continuum mechanics heat equation used in the Budyko-Sellers models. Indeed, by extending Budyko-Sellers models from 2D to 3D (i.e. to include the vertical) and imposing the (correct) conductive – radiative surface boundary conditions, one immediately obtains fractional order equations for the surface temperature. In other words, nonclassical fractional equations and long memories turn out to be necessary consequences of the classical Budyko-Sellers approach.

To understand the FEBE's key new features, recall that linear differential equations can be solved with Green's functions; in the classical integer ordered case, these are based on exponentials. However in the general case where one or more terms are of fractional order, they are instead based on "generalized exponentials", themselves based on power laws. In the FEBE, there are two distinct power law regimes with a transition at the relaxation time (estimated to be of the order of a few years, see below). While the low frequency Green's function can be very close to Hébert et al. (2021)'s truncated power law CRF, the high frequency regime is able to produce internal variability coherent with the observed scaling and fractional Gaussian noise used for skillfully forecasting the stochastic (internal) variability at monthly, seasonal, interannual (macroweather) scales (Lovejoy et al., 2015; Del Rio Amador and Lovejoy, 2019, 2021a). In short, there are theoretical arguments as well as empirical evidence

that the FEBE accurately models the Earth's temperature response to both internal and external forcing over macroweather and climate time scales.

The following text introduces the FEBE (Sect. 2.1), describes the radiative forcing, temperature and GCM simulations that are used (Sect. 2.2), and introduces Bayesian inference for determining the model and forcing parameters (Sect. 2.3). Using these we present the probability distribution functions for the parameters (Sect. 3.1 and 3.2), estimate the Equilibrium Climate Sensitivity (ECS) and Transient Climate Response (TCR) (Sect. 3.3). Using our parameters discuss the reliability and statistically analyse the FEBE (Sect. 4), produce global projections to 2100 using the Representative Carbon Pathways (RCPs) and Shared Socioeconomic Pathways (SSPs), and estimate the probability of exceeding various warming thresholds all which we compare to the corresponding CMIP5 and CMIP6 GCM Multi-Model Ensembles (MMEs) (readers only wanting results, can skip to Sect. 5).

## 2 Methods and Material

### 2.1 The FEBE

The zero-dimensional FEBE may be written:

$$\tau^h \, _{-\infty}D_t^h T + T = s\mathcal{F}; \quad \mathcal{F}(t) = F(t) + f(t), \quad 0 \leq h \leq 1 \tag{1}$$

(Lovejoy, 2019a; Lovejoy et al., 2021). Where $T(t)$ is the Earth temperature anomaly with respect to a reference temperature ($\lim_{t\to-\infty} T(t) = 0$), $\tau$ is the relaxation time, $s$ is the climate sensitivity, $\mathcal{F}(t)$ is the anomalous external radiative forcing which is the sum of stochastic $f(t)$ and deterministic $F(t)$ components, and $h$ is the order of the Weyl fractional derivative (see e.g. Podlubny, 1999):

$$_{-\infty}D_t^h T = \frac{1}{\Gamma(1-h)} \int_{-\infty}^t (t-u)^{-h} T'(t) dt', \quad T'(u) = \frac{dT}{du} \tag{2}$$

where $\Gamma$ is the Gamma function. If this derivative is integrated by parts and the limit $h \to 1$ is taken, using $\lim_{t\to-\infty} T(t) = 0$, $_{-\infty}D_t^h T = \frac{dT}{dt}$ so that we recover the standard box EBE (Lovejoy et al., 2021).

If we solve the FEBE using Green's functions, we obtain:

$$T(t) = s \int_{-\infty}^t G_{0,h}(t-u)\mathcal{F}(u) du. \tag{3}$$

Where $G_{0,h}$ is the impulse (Dirac) response Green's function, for the FEBE it is given by:

$$G_{0,h}(t) = \begin{cases} \tau^{-1} \left(\frac{t}{\tau}\right)^{h-1} E_{h,h}\left(-\left(\frac{t}{\tau}\right)^h\right); & t \geq 0 \\ 0; & t < 0. \end{cases} \tag{4}$$

Where:

$$E_{\alpha,\beta}(z) = \sum_{k}^{\infty} \frac{z^k}{\Gamma(\alpha k + \beta)} \tag{5}$$

is the "$\alpha, \beta$ order Mittag-Leffler function" (these and most of the following results are in the notation of Podlubny (1999). The condition $G_{h,x}(t) = 0$ for $t < 0$ is needed to respect causality, in what follows implicitly assumes this for all Green's functions. The Mittag-Leffler functions are often called "generalized exponentials", the classical $h = 1$ box model is the (exceptional) ordinary exponential: $E_{1,1}(z) = e^z$.

Mathematically when $0 < h < 1$, the FEBE is a "fractional relaxation equation" where $\tau$ quantifies the slow, power law approach to a new thermodynamic equilibrium. Rather than express solutions in terms of the impulse response $G_{0,h}$, it is often more convenient to use the step response $G_{1,h}$:

$$G_{1,h}(t) = \int_0^t G_{0,h}(u)du = \left(\frac{t}{\tau}\right)^h E_{h,h+1}\left(-\left(\frac{t}{\tau}\right)^h\right). \tag{6}$$

Such that the temperature response can be written as:

$$T(t) = s \int_{-\infty}^t G_{1,h}(t-u)F'(u)du, \quad F'(u) = \frac{dF}{du}. \tag{7}$$

$G_{1,h}$ has the advantage of being dimensionless, and it also has a simple interpretation as being the response to a step forcing such as that found in numerical $CO_2$ doubling experiments. At high frequencies ($t \ll \tau$), important for modelling and predicting the internal variability we have:

$$G_{0,h,high}(t) = \frac{1}{\tau\Gamma(h)}\left(\frac{t}{\tau}\right)^{h-1}; \quad G_{1,h,high}(t) = \frac{1}{\Gamma(h+1)}\left(\frac{t}{\tau}\right)^h; \quad t \ll \tau. \tag{8}$$

These correspond to taking the first terms in the series expansions for the Mittag-Leffler functions in eqs. 4, 6. If we consider the response to Gaussian white noise forcing, $\gamma(t)$, then $G_{0,h}(t) \propto t^{h-1}$ implies that $T(t)$ is approximately a fractional Gaussian noise (fGn) with statistical scaling exponent $h$ (when forced by a Gaussian white noise, the FEBE response is exactly a fractional Relaxation noise, see Lovejoy, 2019a). In Lovejoy et al. (2015); Lovejoy (2015a) the high frequency approximation with an exponent corresponding to $h = 0.3$ was used, in Del Rio Amador and Lovejoy (2019), forecasts with the more accurate estimate $h \approx 0.4 \pm 0.05$ (see below) was used.

To see if this is compatible with the value estimated from the low frequency response to external forcings consider the low frequency behaviour ($t \gg \tau$), important for modelling and projecting the multidecadal responses to external forcing:

$$G_{0,h,low}(t) = \frac{-1}{\tau\Gamma(-h)}\left(\frac{t}{\tau}\right)^{-1-h}; \quad G_{1,h,low}(t) = 1 - \frac{1}{\Gamma(1-h)}\left(\frac{t}{\tau}\right)^{-h}; \quad t \gg \tau \tag{9}$$

(note $\Gamma(-h) < 0$ for $0 < h < 1$). In the box model, $h = 1$, case we have exactly $G_{1,1}(t) = 1 - e^{-t/\tau}$ whereas when $h < 1$, the exponential approach to thermodynamic equilibrium is replaced by a power law. Hébert et al. (2021) used $G_1(t) =$

$1 - \left(1 + \frac{t}{\tau}\right)^{H_F}$ with $H_F \approx -0.5^{+0.4}_{-0.5}$ corresponding for $t \gg \tau$ to $h = -H_F \approx 0.5$ which is thus the same h value as that corresponding to the internal forcing. It is thus plausible that the FEBE models both high and low frequency regimes with the unique exponent $h \approx 0.4$. Indeed it was this empirical finding that pre-dated and motivated the discovery of the FEBE.

## 2.2 Data

### 2.2.1 Radiative Forcing Data

We consider natural and anthropogenic sources of external forcing: solar and volcanic, greenhouse gases and aerosols. We use the standard semi-empirical carbon dioxide concentration to forcing relationship (Myhre et al., 1998):

$$F_{CO_2}(\rho) = 3.71 W m^{-2} log_2 \frac{\rho}{\rho_0}. \tag{10}$$

Where $F_{CO_2}$ is the forcing due to carbon dioxide, $\rho$ is the concentration of carbon dioxide and $\rho_0$ is the preindustrial concentration of carbon dioxide which we take to be 277ppm (Solomon, 2007).

We follow the CMIP5 recommendations for anthropogenic and solar forcing, while volcanic forcing is unprescribed (Taylor et al., 2012). The anthropogenic CMIP6 radiative forcings follow Smith et al. (2018a).

### 2.2.2 Greenhouse Gas Forcing

The global climate is warming and most of the observed changes are due to increases in the concentration of anthropogenic greenhouse gases (GHGs) (IPCC, 2013). Future anthropogenic forcing is prescribed in the Representative Concentration Pathways (RCPs), established by the IPCC for CMIP5 simulations: we considered RCP 2.6, RCP 4.5, and RCP 8.5 (Meinshausen et al., 2011b). RCP 6.0 was omitted in this study since fewer CMIP5 modelling groups performed the associated run. In the CMIP6 simulations the anthropogenic forcings are prescribed in the Shared Socioeconomic Pathways (SSPs) (Meinshausen et al., 2020); we investigate SSP 1-26 (strong mitigation), SSP 2-45 (middle of the road) and SSP 5-85 (strong emission) scenarios, designated as high priority for IPCC AR6 and are counterparts to the previous RCP scenarios above.

The RCP scenarios are derived from estimates of emissions computed by a set of Integrated Assessment Models (IAM), these emissions are converted to concentrations using the Model for the Assessment of Greenhouse-gas Induced Climate Change (MAGICC, Meinshausen et al., 2011a), while for the SSP scenarios the emissions are converted to forcings using the Finite Amplitude Impulse Response model (FAIR, Smith et al., 2018a). These scenarios will allow us to compare our results from the FEBE with CMIP5/6 simulations.

The wide spread between the scenarios allows for the investigation of the consequences of various future policies, from strong mitigation (RCP 2.6, SSP 1-26) to no-policy reference (RCP 8.5, SSP 5-85) shown in figure 1 (bottom). For RCP 2.6 and SSP 1-26, the strongest mitigation scenarios, the total radiative forcing has a peak at approximately $3 W m^{-2}$ around the year 2050 and declines thereafter due to large scale deployment of negative emission technologies. RCP 4.5 and SSP 2-45 are stabilization scenarios, with the total radiative forcing rising until the year 2070 and with stable concentrations after the year 2070. In contrast, RCP 8.5 and SSP 5-85 are continuously rising radiative forcing pathways in which the radiative forcing levels

by the end of the $21^{st}$ century reach approximately $8.5 Wm^{-2}$. Current emissions fall somewhere between the $4.5 Wm^{-2}$ and $8.5 Wm^{-2}$ scenarios.

In this paper we use the forcing due to carbon dioxide equivalent, $F_{CO2_{Eq}}$, as the measure of our anthropogenic forcing, $F_{Ant}$, given in the RCP and SSP scenarios. The anthropogenic forcing from gases corresponds to the effective radiative forcing produced by long lived GHGs $F_{GHG}$: carbon dioxide, methane, nitrous oxide and fluorinated gases, controlled under the Kyoto protocol, and ozone depleting substances, controlled under the Montreal Protocol. We show the anthropogenic forcings for each RCP and SSP scenario in figure 1.

### 2.2.3 Aerosol Forcing

Aerosols are a strong component of radiative forcing associated with anthropogenic emissions, resulting from a combination of direct and indirect aerosol effects. There exists high uncertainty of the aerosol forcing, arising from a poor understanding of how clouds respond to aerosol perturbations (Penner et al., 2001; Ramaswamy et al., 2001), compared to the fairly well constrained GHG forcing. We therefore follow (Forest et al., 2002; Harvey and Kaufmann, 2002; Forest et al., 2006; Padilla et al., 2011; Hébert et al., 2021) and introduce the aerosol linear scaling factor $\alpha$ to account for our poor knowledge of aerosol forcing.

We obtained the CMIP5 aerosol forcing from the total $CO_{2_{EQ}}$ forcing by subtracting the combined effective radiative forcing of the gases controlled by the Kyoto protocol, $F_{Kyt}$, and from those controlled under the Montreal protocol, $F_{Mtl}$. $F_{Mtl}$ is given in CFC-12 equivalent concentration and we use the relation from Ramaswamy et al. (2001) to convert this to $Wm^{-2}$.

The total amount of aerosol forcing in 2005 given at the 90% CI in the IPCC AR5 is $[-1.9, -0.1] Wm^{-2}$. However since then, attempts have been made to better constrain this value; Stevens (2015) argues that extreme aerosol forcings (more negative than $-1 Wm^{-2}$) are implausible. Using results from Murphy et al. (2009), Stevens (2015) supports tightening the upper and lower bounds of the aerosol forcing, revising it to $[-1.0, -0.3] Wm^{-2}$ although the wider range from the IPCC's AR5 is still supported by the more comprehensive study by Bellouin et al. (2020).

The prescribed CMIP6 SSP aerosol forcing, $F_{Aer_{SSP}}$, contains contributions from aerosol-radiation interactions and from aerosol cloud interactions: $F_{ari}$ and $F_{aci}$ (Smith et al., 2018a). $F_{ari}$ includes the direct radiative effect of aerosols, in addition to rapid adjustments due to changes in the atmospheric temperature, humidity and cloud profile (formerly the "semi-direct effect"), and is calculated using multi-model results from Aerocom (Myhre et al., 2013). $F_{aci}$ describes how aerosols affect clouds in the radiation budget and is calculated from the aerosol model of Stevens (2015), which includes a logarithmic dependence of $F_{aci}$ on sulphates, black carbon and organic carbon emissions - the source of the difference in aerosol forcing shapes between $F_{Aer_{RCP}}$ and $F_{Aer_{SSP}}$ shown in figure 1 (bottom).

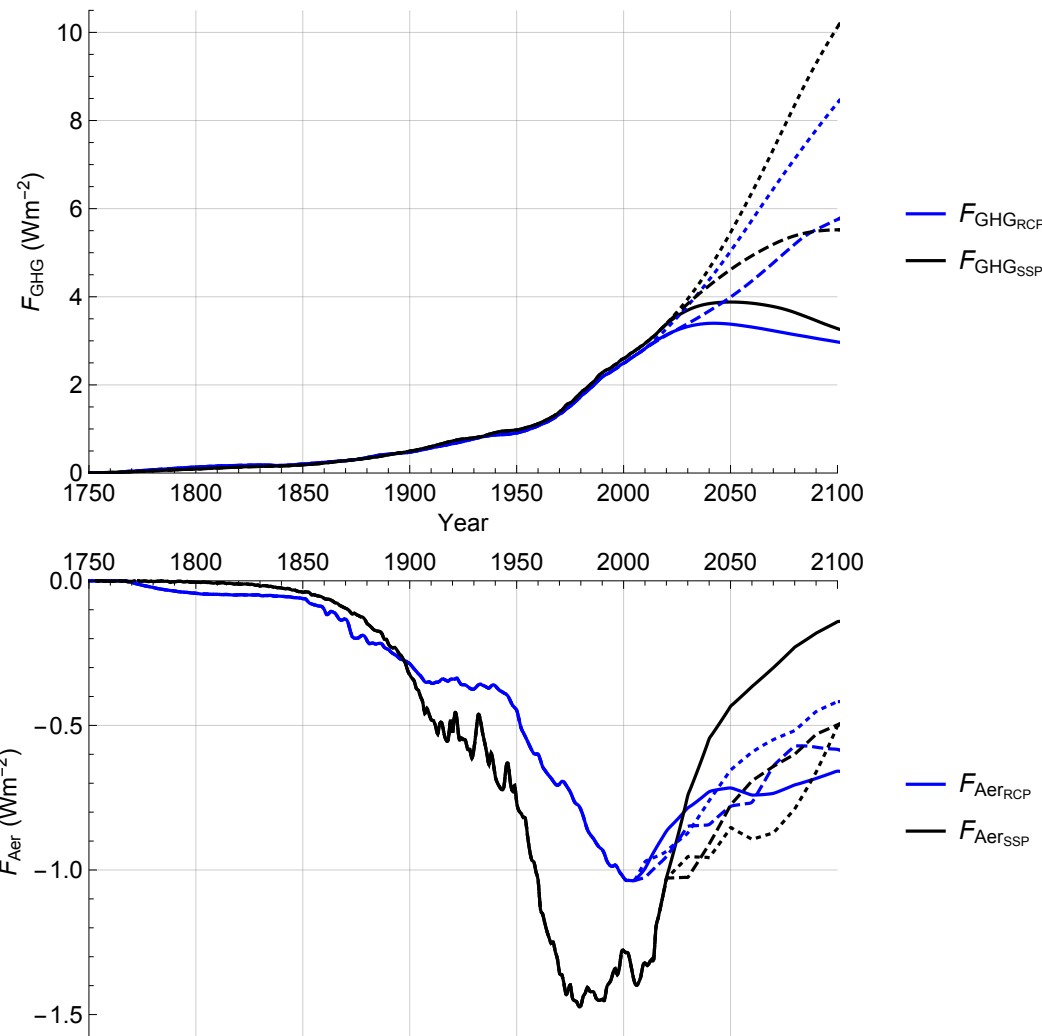

**Figure 1.** (top) The anthropogenic forcing series, the sum of the greenhouse gas forcing $F_{GHG}$ and respective aerosol forcing series $F_{Aer_{RCP}}$ (black) or $F_{Aer_{SSP}}$ (blue) are shown over the historical period and projection period until 2100 for RCP 2.6/SSP 1-26 (solid), RCP 4.5/SSP 2-45 (dashed), and RCP 8.5/SSP 5-85 (dotted).

(bottom) The anthropogenic aerosol forcing series used, $F_{Aer_{RCP}}$ (blue) and $F_{Aer_{SSP}}$ (black) following the same scheme as above. Updated from Hébert et al. (2021).

### 2.2.4 Solar Forcing

The other external forcings considered are solar and volcanic. Although there exist other natural forcings such as mineral dust and sea salt, they are small and will be implicitly included with the internal variability. We use the CMIP5 recommendation for solar forcing, $F_{Sol}$, a reconstruction obtained by regressing sunspot and faculae time series with total solar irradiance (TSI) (Wang et al., 2005), shown in figure 2. Following Meinshausen et al. (2011b), the solar forcing anomaly is calculated as the change in solar constant over the average value of the two 11-year solar cycles from 1882 to 1904 divided by 4 (the effective fraction of the surface of the Earth which is exposed to the sun) and multiplied by 0.7 (representing planetary co-albedo). To extend solar forcing to the future we follow CMIP5 and reproduce solar cycle 23 (the last one prior to 2008) as the assumed future solar forcing.

### 2.2.5 Volcanic Forcing

The volcanic forcings series, $F_{Vol}$, used in this study was generated from the volcanic optical depths, $\tau_V$. Over the 1850 to 2012 period we use the approximate relation: $F_{Vol} \approx -27 W m^{-2} \tau_V$, obtained from the Goddard Institute for Space Science (GISS) website (Sato, 2012). We follow Hébert et al. (2021), extending the series to 1765 using the optical depth reconstruction of Crowley et al. (2008), and setting volcanic forcing to zero for the future.

It is well established that volcanic forcing must be scaled down by 40-50% in order to produce a comparable effect on surface temperature, and thus most EBMs linearly scale volcanic forcing (Tomassini et al., 2007; Ring et al., 2012; Lewis and Curry, 2015; Gregory and Andrews, 2016). However the amplitude of the volcanic forcing is not the only issue; volcanic forcings are highly intermittent (spiky). The intermittency can be quantified in a multifractal framework (Lovejoy and Schertzer, 2013; Lovejoy and Varotsos, 2016) by the intermittency parameter $C_1$ which corresponds to the fractal codimension (i.e. 1-D, where D, is the fractal dimension of the part of series that gives the dominant contribution to the mean of the series) characterizing the sparseness of volcanic "spikes" of mean amplitude. There is also a multifractal index $\alpha_{MF}$ that describes how quickly the intermittency changes as we move away from the mean. Since linear response models do not alter the intermittency, the volcanic series must first be non-linearly transformed before being introduced into a linear response framework. With the effective volcanic forcing $F_{Vol_\nu}$, the volcanic intermittency correction exponent $\nu$ and the mean of the whole volcanic series $\langle F_{Vol} \rangle$, we follow Hébert et al. (2021) using a non-linear relation to change the intermittency so that the transformed signal can be linearly related to the temperature:

$$\frac{F_{Vol_\nu}}{\langle F_{Vol} \rangle} = \frac{F_{Vol}^\nu}{\langle F_{Vol}^\nu \rangle}. \tag{11}$$

The normalization is such that the mean is unchanged: $\langle F_{Vol_\nu} \rangle = \langle F_{Vol} \rangle$; the average volcanic forcing is conserved - this was done for simplicity and if needed future work could include another scaling parameter (this is slightly different than the normalization used in Hébert et al. (2021)). The volcanic intermittency correction exponent, $\nu$, required to reduce the intermittency parameter of the volcanic forcing, $C_{1,F_V}$, to equal the corresponding parameter of the temperature response,

$C_{1,T_V}$, can be calculated theoretically using:

$$C_{1,F_V}\nu^{\alpha_{MF}} = C_{1,T_V} \tag{12}$$

where $\alpha_{MF}$ is the multifractality index of the volcanic forcing, $C_1$ is the codimension of the mean (see ch. 4, Lovejoy and Schertzer, 2013).

The volcanic response appears to be non-linear as the intermittency ("spikiness", sparseness of the spikes) parameter $C_1$
changes from about $C_{1,F_V} \approx 0.16$ for the input volcanic forcing to $C_{1,T} \approx 0.03$ for the temperature response: the latter is therefore much less intermittent than the former although it is possible that the estimated $C_1$ changes slightly due to finite size effects and internal variability. Assuming $\alpha_{MF} \approx 1.5$ (Lovejoy and Schertzer, 2013; Lovejoy and Varotsos, 2016), we find an approximate but plausible theoretical estimate of the volcanic intermittency correction exponent $\nu \approx 0.3$.

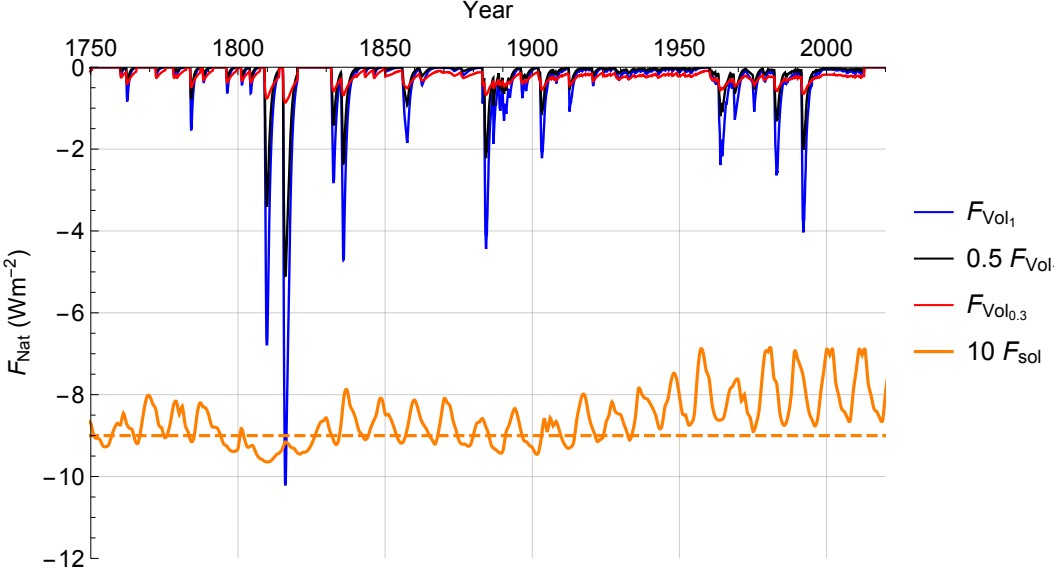

**Figure 2.** Volcanic forcing $F_{Vol_1}$ (blue) is shown alongside two transformed versions: linearly damped by a constant 0.5 coefficient (black), and non-linearly transformed using equation with $\nu = 0.3$ (red). The solar forcing $F_{Sol}$ (orange) has been shifted down by -9 and amplified by a factor of 10 for clarity. Adapted from Hébert et al. (2021).

### 2.2.6 Internal Stochastic Forcing

We consider the standard assumption about internal variability that it is forced by a Gaussian "delta correlated" white noise (Hasselmann, 1976):

$$f(t) = \sigma\gamma(t); \quad \langle\gamma(t)\rangle = 0; \quad \langle\gamma(t)\gamma(u)\rangle = \delta(t-u), \tag{13}$$

where $f(t)$ is the noise at infinite resolution, $\gamma(t)$ is a "unit" white noise and $\sigma$ is its amplitude. When averaged to resolution $\tau_r = 1$ month, the average forcing has amplitude $\langle f_{\tau_r}^2 \rangle^{1/2} = \sigma_\tau = \frac{\sigma}{\sqrt{\tau_r}}$. In comparison, the internal variability of the mean

observational temperature series is equal to the observed series with the forced temperature response removed. We take the global annually averaged monthly temperature anomaly to be $\sigma_{T,\tau_r} \approx \pm 0.14°C$, where $\tau_r$ is the resolution (taken to be monthly in this case).

Using Lovejoy et al. (2021) and $\sigma_{T,\tau_r}$, we can relate $\sigma_{T,\tau_r}$ and $\sigma_{f,\tau_r}$:

$$\sigma_{f,\tau_r} = \frac{\sigma_{T,\tau_r} K_h}{s} \left( \frac{\tau}{\tau_r} \right)^h, \tag{14}$$

$$K_h = \sqrt{\frac{\pi}{2cos(\pi\left(h - \frac{1}{2}\right))\Gamma(-1-2h)}}. \tag{15}$$

Where $K_h$ is a standard normalization constant, $\tau$ is the relaxation time, and $s$ is the climate sensitivity parameter; eq. 14 is an approximation to the FEBE response to white noise forcing valid at short time scales $\tau_r \ll \tau$. If we introduce a white noise forcing, with the standard deviation calculated using eq. 14, the FEBE response will correspond to an internal variability term with realistic amplitude and autocorrelation structure.

Working in a linear framework we write the forcing series, $\mathcal{F}$, as the sum of the deterministic forcings, $F$,(GHG, aerosol, solar and volcanic) and the white noise forcing:

$$\mathcal{F}(\alpha,\nu;t) = F_{GHG}(t) + \alpha F_{Aer}(t) + F_{Sol}(t) + F_{Vol_\nu}(t) + \sigma_{f,\tau_r}\gamma_{\tau_r}(t); \quad F(t) = \langle \mathcal{F}(t) \rangle = F_{GHG}(t) + \alpha F_{Aer}(t) + F_{Sol}(t) + F_{Vol_\nu}(t), \tag{16}$$

where $\gamma_{\tau_r}(t)$ is a unit white noise at resolution $\tau_r$ and $\langle \cdot \rangle$" is the mean ensemble (statistical) average.

### 2.2.7 Surface Air Temperature Data and CMIP5/6 Simulations

We used five historical records of surface air temperature for our analysis each spanning the period 1880-2020, with median monthly temperature anomalies in relation to the reference period of 1880-1910: HadCRUT4 (Morice et al., 2012), the Cowtan & Way reconstruction version 2.0 (C&W, Cowtan and Way, 2014b, a; Cowtan et al., 2015), GISS Surface Temperature Analysis (GISTEMP, Lenssen et al., 2019), NOAA Merged Land Ocean Global Surface Temperature Analysis Dataset (NOAA-GlobalTemp, Zhang et al., 2019; Huang et al., 2020) and Berkley Earth Surface Temperature (BEST, Rohde and Hausfather, 2020).

The HadCRUT4 dataset is a combination of the sea-surface temperature records: HadSST3 compiled by the Hadley Centre of the UK Met Office along with land surface station records: CRUTEM4 from the Climate Research Unit in East Anglia; the Cowtan and Way dataset uses HadCRUT4 as raw data, but interpolates missing data that would lead to bias especially at high latitudes by infilling missing data using an optimal interpolation algorithm (kriging); we use the dataset with land air temperature anomalies interpolated over sea-ice. The GISTEMP dataset combines the Global Historical Climate Network version 3 (GHCNv3) land surface air temperature records with the Extended Reconstructed Sea Surface Temperature version 4 (ERSST) along with the temperature dataset from the Scientific Community on Antarctic Research (SCAR) and is compiled by the Goddard Institute for Space Studies; the NOAA National Climate Data Center uses GHCNv3 and ERSST but applies

different quality controls and bias adjustments. The final data, BEST, makes use of its own land surface air temperature product along with a modified version of HadSST.

The selected CMIP5 models have monthly historical simulation outputs available over the 1860 to 2005 period along with outputs of scenario runs from 2005 to 2100 for RCP 2.6, RCP 4.5, and RCP 8.5, summarized in table A1. The CMIP6 model outputs have monthly historical simulations from 1860 to 2014 and future projections based on the SSP scenarios 1-26, 2-45

and 5-85 (Forster et al., 2020), climate sensitivity of models are summarized in table A2 (Flynn and Mauritsen, 2020).

## 2.3 Bayesian Parameter Estimation

In this section we establish a procedure to estimate the probability distribution associated with the climate sensitivity: $s$, model parameters: $\tau$, $h$ and forcing parameters: $\alpha, \nu$. To estimate them, we relate the forcing to surface air temperature data using the FEBE with a multi-parameter Bayesian technique. To apply Bayesian inference we require temperature observations, a

290 statistical model that relates forcing data to temperature, and prior information about the model parameters (priors). Bayesian inference is chosen due to its ability to better constrain model parameters by using information from different sources including data and models.

Through this framework each parameter combination ($h$, $\tau$ for $G_{0,h}$ and $\alpha$, $\nu$ for $F$ as well as $s$) produces a time-dependent forced response which is associated with a likelihood that depends on how well the corresponding model output matches

the observational temperature records over the historic period. To see how this works, recall that the FEBE describes the temperature response to the sum of the external deterministic forcing $F(t)$ and an amplitude $\sigma$ internal stochastic forcing $\sigma\gamma(t)$:

$$T(t) = T_{ext}(t) + T_{int}(t); \quad \begin{array}{l} T_{ext}(t) = sG_{0,h}(t) * F(t) \\ T_{int}(t) = sG_{0,h}(t) * \sigma\gamma(t) \end{array}, \tag{17}$$

where $T_{ext}$, $T_{int}$ are the responses. Any given set of parameters defines a forced temperature response $T_{ext}(t)$; and when

removed from the observation temperature series, they define a series of residuals:

$$T_{res}(t) = T(t) - T_{ext}(t) = T_{int}(t) = sG_{0,h}(t) * \sigma\gamma(t). \tag{18}$$

The residuals are thus equal to the internal temperature variability, i.e. the response to the internal forcing $\sigma\gamma(t)$. Here we make the usual assumption that $\gamma(t)$ is a Gaussian white noise so that $T_{res}(t) = T_{int}(t)$ is a fractional Relaxation noise process (fRn, Lovejoy, 2019a). However, for scales shorter than the relaxation time $\tau$ (of the order of years), the fRn process is very

close to a fractional Gaussian noise (fGn) process (due to the approximation $G_{0,h} \approx G_{0,high,h}$, eq. 8). Thus, rather than making an ad hoc assumption about the statistics of the residuals, in our approach the statistics are given by the model itself (a key improvement from Hébert et al. (2021)). The fGn approximation takes into account the strong power law correlations induced by the fractional derivative term in the FEBE and it is generally valid except at the low frequencies that only weakly influence the likelihood function. An fGn model for the residuals is more realistic with respect to the autocorrelation function of tem-

perature data (Lovejoy et al., 2015) and thus produces more conservative credible interval in comparison to other exponential

decorrelation models such as an AR(1) since the latter underestimate the decorrelation time, and thus overestimate the effective sample size.

To calibrate the FEBE, we take the time-dependent forced response calculated for each parameter combination and remove it from the temperature series to obtain a series of residuals which represent an estimator of the historical internal variability. The likelihood function ($\mathcal{L}$) corresponds to the probability ("$Pr$") of observing the series $T(t)$ conditioned on the parameters: $s, h, \tau, \alpha, \nu$ (right hand side), assuming the residuals are a fGn process with parameter $h$, and zero mean:

$$\mathcal{L}(s, h, \tau, \alpha, \nu | T(t)) = Pr(T(t) | s, h, \tau, \alpha, \nu). \tag{19}$$

Using Bayes' rule, we can obtain the posterior probability distribution function (PDF) for our parameters using the likelihood function (an a priori probability) and the prior distribution for the parameters, $\pi(s, h, \tau, \alpha, \nu)$:

$$Pr(s, h, \tau, \alpha, \nu | T(t)) = \frac{Pr(T(t) | s, h, \tau, \alpha, \nu) \pi(s, h, \tau, \alpha, \nu)}{Pr(T(t))}. \tag{20}$$

We use the following Mathematica 12.2 (Wolfram Research, Inc., 2020) functions: LogLikelihood[proc, data], FractionalGaussianNoiseProcess[$\mu$, $\sigma'$, $h'$], and EstimatedProcess[data, proc] to calculate the maximum likelihood of those residuals to be a fGn corresponding to our error model. Note that the Hurst exponent $h'$ used within Mathematica 12.2 describes the scaling behaviour of the associated fractional Brownian motion obtained by integrating the fGn. The notation $h = h' - 1/2$ corresponds to the associated parameter in Lovejoy et al. (2015) which directly describes the scaling associated with the fluctuations of the fGn itself.

The priors chosen here are intended to reflect knowledge about the historical climate system. Following Del Rio Amador and Lovejoy (2019) who estimated $h$ from the statistics of the response of the internal forcing, the prior distribution for the scaling parameter is taken to be a normal distribution centered around 0.4 with a standard deviation of 0.1 (twice that of Del Rio Amador and Lovejoy (2019), i.e N(0.4,0.1)). For the relaxation time $\tau$, we use the normal distribution of the fast time response of the "two-box" exponential model that corresponds to $h = 1$, found by Geoffroy et al. (2013) for a suite of 12 CMIP5 GCMs: N(4yrs,2yrs), with the standard deviation doubled of the original work so as to be a weakly informative prior. When considering the aerosol scaling parameter, $\alpha$, we take the prior distribution to be a normal distribution, N(1.00,0.55) which has a 90% CI and mean coherent with the IPCC AR5 best range for the modern value of aerosol forcing, $F_{Aer} \approx -1.0 W m^{-2}$, in the series we used. For the remaining two parameters, $s$ and $\nu$, we assume non-informative uniform priors over the range of parameters; $s \in [1.0, 4.0]$ and $\nu \in [0.0, 1.0]$. All prior distributions are independent.

Using Bayes, eq. 20, we then fit a multivariate Gaussian distribution to our five-dimensional parameter space, posterior distribution $Pr(s, h, \tau, \alpha, \nu | T(t))$, which will be used to draw sets of parameters to generate future forced temperature projections. The multivariate Gaussian approximation is built by using the means and variances of all parameters through integrating the joint probability to obtain five marginal probabilities, and calculating the covariance between all pairwise parameters using their "joint" marginal distributions as to take into account potentially large correlations between parameters. The five dimensional posterior parameter space, $(s, h, \tau, \alpha, \nu)$ is thus defined by a multivariate normal distribution:

$$P(\boldsymbol{x}; \boldsymbol{\mu}, \boldsymbol{\Sigma}) = \frac{1}{(2\pi)^{\frac{5}{2}} |\boldsymbol{\Sigma}|^{\frac{1}{2}}} e^{-(\boldsymbol{x} - \boldsymbol{\mu})^t \boldsymbol{\Sigma}^{-1} (\boldsymbol{x} - \boldsymbol{\mu})/2}, \tag{21}$$

where $x = \{s, \tau, h, \alpha, \nu\}$, the vector of the means is $\mu$ and the 5×5 covariance matrix $\Sigma$.

## 3 Results

Using Bayes' theorem as described above, we derive probability density functions (PDFs) for the model and forcing parameters of the FEBE from the mean likelihood functions of the five observational datasets. The different observational datasets are treated as dependent due to the use of overlapping raw data, with the differences between series coming partly from the different processing of the raw data by different teams. This corresponds to putting the datasets into a Bayesian framework where each has equal a priori probability: HadCRUTv4, C&W, GISTEMP, NOAAGlobalTemp and BEST ($n = 5$).

$$Pr(s, h, \tau, \alpha, \nu | T(t)) = \frac{1}{n} \sum_{i=1}^{n} Pr(s, h, \tau, \alpha, \nu | T_i(t)). \tag{22}$$

Following IPCC methodologies, we report the "very likely" credible interval at the 90% credible level throughout this work along with median estimates for the all ensemble spreads. The complete suite of model and forcing parameters and climate sensitivities are summarized in tables 1 and 2. In addition we include a comparison of the same parameters for the Half-order EBE (HEBE) ($h = 1/2$) that is a consequence of the classical continuum heat equation (Lovejoy, 2021a, b), as well as with the precursor Scaling Climate Response Function (SCRF) model (Hébert et al. (2021)) which differs primarily in the treatment of high frequencies in table 3. The FEBE value of $h$ is slightly less than $1/2$ corresponds to the fractional heat equation.

### 3.1 The Model: Green's Function Parameters: $h, \tau$

#### 3.1.1 The Scaling Exponent $h$

The model is characterized by $h$ and $\tau$, where the exponent $h$ of the FEBE is the most fundamental. For $h$, we found a 90% CI of $[0.33, 0.44]$, with a median value of 0.38 when using $F_{Aer_{RCP}}$, and while using $F_{Aer_{SSP}}$ we found a similar median of 0.38 with 90% CI of [0.32,0.44]. We can already note that it is close to the HEBE value $h = \frac{1}{2}$ and other empirical estimates for power law impulse Green's functions ($G(t) \approx t^{-H_F - 1}$) with $h = -H_F \approx 0.5_{+0.5}^{-0.4}$ (Lovejoy et al., 2017; Hébert et al., 2021). The NOAA dataset differs the most from all others, the exact cause of the difference is not clear although it arises from the MLOST dataset's use of a complex frequency algorithm with low-frequency tuning (Smith et al., 2008). This low-frequency tuning along with the spatio-temporal smoothing applied in the MLOST dataset is likely the cause of a slightly higher $h$ (i.e. a smoother temperature series).

#### 3.1.2 The Relaxation Time $\tau$

The second model parameter is the relaxation time $\tau$ that characterizes the approach to equilibrium. From the point of view of parameter estimation $\tau$ is a difficult parameter to determine since it is inversely correlated with $s$: a large $\tau$ can be somewhat compensated by a smaller $s$ and vice versa. We obtained the a posteriori median value of 4.7 years and 90% CI of [2.4,7.0] years when using $F_{Aer_{RCP}}$, and nearly identical results using $F_{Aer_{SSP}}$.

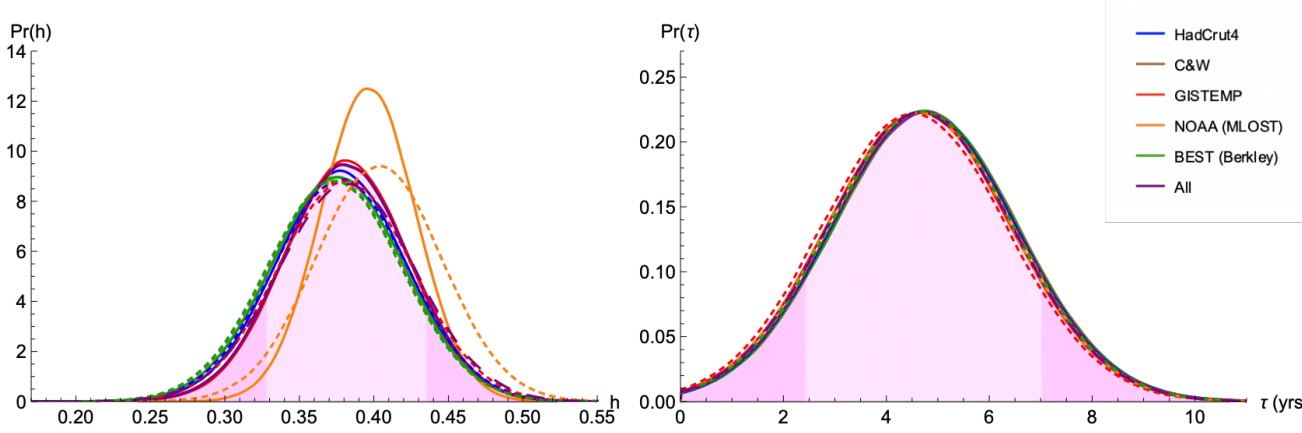

**Figure 3.** For each observational dataset and their average, PDFs are shown for the model parameters: the scaling parameter $h$ (left), and the transition time $\tau$ (right). Shown are the PDFs for parameter estimation based on both $F_{Aer_{RCP}}$ (solid) and $F_{Aer_{SSP}}$ (dashed). The average PDFs of the five observation datasets using $F_{Aer_{RCP}}$ is shown as the main result with shading, with darker 5% tails.

Presented in figure 4 (top) are the step-response Green's function, $G_1(h,\tau;t)$, of the FEBE with the parameters $h$ and $\tau$ along with its 90% CI, shown alongside the IPCC two-box model Green's function (IPCC, 2013; Held et al., 2010; Geoffroy
et al., 2013). Considering $G_1(t)$ (blue), at scales below a few years where the box models or the Hébert et al. (2021) truncated scaling model are smooth, the FEBE has a singular response. This enables it to reproduce the statistics of the internal variability as well as to be more sensitive to volcanic forcings. Even up to scales of 25 years, the $G_1$ (blue) responds much faster than the IPCC (black), yet the approach to the asymptotic value 1 corresponding to energy balance is substantially slower. This can also be seen in the ramp-response Green's functions, $G_2$, the integral of $G_1$ (bottom). For comparison, each was normalized
by the value at 70 years - the standard ramp time for TCR (Collins et al., 2013). At multi-year resolution (ignoring the high frequency variability), over the scale of the anthropocene there is little difference between the FEBE and IPCC, with FEBE having a more gradual response. This contributes to the somewhat cooler FEBE centennial scale projections when compared with those from the two-box model.

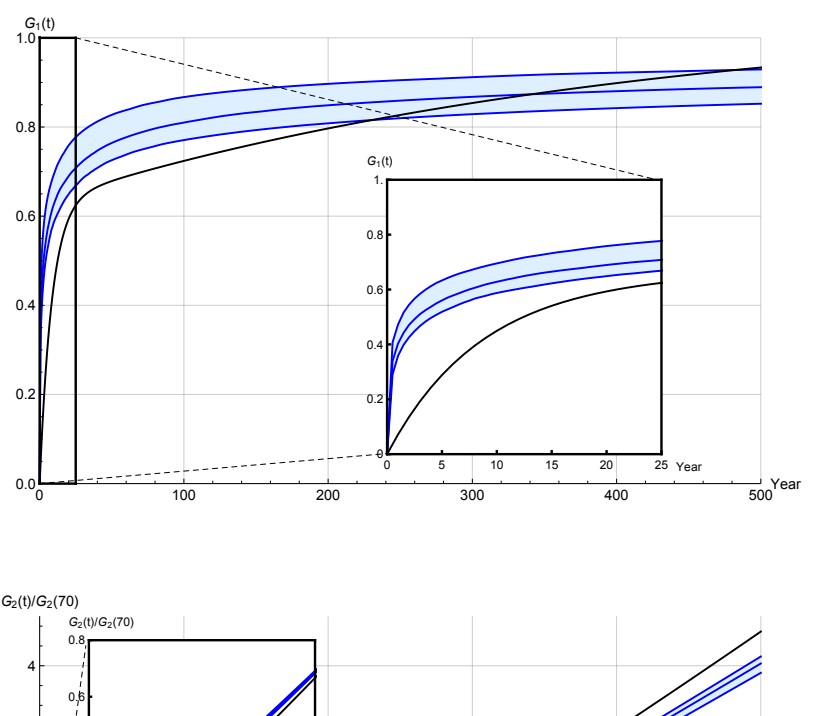

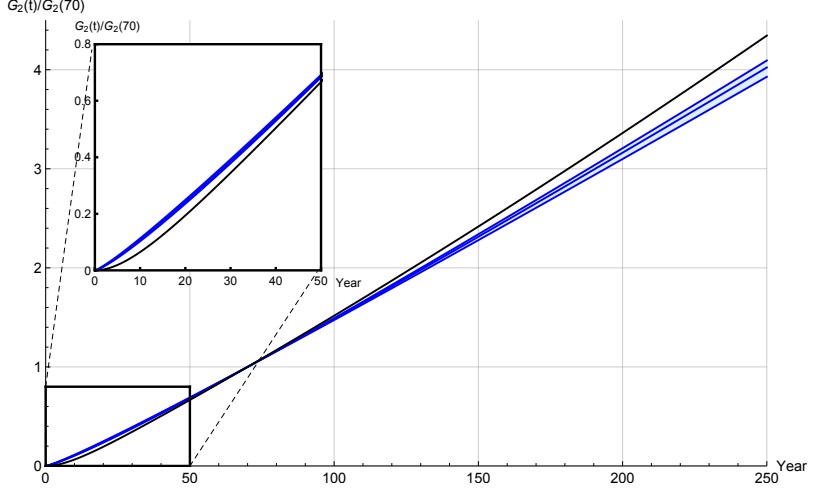

**Figure 4.** (top) The median and 90% CI of the FEBE step-response Green's function, $G_1(h,\tau;t)$, compared to the IPCC two-box model Green's function (black). (bottom) The median and 90% CI of the FEBE normalized ramp-response Green's function, $G_2(h,\tau;t)$, compared to the IPCC two-box model Green's function (black).

## 3.2 Characterizing the Forcing

### 3.2.1 Aerosol Linear Scaling Factor $\alpha$

The aerosol linear scaling factor $\alpha$ that effectively re-calibrates the aerosol forcing (figure 5 left, solid line) was found to have a median value of 0.6 with a 90% CI of [0.2,1.0] for the CMIP5 $F_{Aer_{RCP}}$ series. However when using the CMIP6 sulphate

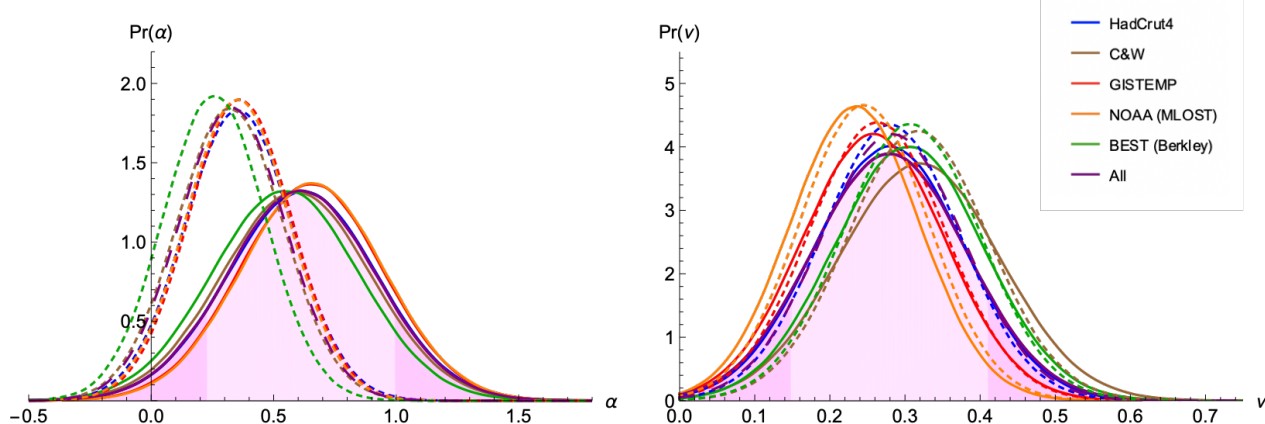

**Figure 5.** For each observational dataset and their average, PDFs are shown for the forcing parameters: the aerosol scaling factor $\alpha$ (left), and the volcanic intermittency correction exponent $\nu$ (right). Again, shown are the PDFs for parameter estimation based on both $F_{Aer_{RCP}}$ (solid) and $F_{Aer_{SSP}}$ (dashed). The average PDFs of the five observation datasets using $F_{Aer_{RCP}}$ is shown as the main result with shading, with darker 5% tails

emissions based aerosol forcing series, $F_{Aer_{SSP}}$, we find support for a weaker and better constrained aerosol forcing, recalibration $\alpha$ with a median of 0.33 and 90% CI of [0.05, 0.61] (figure 5 left, dashed line). In both cases an aerosol recalibration factor of 1 corresponds to the modern (2005) aerosol forcing value of about $-1.0 W m^{-2}$, but we find in both cases that $\alpha < 1$. The result from two independent aerosol forcing series again shows that the forcing associated with aerosols is still widely uncertain and overpowered, supporting post-AR5 studies that found aerosol forcings simulated by GCMs were unrealistic (Zhou and Penner, 2017; Sato et al., 2018; Bellouin et al., 2020), and that aerosol forcing was weaker when climate feedbacks were allowed (Nazarenko et al., 2017).

### 3.2.2 Volcanic Intermittency Correction Exponent $\nu$

The volcanic intermittency correction exponent $\nu$ was found to have a posterior median value of 0.28 with 90% CI of [0.15,0.41] when using $F_{Aer_{RCP}}$ and similar median value 0.28 with 90% CI of [0.16,0.40] when using $F_{Aer_{SSP}}$ (recall $\nu = 0$ implies a constant mean forcing and the original series is recovered with $\nu = 1$). Both contain the theoretically calculated $\nu$ within their 90% CI ($\nu = 0.32$). This result confirms that volcanic forcing is generally overpowered since $\nu = 1$ has nearly null probability as seen in figure 5. Thus, the original volcanic series described without the intermittency correction does not reproduce well, within the FEBE model presented, the cooling events observed in instrumental records following eruptions: volcanic cooling would be overestimated. As noted in the case for the exponent, $h$, the NOAA dataset noticeably differs from the others, the spatio-temporal smoothing applied in the MLOST dataset is likely the cause of a lower $\nu$ (i.e. a smoother volcanic forcing).

In figure 6 we compare the total forcing series, $F_{Tot}(t)$ (black), IPCC AR5, eq. (16) where $\alpha = \nu = 1$, with the adjusted forcing series, $F_{Tot}(\alpha, \nu; t)$ (blue). During the historical period, the intermittency and strength of the strong volcanic events

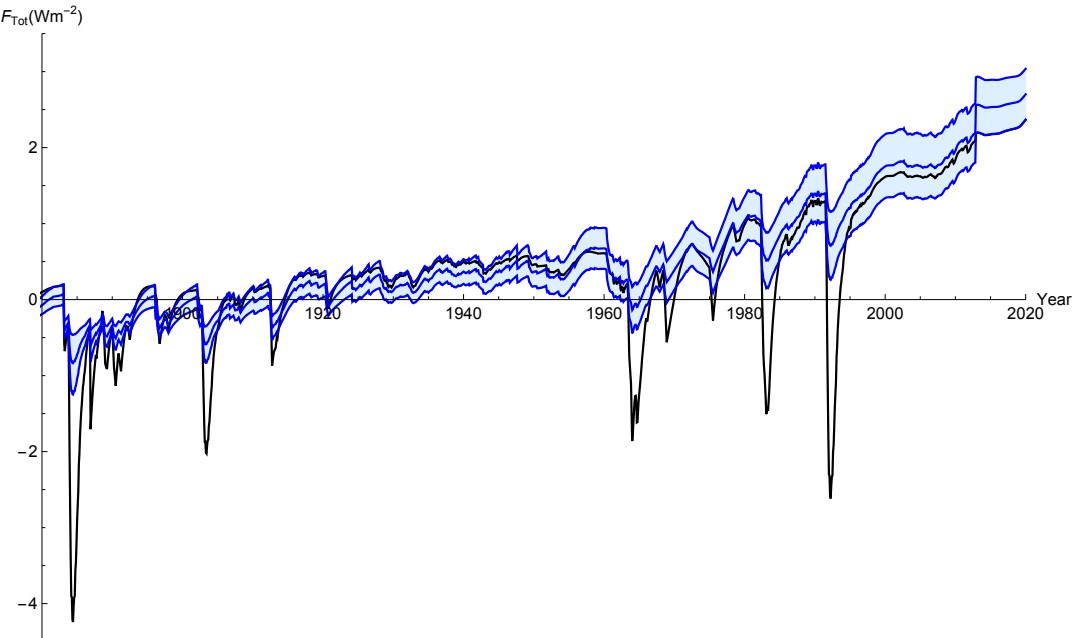

**Figure 6.** The total historic (1880–2020) forcing series prescribed by the IPCC using, $F_{Aer_{RCP}}$ (black) compared to the adjusted forcing, $F_{Tot}(\alpha, \nu; t)$ (blue) which takes into account aerosol and volcanic corrections, shown with 90% CI.

is greatly reduced and in the recent past the median adjusted forcing series is higher than the unadjusted forcing due to the reduced aerosol forcing strength. This adjusted forcing series consequently contributes to a lower climate sensitivity, presented in the following section, due to the historic negative forcings of volcanoes and aerosols being adjusted to closer match historical observations, eliminating the need for a high climate sensitivity to compensate.

### 3.3 Climate Sensitivity

#### 3.3.1 Climate Sensitvity Parameter, $s$

The climate sensitivity parameter $s$, refers to the equilibrium change in the annual GMST following a unit change in radiative forcing. Its inverse is the climate feedback parameter, the increase in radiation to space per unit of global warming. We find $s$ to have a median value of $0.56\ K(Wm^{-2})^{-1}$ with 90% CI $[0.45,0.67]\ K(Wm^{-2})^{-1}$ using $F_{Aer_{RCP}}$, and when using $F_{Aer_{SSP}}$ we find median $0.52K(Wm^{-2})^{-1}$ with 90% CI $[0.43,0.61]\ K(Wm^{-2})^{-1}$ (figure 7). Both on the lower end of the CMIP5 MME climate sensitivity parameter of median $1K(Wm^{-2})^{-1}$ and 90% CI $[0.5, 1.5]K(Wm^{-2})^{-1}$ but within the 90% CI. Although both estimates are below the CMIP6 MME 90% CI $[0.63, 1.50]K(Wm^{-2})^{-1}$, with a median of $0.92K(Wm^{-2})^{-1}$ which has been criticized as being too high (Zelinka et al., 2020; Tokarska et al., 2020; Flynn and Mauritsen, 2020).

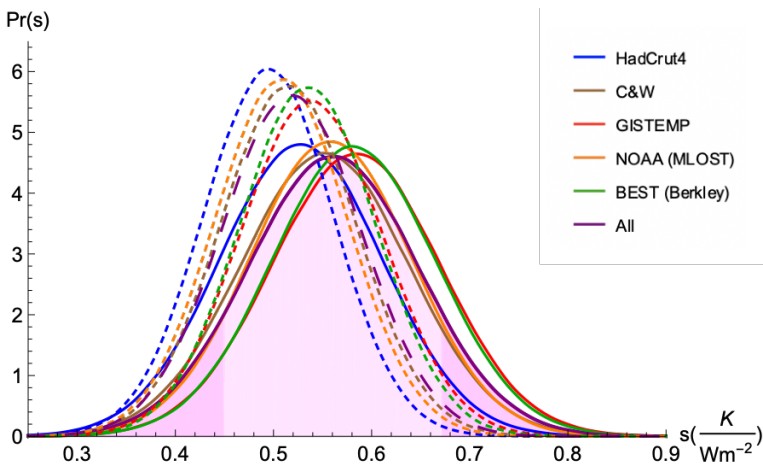

**Figure 7.** For each observational dataset and their average, PDFs are shown for the climate sensitivity parameter $s$ (the ECS, here in units of $K(Wm^{-2})^{-1}$), $F_{Aer_{RCP}}$ (solid) and $F_{Aer_{SSP}}$ (dashed).

### 3.3.2 Equilibrium Climate Sensitivity

Two standard types of climate sensitivity used for inter-model comparisons: Equilibrium Climate Sensitivity (ECS) and Transient Climate Response (TCR) - our results are summarized in table 2.

     If atmospheric $CO_2$ was increased to double pre-industrial concentrations and then held there, the planet would only slowly reach a new thermodynamic equilibrium. This delay is largely because the world's oceans take a long time to heat up in response to the enhanced greenhouse effect. The Equilibrium Climate Sensitivity (ECS) is the amount of warming achieved

when the entire climate system reaches 'equilibrium' or the steady-state temperature response to a doubling of $CO_2$. By the definition of the temperature response to external forcings in eq. 7, the climate sensitivity parameter is the equilibrium climate sensitivity. The two are equivalent to within a constant factor: the number of $Wm^{-2}$ per $CO_2$ doubling, the standard value being $3.71Wm^{-2}/(CO_2$ doubling) (IPCC, 2013).

     The PDF for ECS shown in figure 8 (left), for both aerosols series was found to have a 90% CI of $[1.6, 2.4]K$ and a median

value of $2.0K$ when using $F_{Aer_{RCP}}$, and median of $1.8K$ and 90% CI [1.5,2.2] using $F_{Aer_{SSP}}$ (see table 2). These results are lower than those found in the CMIP5 MME which had a best value of $3.2K$, but our 90% CI bounds are more narrow, laying within the CMIP5 MME range of $[1.9, 4.5]K$. Although when we consider the expanded ECS 90% CI of $[1.5, 4.5]K$ considered in IPCC (2013), which takes into account both the CMIP5 MME and historical estimates, we see that the FEBE estimates are wholly within this range and much less uncertain. For the CMIP6 MME which has a 90% CI of $[2.0, 5.5]K$ and

mean estimate $3.7K$, our best estimate using the corresponding $F_{Aer_{SSP}}$ is slightly below the lower credible due to the upward shift of ECS estimates seen in CMIP6 models (Zelinka et al., 2020), but again has a more narrow CI.

### 3.3.3 Transient Climate Response

Conventionally, TCR quantifies the temperature change that would occur if $CO_2$ levels increase by $1\%$ (compounded) per year until they double ($\approx 70$ years). Since the $CO_2$ forcing is logarithmically dependent on $CO_2$ concentration, the TCR is then simply the global temperature increase that has occurred at the point in time that a linearly increasing forcing reaches double pre-industrial levels.

The derived PDFs for TCR are shown in figure 8 (middle) and summarized in table 2. Our TCR was found to have a $90\%$ CI of $[1.2, 1.8]K$ with a median of $1.5K$ when using $F_{Aer_{RCP}}$, while when using $F_{Aer_{SSP}}$ we find a median 1.4K and $90\%$ CI of [1.1,1.6]K. Both estimates are lower and more constrained, but within the $90\%$ CI given by the CMIP5 MME: a $90\%$ CI of $[1.2, 2.4]K$ and a best value of $1.8K$, and by the CMIP6 MME: $90\%$ CI of $[1.2, 2.8]K$ with best value of $2.0K$.

The ECS and TCR estimates using the SSP scenarios with the FEBE are lower than those using RCP due to the overly strong aerosols over the historical period in the SSPs which require a lower aerosol linear factor along with lower ECS to best match the historical temperature record. The difference between the shape of the RCP and SSP aerosol forcing can also account for this.

The TCR-to-ECS ratio is a non-dimensional measure of the fraction of committed warming already realised after a steady increase in radiative forcing, in this case a doubling of $CO_2$, this quantity is generally referred to as realised warming fraction (RWF) (Stouffer, 2004; Solomon et al., 2009; Millar et al., 2015); it is a nondimensional memory parameter. A model with a low RWF will indicate that global warming may continue for centuries after emissions have stopped. We present the TCR-to-ECS ratio in figure 8 (right), having a $90\%$ CI [0.70, 0.78] and median 0.73 using $F_{RCP}$ parameters. Similar results are found using $F_{SSP}$ parameters, a median of 0.72 and $90\%$ CI [0.71, 0.79]. From figure 8 and table 2, we see that the TCR-to-ECS ratio is higher than both generations of MME $90\%$ CI, a consequence of lower ECS and TCR values, and similar uncertainty.

In the next section we show that with a lower and more constrained climate sensitivity parameter (figs. 7 and 8), the adjusted forcings (figure 6) and long memory process of the FEBE produce future projections that tend to be cooler than the CMIP5/6 projections, yet remain within their $90\%$ CI.

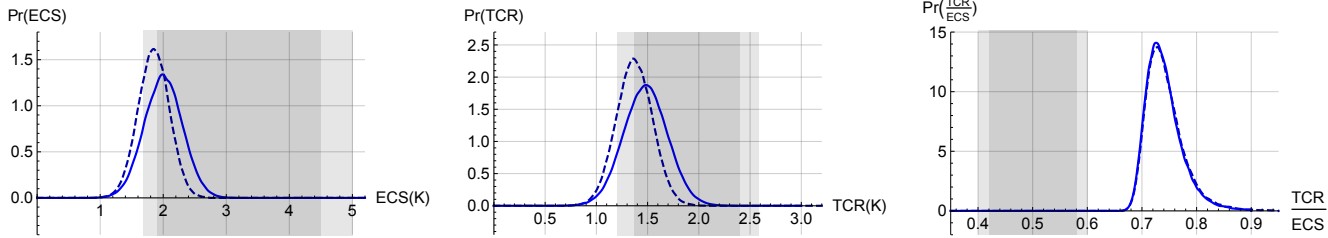

**Figure 8.** The PDFs for ECS (left), TCR (middle) and the TCR:ECS ratio (right) are derived using $F_{Aer_{RCP}}$ (solid) and $F_{Aer_{SSP}}$ (dashed). The associated $90\%$ CI (bars under the axis), the CMIP5 MME $90\%$ CI (dark gray shading), and the CMIP6 MME $90\%$ CI (light gray shading).

**Table 1.** Model and Forcing parameter medians for FEBE calibrated over the historical period (1880–2020) using $F_{Aer_{RCP}}$ and $F_{Aer_{SSP}}$, along with their corresponding 90% credible intervals.

| | Median $h$ | $h$ 90% CI Range | Median $\tau$ [years] | $\tau$ 90% CI Range [years] | Median $\alpha$ | $\alpha$ 90% CI Range | Median $\nu$ | $\nu$ 90% CI Range | Median $s$ $\left[\frac{K}{Wm^{-2}}\right]$ | $s$ 90% CI Range $\left[\frac{K}{Wm^{-2}}\right]$ |
|---|---|---|---|---|---|---|---|---|---|---|
| $F_{Aer_{RCP}}$ | 0.38 | [0.33, 0.44] | 4.7 | [2.4, 7.0] | 0.6 | [0.2, 1.0] | 0.28 | [0.15, 0.41] | 0.56 | [0.45, 0.67] |
| $F_{Aer_{SSP}}$ | 0.38 | [0.32, 0.44] | 4.7 | [2.4, 7.0] | 0.33 | [0.05, 0.61] | 0.28 | [0.16, 0.40] | 0.52 | [0.43, 0.61] |

**Table 2.** The calculated ECS and TCR medians using both parameters corresponding to $F_{Aer_{RCP}}$ and $F_{Aer_{SSP}}$, along with their corresponding 90% credible intervals.

| | Median TCR [K] | TCR 90% CI Range [K] | Median ECS [K] | ECS 90% CI Range [K] | Median TCR/ECS Ratio | TCR/ECS Ratio 90% CI Range |
|---|---|---|---|---|---|---|
| $F_{Aer_{RCP}}$ | 1.5 | [1.2, 1.8] | 2.0 | [1.6, 2.4] | 0.73 | [0.70, 0.78] |
| $F_{Aer_{SSP}}$ | 1.4 | [1.1, 1.6] | 1.8 | [1.5, 2.2] | 0.74 | [0.71, 0.79] |

**Table 3.** Model and Forcing parameter medians using $F_{Aer_{RCP}}$ for FEBE, the classical continuum mechanics HEBE ($h = \frac{1}{2}$) and the SCRF model (Hébert et al., 2021) calibrated over the historical period, along with their corresponding 90% credible intervals.

| | Median $h$ | $h$ 90% CI Range | Median $\tau$ (years) | $\tau$ 90% CI Range (years) | Median $\alpha$ | $\alpha$ 90% CI Range | Median $\nu$ | $\nu$ 90% CI Range | Median ECS [K] | ECS 90% CI Range [K] |
|---|---|---|---|---|---|---|---|---|---|---|
| FEBE | 0.38 | [0.33, 0.44] | 4.7 | [2.4, 7.0] | 0.6 | [0.2, 1.0] | 0.28 | [0.15, 0.41] | 2.0 | [1.6, 2.4] |
| HEBE | 1/2 | - | 4.7 | [2.4, 7.0] | 0.48 | [0.10, 0.86] | 0.33 | [0.16, 0.51] | 1.8 | [1.4, 2.3] |
| SCRF | 0.5 | [0.3, 0.7] | 2.0 | - | 0.8 | [0.1, 1.3] | 0.55 | [0.25, 0.85] | 2.3 | [1.8, 3.7] |

## 4  Discussion

With the above collection of model and forcing parameter probability distributions, the FEBE was used to reconstruct the temperature over the historical period, as well as make projections of the forced temperature response for the coming century using forcings prescribed by the RCP and SSP scenarios.

The CI provided for the MME corresponds to the spread between the different GCMs, "structural uncertainty", while for the FEBE it is parametric uncertainty (Bretherton, 2012). In both cases, the projections are deterministic but with uncertainty limits due to their respective model uncertainties. Both yield an estimate of the forced response but with qualitatively different uncertainty bounds.

For the FEBE, the spread of the forced projections is purely from the uncertainty in the parameters: the contribution to uncertainty from internal variability has been averaged out (it is effectively the average over an infinite ensemble of realizations of internal variability). In order to make projections we therefore draw samples of parameters from the (correlated) multidimensional parameter space (approximated by the multivariate normal distribution in eq. 21), by using a Monte Carlo method. Once a random set of parameters has been chosen, realizations of the forced temperature response are generated using eq. 3 and a numerical convolution. It should be noted this Monte Carlo sampling is simply a convenient numerical technique for performing high dimensional probability space integrals, it does not imply any stochasticity in the projections which although are parametrically uncertain, nevertheless are deterministic responses to purely deterministic forcing. However, the Monte Carlo methods do introduce standard Monte Carlo numerical uncertainty, but this was made quite small by using a large number (500) of Monte Carlo realizations. Once we have our ensemble of projections, we remove the pre-industrial baseline (such that the temperature anomaly over 1880–1910 is zero) and calculate the desired credible intervals of the forced response. We consider the historical period coinciding with the range of observation temperature records (1880-2020) and make all comparisons to this period, acknowledging that the CMIP5 GCMs historical reconstruction ended in 2005 and for the CMIP6 GCMs in 2014.

## 4.1   Reliability and Historical Reconstructions: 1880–2020

In this section we present the full historical reconstruction using the FEBE observation-based projections with those from the GCMs in the CMIP5/6 MME. In order to make a proper comparison with data we must include both the forced deterministic temperature response, with its purely parametric uncertainty, as well as the internal variability of the mean observational temperature series, estimated to be $\approx \pm 0.14°\text{C}$ (monthly resolution). The two uncertainties were combined assuming the statistical independence of the internal forcing and the parametric uncertainty: the errors therefore add in quadrature.

An important characteristic of probabilistic forecasts is their reliability that quantifies the difference between the forecast and actual probability distributions. Consider for example, a set of predictions derived from ensemble forecasts. In some realizations it is predicted that the chance of above-average seasonal-mean temperature for the coming season will be 70%. If the probabilistic forecast system is reliable, then one can expect that in 70% of these predictions the actual seasonal-mean temperature will be above average (Annan and Hargreaves, 2010; Weisheimer and Palmer, 2014). In figure 9, we can verify the reliability of the FEBE. We see that as expected, the temperature observations fall closely within the 90% CI of the FEBE historical reconstruction (i.e. the ensemble average of the response to both internal and external forcing). More precisely, at the monthly resolution in figure 9, the historical mean temperature (red) is within the 90% CI of the FEBE forced response (with internal variability added) 89.9% of the months using the RCP scenario (left) or 90.2% of the months using the SSP scenario (right). The accuracy of this uncertainty verifies both the underlying model and Bayesian parameter estimation method.

This is expected for a reliable model and is an analogous validation of probabilistic aspects of the projection as unlike weather forecasts where we have many past test cases, climate change projections cannot be calibrated in the same manner (Stainforth et al., 2007; Tebaldi and Knutti, 2007; Knutti et al., 2010). In both reconstructions it is possible that the end of the second world war (1945) temperature spike which lies out of the FEBE 90% CI may be explained due to biases associated with bucket and engine room intake measurements (Chan and Huybers, 2021).

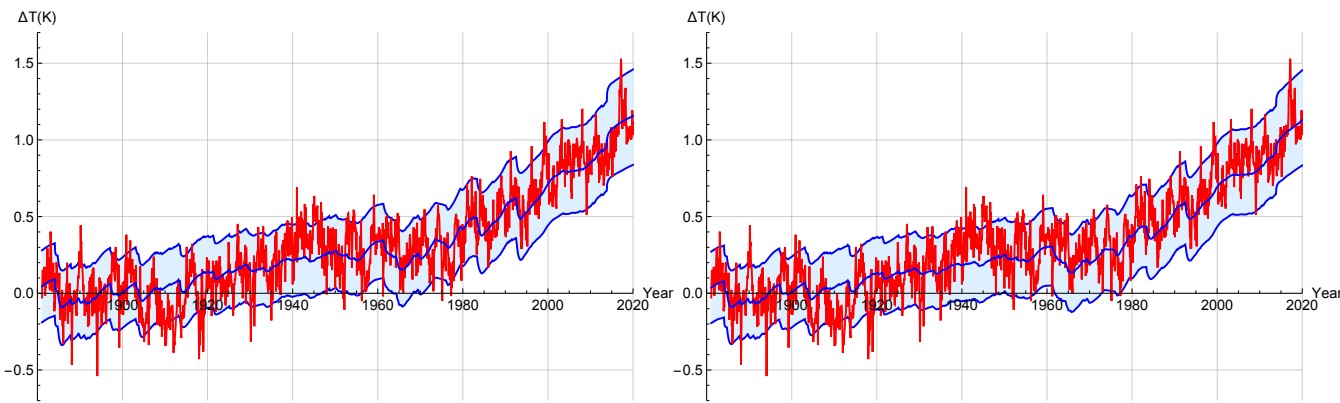

**Figure 9.** (left) The historical reconstruction (forced temperature response and internal variability) of the FEBE, with parameters calibrated using $F_{Aer_{RCP}}$ (blue) alongside mean of 5 observational temperature series (red) at monthly resolution; 90% CI (due to parametric uncertainty and internal variability) are indicated (shaded). (right) Same as left except using $F_{Aer_{SSP}}$ parameters and forcing.

## 4.2 The Amplitude of the Internal Forcing

The small scale limit of the validity of the FEBE is not known, although it is likely to be $\sim$1 month (roughly the weather-macroweather transition time scale). Justification comes from the success of the high frequency FEBE limit that successfully forecasts monthly and seasonal temperatures (Del Rio Amador and Lovejoy, 2019, 2021a, b). As discussed earlier (eq. 14) the FEBE predicts the (stochastic) response to the internal forcing. The standard deviation $\sigma_f$ of $f(t)$ is the amplitude of the internal forcing assumed to be a Gaussian white noise, which can be estimated using eqs. 14, and 15, and $\sigma_T \approx \pm 0.14°$C. Using our $F_{RCP}$ (and $F_{SSP}$) parameter estimates, we find a mean estimate of the forcing standard deviation, $\sigma_f$, to be 3.2 $Wm^{-2}$ (3.3 $Wm^{-2}$) and 90% CI of $[2,1,4.2]$ $Wm^{-2}$ ($[2.3,4.3]$ $Wm^{-2}$) (at a monthly resolution). If we introduce a white noise forcing with $\sigma_f$ amplitude the FEBE recreates the amplitude of the internal temperature variability response and its change as a function of time scale/resolution.

This estimate of the internal variability forcing can be compared with that of Harries and Belotti (2010) who examine the net energy flux balance at the top of atmosphere (TOA) measured using observations from polar-orbiting spacecraft (at monthly scale). The early observations, using the Nimbus experiments, show an internal variability of the $4.1 \pm 4.0 Wm^{-2}$, while more modern measurements (CERES) in the 2000s show variability of between $\pm 2$ and $\pm 4$ $Wm^{-2}$. Thus our estimate of the internal forcing variability is within estimates of the TOA net energy flux balance.

### 4.3 Statistical Evaluation of the FEBE

As with GCMs, the FEBE predicts both the forced deterministic response as well as the statistical properties of the internally driven stochastic part. We can therefore evaluate the accuracy of the stochastic part by comparing the FEBE temeprature statisics with those from observational time series. It was already shown in Lovejoy et al. (2021) that using a simple "ramp" model that included the deterministic external and stochastic internal variabilities the FEBE roughly predicts both high- and low-frequency scaling regimes. In figure 10 (left) we show one realization of the full FEBE, including the deterministic and stochastic forcings, with median parameters calibrated earlier using $F_{Aer_{RCP}}$ (blue) and $F_{Aer_{SSP}}$ (light blue); the five observation temperature series are shown alongside (gray - shifted up). We compare the model statistics with the 5 globally averaged temperature series using their root-mean-square Haar fluctuations, shown in figure 10 (right). The Haar fluctuation for a series $T(t)$, $\Delta T(\Delta t)$ is the difference between the average of the first and second halves of the interval $\Delta t$. This is a convenient way to characterize variability as a function of time scale in real space, valid for increasing or decreasing average fluctuations. By applying global scale Haar fluctuation analyses Del Rio Amador and Lovejoy (2019) found $H \approx -0.1$ corresponding to $h = H + 1/2 \approx 0.4$.

Below Milankovitch time scales, there are three main scaling regimes observed in the atmosphere: the weather, macroweather and climate Lovejoy (2013). In the macroweather regime, longer than the lifetime of planetary structures ($\sim 10$ days), temperature fluctuations decrease with scale until a transition probably occurs to the climate regime where fluctuations begin to increase. In the industrial epoch this scale is $\sim 20$ years, while in the pre-industrial epoch this scales transition occurs at centuries or millennia Lovejoy (2015b). Over the scale of 1 year to about 10 years (the macroweather regime), the FEBE and the observational temperature series have an approximate slope (indicated by the straight reference line in figure 10) of $h \approx 0.4$. We see a transition in both the FEBE and observations at $\Delta t \gtrsim 10$ years: the transition to the climate regime where fluctuations begin to increase with scale. The fact that the FEBE's fluctuations at the climate regime track the observational data strongly supports the realism of the FEBE for multidecadal projections.

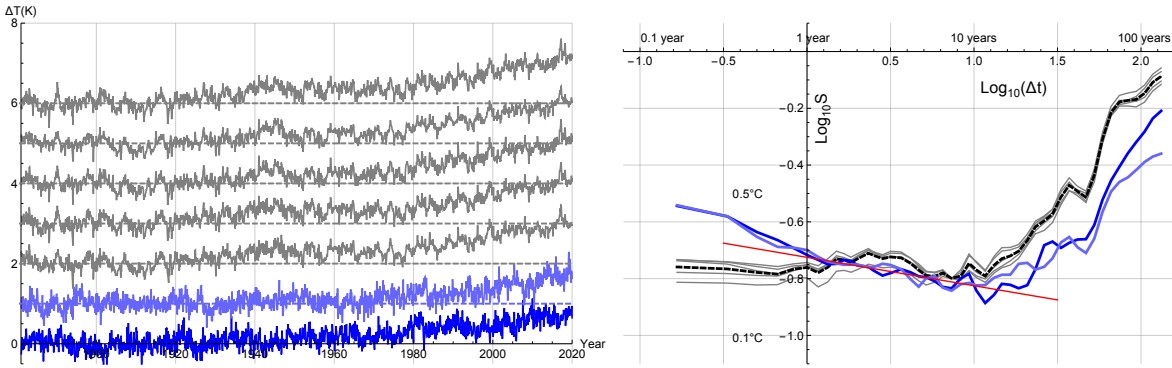

**Figure 10.** (left) The historical reconstruction (forced and internal temperature response) of the FEBE, with parameters calibrated using $F_{Aer_{RCP}}$ (blue) and $F_{Aer_{SSP}}$ (light blue) alongside the 5 observational temperature series (gray - shifted up) at monthly resolution. (right) The Root Mean Square Haar fluctuation structure function $S(\Delta t) = \langle \Delta T(\Delta t)^2 \rangle^{\frac{1}{2}}$ for FEBE reconstruction using $F_{Aer_{RCP}}$ (blue) and $F_{Aer_{SSP}}$ (light blue), and the five globally averaged monthly-resolution temperature time series (gray; mean is shown in dashed black). The reference (red) line has the slope of the approximate median estimate of the scaling exponent $h \approx 0.4$ ($H = h - \frac{1}{2} \approx -\frac{1}{10}$). The stochastic (internal variability) is not expected to be identical; only its statistical character (correlations) and amplitude are expected to be the as the data.

### 4.4 Evaluating the FEBE using Hindprojections Including the Slowdown

We have shown that the FEBE hindcasts are reliable (Sect. 4.1), that they have realistic internal forcings (Sect. 4.2) and realistic statistical variability (Sect. 4.3). Here we evaluate their deterministic responses using hindprojections.

    Unlike the comparison in figure 10 that included the internal variability in order to evaluate the reliability, the following figures are estimates of the ensemble averaged hindprojections i.e. with the internal variability averaged out completely. This is not a reliability check at 1 year resolution, as shown in the prior section, so we do not expect the FEBE to be in the data

range 90% of the time. Rather, the percentage of the time that the FEBE is in the data range is a measure of hindprojection/data agreement about the deterministic forced response part. It is therefore appropriate to compare this with the MME. In figure 11, we compare the 90% CI of the historical temperature observations with the median forced response of both the FEBE using the RCP (left) and SSP (right) historical forcing compared to both the CMIP5 (left) and CMIP6 (right) MMEs. In the inset of figure 11, we show the slowdown ("hiatus') period (1998-2014).

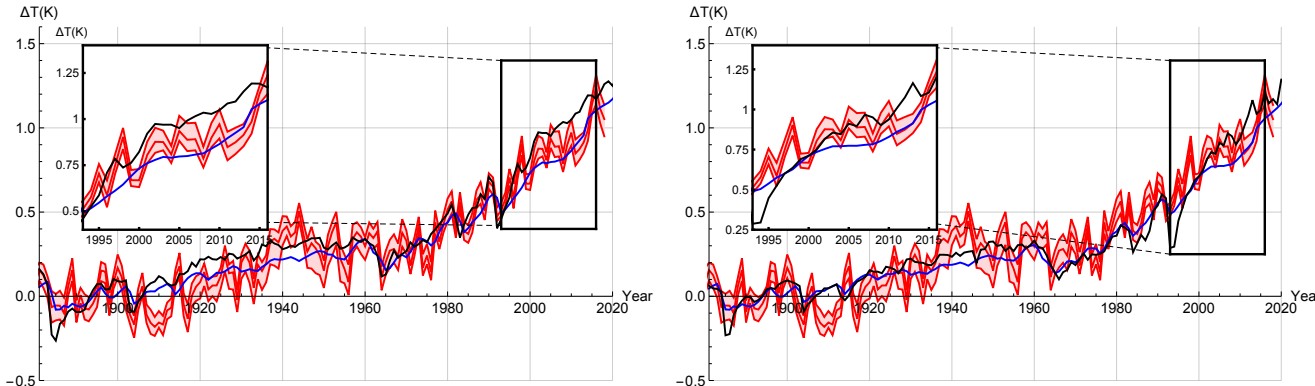

**Figure 11.** (left) The median historical forced component of the FEBE, with parameters calibrated using $F_{Aer_{RCP}}$ (blue), and the median of the CMIP5 MME (black) alongside mean of 5 observational temperature series (red) with their 90% CI indicated (shaded). (right) The median historical forced component of the FEBE, with parameters calibrated using $F_{Aer_{SSP}}$ (blue), and the median of the CMIP6 MME (black) alongside mean of 5 observational temperature series (red) with the 90% CI indicated (shaded).

550 Throughout the historical period, the hindprojection of the FEBE and the median of the CMIP5/6 MME are close. Between 1915–1960 the CMIP5/6 MME is consistently warmer than the FEBE hindprojection and historical temperature records, although generally by less than 0.05K. The slowdown in global warming during the first decade of the 21st century, termed as the slowdown ("hiatus") (Kaufmann et al., 2011; Meehl et al., 2011; Medhaug et al., 2017), is tracked closely by the FEBE hindprojection while the CMIP5/6 MME overshoots (by $0.1K$ to $0.2K$), a well studied divergence between GCMs and ob-

555 servations, shown in figure 11 (insets). This supports (Lovejoy, 2015a, b) which found that the slowdown ("hiatus") could be well predicted by a stochastic fGn model (comparable with the present hindprojection) and concluded the issue to be GCM overprojection.

 Following the monthly resolution reliability confirmation in section 4.1 we can now perform a quantitative comparison between the amount of time the FEBE and CMIP5/6 MME median response is within the bounds of the observational tempera-

560 ture series 90% CI performed with annual resolution data. The median FEBE hindprojection using $F_{Aer_{RCP}}$ is within the 90% CI of the observational temperature series over the whole historic period 47% of the years and over the slowdown is within 70% in comparison to the CMIP5 MME median which is within the whole historic period only 39% and over the slowdown 17%. While the median FEBE hindprojection using $F_{Aer_{SSP}}$ similar results are found, over the whole period: 45% and over the slowdown 35%, in comparison to the CMIP6 MME median which is within the whole historic period 39% and over the

565 slowdown is 30%. In can be seen in both cases that the CMIP MME is generally warmer than the FEBE forced component notably over the period of the slowdown. We see that indeed, the FEBE median forced component in both cases captures the slowdown rather accurately.

## 5 Projections through to 2100

### 5.1 The FEBE and GCM MME Comparisons

We now consider the deterministic (infinite ensemble) FEBE projections to 2100. At first, the temperature increase in each case is nearly identical; the future pathways only diverging into their respective scenarios roughly two decades after their beginning (RCPs begin in 2005, SSPs begin in 2014). Further into the future, the warming rate begins to depend more on the specified scenario, the highest being in RCP 8.5/SSP 5-85 (figure 12 c, f) while they significantly lower in RCP 2.6/SSP 1-26 (figure 12 a, d; tables 4, 5), particularly after about 2050 when the global surface temperature response stabilizes (and declines thereafter).

Of particular interest are the low emissions scenarios, RCP 2.6/SSP 1-26, demonstrating the potential of strong mitigation policies and speculative negative emission technologies where anthropogenic forcing starts decreasing around the mid-2040s. In this scenario, the CMIP5 MME temperature stays below $2K$ throughout the 21st century, whereas the corresponding median FEBE temperature projection never exceeds $1.5K$. Comparing projected warming at 2100 for the RCP 2.6/SSP 1-26 scenario, the FEBE projection reaches a median warming of $1.2K$ with 90% CI of $[1.1, 1.4]K$ while the CMIP5 MME has a 90% CI of

$[0.9, 2.4]K$ and median warming of $1.7K$. When considering the CMIP6 projections for SSP 1-26 (figure 12, d) the median temperature exceeds $2K$ beginning near 2050, whereas the corresponding FEBE projection is consistently lower, only crossing the $1.5K$ threshold briefly. At 2100, the CMIP6 projected temperature reaches $2.2K$ with 90% CI of $[1.5, 2.8]K$ while the FEBE projects a median temperature of $1.5K$ and a narrower spread of $[1.3, 1.8]K$.

While the forcing of the (perhaps most realistic) middle scenario, RCP 4.5/SSP 2-45, stabilizes in the mid 2060s, the

temperature projections continue rising throughout the 21st century for both FEBE and the CMIP5/6 MME (figure 12b, e). At 2100, the FEBE and CMIP5 MME project the temperature reaching $1.9K$ $[1.6, 2.2]K$, and $2.6K$ $[1.8, 3.2]K$ respectively shown in figure 12b. A key point to note is that the FEBE RCP 4.5 projection remains below $2.5K$ of warming by 2100, while the CMIP5 MME is well beyond this threshold. Looking at the CMIP6 projections for SSP 2-45 (figure 12, e) the median temperature exceeds $2K$ beginning near 2050, whereas the corresponding FEBE projection is consistently lower, and begins

to diverge after 2050. At 2100, the CMIP6 projected temperature reaches $3K$ with 90% CI of $[12.1, 4.2]K$ while the FEBE projects a median temperature of $2.3K$ and a narrower spread of $[1.8, 2.8]K$.

The projections of both the FEBE and the CMIP5 MME for the strong emission scenario, RCP 8.5, show alarming warming rates of $3.5K$ with 90% CI $[2.9, 4.1]K$, and $4.8K$ with 90% CI $[3.5, 6.0]K$ in 2100 shown in figure 12c, f. The same quickly increasing trend is seen in the CMIP6 SSP 5-85 scenario with temperatures in 2100 reaching a staggering $6.2K$ with 90% CI

$[4.5, 7.0]K$, while the FEBE projection, although lower at $3.8K$ and having a tighter bound of $[3.5, 4.5]K$, shows the dire consequences of no mitigation. All results shown in figure 12 are summarized in tables 4 and 5.

Whereas the CMIP5 projections differ from the CMIP5 projections due to both model and forcing series changes, the FEBE projections differ only because of the changes in the forcing series. By comparing left and right columns of figure 12 we can help quantify the difference in projected warming caused by the changing of the forcing series and between CMIP model

generations. In the year 2100 we see the FEBE is $1.4K$ above pre-industrial in the RCP 2.6 scenario, whereas in the SSP 1-26 scenario it is $1.5K$ above (a difference of $0.1K$). In comparison, for the same two scenarios, the CMIP5 MME is $1.6K$ and

the CMIP6 MME is $2.0K$ above pre-industrial respectively (a difference of $0.4K$). The same can be done for the two other scenarios: for the RCP 4.5/SSP 2-45 scenarios the FEBE is found to have warming at 2100 of $2.1K$ and $2.3K$ (a difference of $0.2K$) above pre-industrial, while the CMIP5/6 MMEs having warming of $2.6K$ and $3.0K$ (a difference of $0.4K$); in the RCP 8.5/SSP 5-85 scenarios, the FEBE projects warming of $3.6K$ and $4.0K$ (a difference of $0.4K$) with the CMIP5/6 MMEs projecting temperatures of $4.8K$ and $5.4K$ (a difference of $0.6K$) above pre-industrial. This analysis confirms that both the CMIP6 forcings and models are warmer. Therefore we can attribute the difference in warming that comes from the changing of the forcings and the changing of the model generation using the FEBE; the same attribution could be done if the CMIP6 models were rerun using the older RCP scenarios, or vice versa.

Although the FEBE projections are consistently about 15% cooler than the CMIP5 MME, due to the its smaller uncertainty the FEBE 90% CI lies entirely within the corresponding CMIP5 CI. Both projection methods support each other and are thus complementary. When compared to CMIP6 projections although most of 90% CIs overlap, the median CMIP6 temperatures are nearly 65% warmer than the corresponding FEBE median, mainly caused by their overpowered aerosols (Zelinka et al., 2020; Flynn and Mauritsen, 2020) and previously mentioned discrepancy in the future aerosol removal as compared to the RCPs.

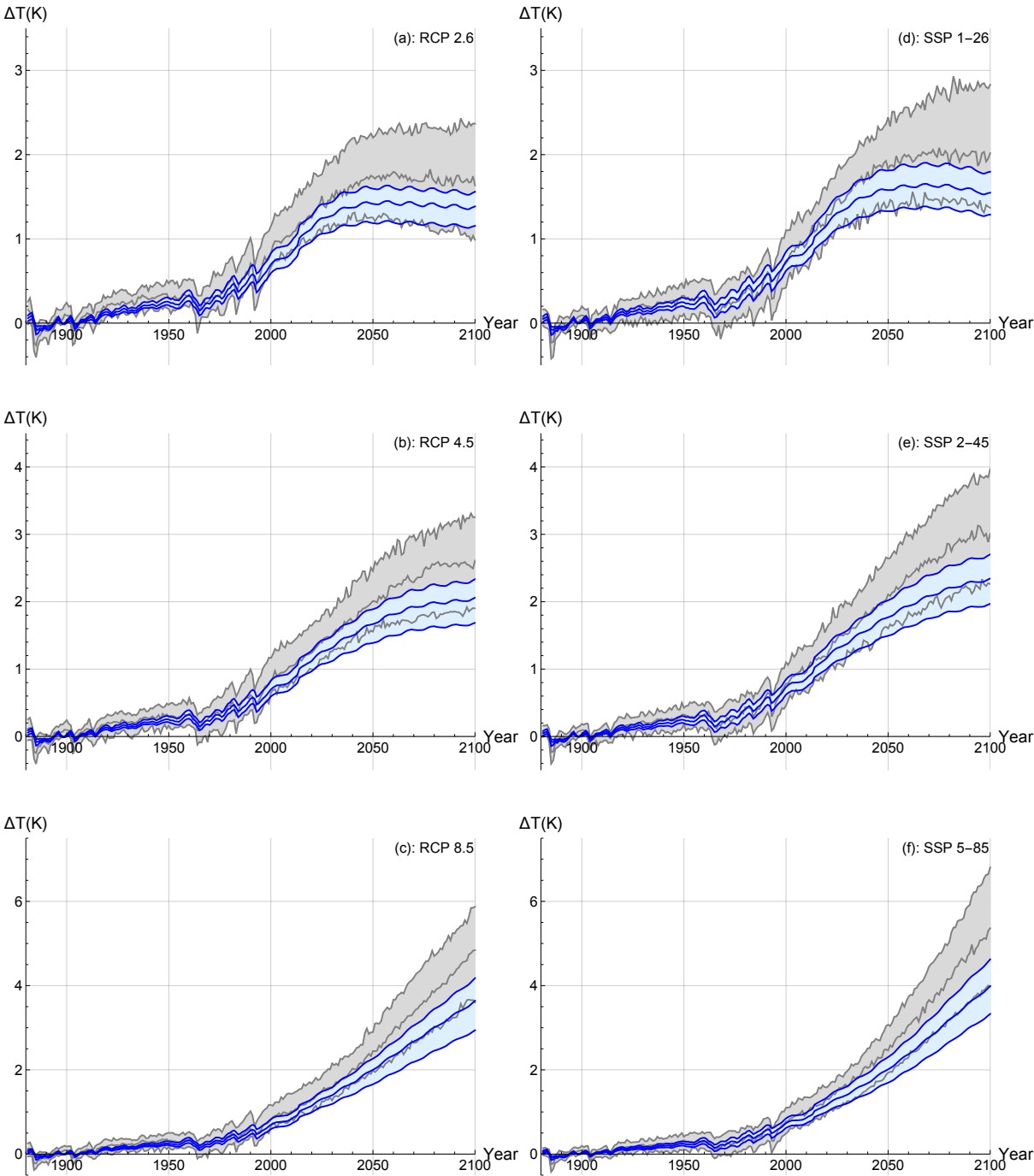

**Figure 12.** The deterministic forced temperature response projected using the FEBE (blue), with parameters calibrated using $F_{Aer_{RCP}}$ (a, b, c) and $F_{Aer_{SSP}}$ (d, e, f) compared with the CMIP5/6 MME projection (gray); 90% CI from the parametric uncertainty are indicated (shaded). The projections until 2100, for RCP 2.6/SSP 1-26 (top), RCP 4.5/SSP 2-45 (middle) and RCP 8.5/SSP 5-85 (bottom), are shown.

**Table 4.** The 90% CI of projected warming relative to pre-industrial reference period (1880 –1910) for the RCP scenarios analysed in this study based on the FEBE and the CMIP5 MME. Summary of figure 12 (a, b, c).

|  | RCP 2.6 | | RCP 4.5 | | RCP 8.5 | |
|  | **FEBE** | **CMIP5** | **FEBE** | **CMIP5** | **FEBE** | **CMIP5** |
|---|---|---|---|---|---|---|
| **2020-2040** | [1.1, 1.5]K | [1.2, 1.9]K | [1.1, 1.5]K | [1.3, 1.9]K | [1.2, 1.6]K | [1.4, 2.0]K |
| **2040-2060** | [1.2, 1.6]K | [1.3, 2.2]K | [1.4, 1.9]K | [1.6, 2.6]K | [1.7, 2.3]K | [2.0, 3.0]K |
| **2060-2080** | [1.2, 1.6]K | [1.2, 2.3]K | [1.6, 2.2]K | [1.8, 3.0]K | [2.2, 3.1]K | [2.6, 4.3]K |
| **2080-2100** | [1.2, 1.6]K | [1.1, 2.4]K | [1.6, 2.3]K | [1.8, 3.2]K | [2.7, 3.8]K | [3.3, 5.3]K |

**Table 5.** The 90% CI of projected warming relative to pre-industrial reference period (1880 –1910) for the SSP scenarios analysed in this study based on the FEBE and the CMIP6 MME. Summary of figure 12 (d, e, f).

|  | SSP 1-26 | | SSP 2-45 | | SSP 5-85 | |
|  | **FEBE** | **CMIP6** | **FEBE** | **CMIP6** | **FEBE** | **CMIP6** |
|---|---|---|---|---|---|---|
| **2020-2040** | [1.2, 1.6]K | [1.2, 1.9]K | [1.2, 1.6]K | [1.2, 2.0]K | [1.2, 1.7]K | [1.2, 2.0]K |
| **2040-2060** | [1.3, 1.8]K | [1.4, 2.3]K | [1.5, 2.0]K | [1.6, 2.7]K | [1.7, 2.3]K | [1.9, 3.0]K |
| **2060-2080** | [1.4, 1.9]K | [1.5, 2.6]K | [1.8, 2.4]K | [1.9, 3.2]K | [2.3, 3.2]K | [2.8, 4.4]K |
| **2080-2100** | [1.3, 1.8]K | [1.4, 2.8]K | [1.9, 2.6]K | [2.2, 3.8]K | [3.0, 4.2]K | [3.6, 6.0]K |

## 5.2 Probabilities of Exceeding Critical Warming Thresholds

We can also use the FEBE to estimate the probability of exceeding various warming thresholds. Important tipping points have been established which could lead to irreversible changes in major ecosystems and the planetary climate if certain threshold in warming are exceeded (Schurer et al., 2017; Smith et al., 2018b; Iseri et al., 2018) . Using the FEBE and CMIP5/6 MME we calculate the probability of temperature exceeding $1.5K$ and $2.0K$ (figure 13).

According to the FEBE for the low emission scenario RCP 2.6, it is unlikely to exceed the $1.5K$ threshold in 2100 ($< 10\%$) while it is much more likely to exceed this threshold according to CMIP5 MME ($67\%$). The FEBE has a negligible probability of exceeding $2K$ while the CMIP5 MME has a $26\%$ probability. While in the SSP 1-26 scenario, the FEBE peaks at below $50\%$ probability of exceeding $1.5K$ and has a negligible probability of exceeding $2K$ as before; in comparison according to the CMIP6 MME it is nearly certain to cross $1.5K$ threshold, while the probability of the $2K$ threshold being exceeded hovers around $60\%$ even under strong mitigation.

In the RCP 4.5 scenario, the probability of the FEBE exceeding the $1.5K$ threshold is extremely likely ($> 95\%$) although it occurs in 2070 - about 22 years later than that projected by the CMIP5 MME. For the SSP 2-45 scenario, we see the FEBE trails the CMIP6 MME until around 2035, after which exceeding $1.5K$ becomes very likely for both (near 2045). Similarly, the $2K$ overshoot, as projected by the FEBE will be avoided with a probability of $< 40\%$ but will most likely not be avoided

according to the CMIP5 MME (89% probability). Again we see the FEBE lags behind the CMIP6 MME, before they begin to converge around 2080, approaching a very likely probability to exceed the threshold.

For the final high emission, business as usual, RCP 8.5 and SSP 5-85 scenarios; both the FEBE and CMIP5/6 MME project that exceeding the $1.5K$ threshold is virtually inevitable by 2100. Although in the FEBE projection it is extremely likely that this threshold is exceeded nearly 15 years after the CMIP5/6 MME projections of 2040. The same is found for the $2K$ threshold, with both the FEBE and CMIP5/6 MME exceeding the threshold about 15 years after the $1.5K$ threshold. These results are all summarized in table 6.

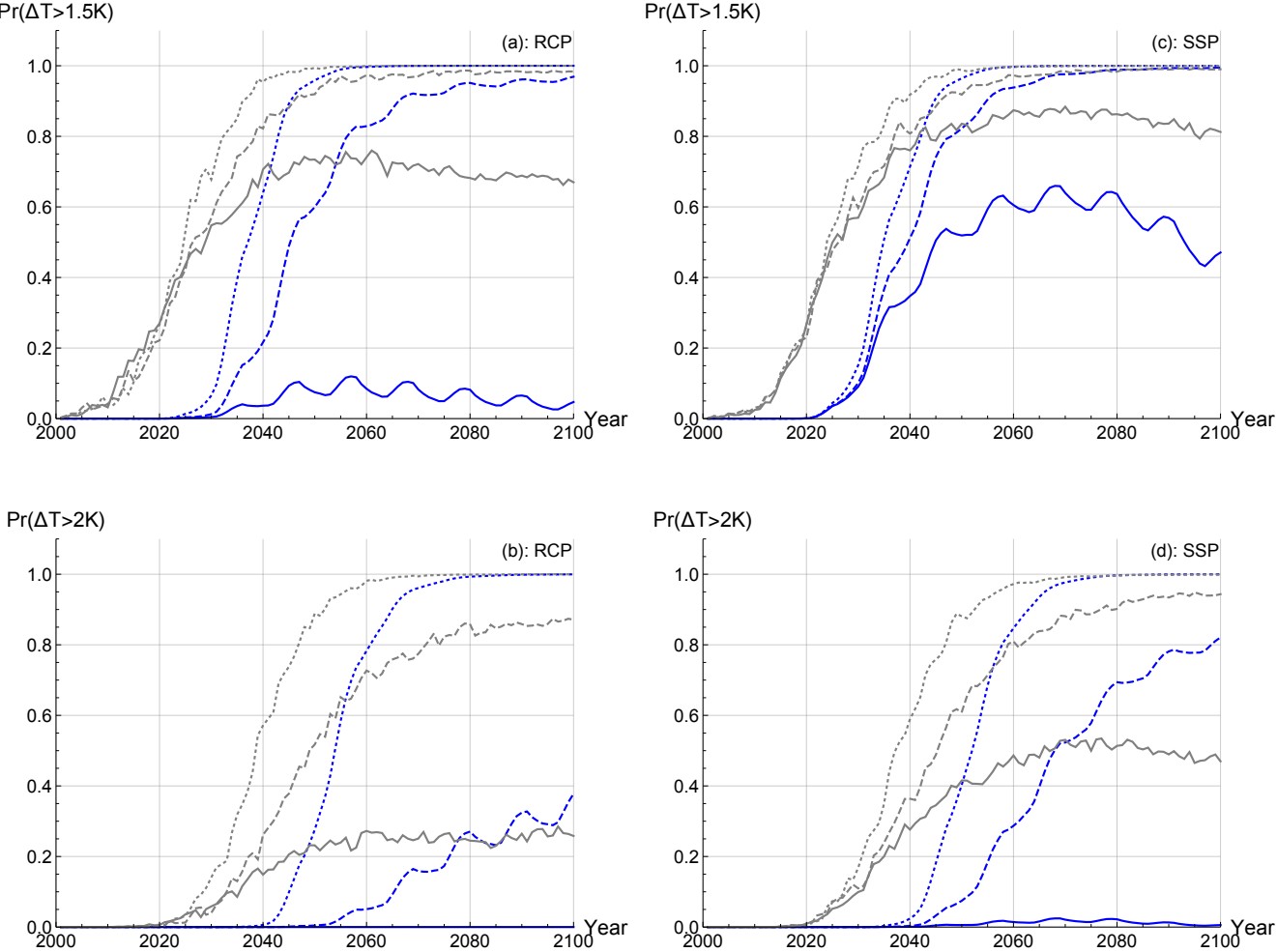

**Figure 13.** The probability for the global mean surface temperature of exceeding a 1.5$K$ threshold (top: a,c), and a 2$K$ (bottom - b,c) are given as a function of years for the FEBE (blue), using $F_{Aer_{RCP}}$ (a,b) or $F_{Aer_{SSP}}$ (c, d) and for the CMIP5/6 MME (gray). The three scenarios are considered for each case: RCP 2.6/SSP 1-26 (solid), RCP 4.5/SSP 2-45 (dashed), and RCP 8.5/SSP 5-85 (circles).

**Table 6.** List of RCP and SSP scenarios analysed in this study and the probabilities of exceeding $1.5°C$ or $2°C$ based on the FEBE and the CMIP5/6 MME. Summary of figure 13.

| | Probability of Exceeding $1.5°C$ at 2100 | | | Probability of Exceeding $2°C$ at 2100 | | |
|---|---|---|---|---|---|---|
| | **FEBE** | **CMIP5** | **CMIP6** | **FEBE** | **CMIP5** | **CMIP6** |
| **RCP 2.6** | 0.1% | 67.0% | - | 0.0% | 25.8% | - |
| **RCP 4.5** | 96.9% | 98.4% | - | 37.8% | 87.1% | - |
| **RCP 8.5** | 100% | 100% | - | 100% | 100% | - |
| **SSP 1-26** | 47.1% | - | 81.2% | 0.0% | - | 47.0% |
| **SSP 2-45** | 99.5% | - | 99.0% | 82.1% | - | 94.3% |
| **SSP 5-85** | 100% | - | 100% | 100% | - | 99.9% |

## 6 Summary

In the following section we summarize the key results presented earlier in the paper: model and forcing parameters (see table 1), the climate sensitivities (see table 2), the projected warming at 2100 (see tables 4 and 5) and the probabilities of exceeding warming thresholds of $1.5K$ and $2.5K$ (see table 6).

### 6.1 Parameter Estimates

The two parameters that characterize the model, $h$ and $\tau$, were estimated. The fundamental scaling exponent $h$, was found to have a median value of 0.38 and 90% CI of $[0.33, 0.44]$ using $F_{Aer_{RCP}}$, and similar median value 0.38 and 90% CI of $[0.32, 0.44]$ for $F_{Aer_{SSP}}$. Both estimates are near $h$ estimated for the Scaling Climate Response Function (Hébert et al., 2021) and the phenomenological HEBE (Lovejoy, 2021a, b). The relaxation time scale $\tau$, characterizing the approach to equilibrium was found to have median value of 4.7 years and 90% CI of $[2.4, 7.0]$ years when using $F_{Aer_{RCP}}$, and nearly identical results using $F_{Aer_{SSP}}$. The estimated relaxation time is comparable to other box model fast relaxation times (Schwartz, 2008; Held et al., 2010; Geoffroy et al., 2013; Rypdal and Rypdal, 2014) as the one box (EBE) model is a special case of the FEBE with $h = 1$.

The FEBE model also adjusts the deterministic forcings, notably the aerosol and volcanic forcing series which must be scaled (the former linearly and the latter non-linearly) for the temperature response to best match historical temperature records. From our analysis we find a more constrained aerosol forcing. For the $F_{Aer_{RCP}}$ we found a median recalibration factor $\alpha$ of 0.6 with 90% CI $[0.2, 1.0]$. Following Stevens (2015) this supports a revision of the global modern (2005) aerosol forcing 90% CI to a narrower range $[-1.0, -0.2]Wm^{-2}$. Using the CMIP6 aerosols, $F_{Aer_{SSP}}$, we found a median $\alpha$ value of 0.33 and 90% $[0.05, 0.61]$ implying a weaker and more tightly constrained modern (2005) aerosol forcing of $[-0.9, -0.1]Wm^{-2}$. For volcanism, the

non-linear intermittency exponent, $\nu$, was found have median value of 0.28 with 90% CI of [0.15, 0.41] using $F_{Aer_{RCP}}$ and a median 0.28 with similar 90% CI of [0.16, 0.40] using $F_{Aer_{SSP}}$.

In comparison to IPCC AR5 and to the CMIP6 MME, we find lower likely ranges for the climate sensitivity parameter, ECS and TCR when using the FEBE with $F_{Aer_{RCP}}$ (or $F_{Aer_{SSP}}$). For projections, perhaps the most important parameter is $s$, the climate sensitivity parameter that determines the temperature response following an increase in forcing. We find $s$ to have a median value of $0.56\ K(Wm^{-2})^{-1}$ with 90% CI $[0.45, 0.67]\ K(Wm^{-2})^{-1}$ using $F_{Aer_{RCP}}$, and when using $F_{Aer_{SSP}}$ we find median $0.52 K(Wm^{-2})^{-1}$ with 90% CI $[0.43, 0.61]\ K(Wm^{-2})^{-1}$. Again see a lower median for the ECS in comparison to the IPCC AR5 (and CMIP6 MME) estimates for their corresponding forcings, the 90% CI range is reduced from $[1.5, 4.5]K$ $([2.0, 5.5]K)$ to $[1.6, 2.4]K$ $([1.5, 2.2]K)$ and the median value is lowered from $3.0K$ $(3.7K)$ to $2.0K$ $(1.8K)$. Several recent observation-based studies (Otto et al., 2013; Skeie et al., 2014; Johansson et al., 2015; Lewis and Curry, 2015, 2018) have also reported lower ECS upper bounds. We also estimated the derived quantity, the Transient Climate Response (TCR), the temperature increase following a linear doubling in forcing over 70 years. For the TCR, where the 90% CI range shrinks from $[1.0, 2.5]K$ $([1.2, 2.8]K)$ to $[1.2, 1.8]K$ $([1.1, 1.6]K)$ and the median estimate decreases from $1.8K$ $(2.0K)$ to $1.5K$ $(1.4K)$.

## 6.2 Hindcasts

With all necessary parameters of the FEBE calibrated on observational temperature series we evaluated the FEBE reliability showing that it is able to reconstruct the historical temperatures (Sect. 4.1), that it can reproduce the response amplitude to a internal white noise forcing (Sect. 4.2), and produces realistic temperature fluctuations over a wide range of scales (Sect. 4.3). Having shown that the FEBE reproduces historical temperatures and their statistics, we then produced deterministic temperature projections to 2100 using the RCPs: 2.6, 4.5, 8.5 and SSPs: 1-26, 2-45, 5-85 comparing them to their respective CMIP5 and CMIP6 MMEs relative to the pre-industrial baseline of 1880–1910.

## 6.3 Projections

In the low emission scenario RCP 2.6 (SSP 1-26) the FEBE projects the 90% CI of the temperature at 2100 to be $[1.2, 1.6]$K $([1.3, 1.8]$K) as compared to the CMIP5 (CMIP6) MME of $[1.1, 2.4]$K $([1.4, 2.8]$K). The middle scenario, RCP 4.5 (SSP 2-45) the FEBE projects warming reaching $[1.6, 2.3]$K $([1.9, 2.6]$K), narrower than the CMIP5 (CMIP6) MME warming of $[1.8, 3.2]$K $([2.2, 3.8]$K). While in the high emission scenario, RCP 8.5 (SSP 5-85) both the FEBE and CMIP5 (CMIP6) MME project extreme temperature increases of $[2.7, 3.6]$K $([3.0, 4.2]$K) and $[3.3, 5.3]$K $([3.6, 6.0]$K), highlighting the need for strong emission mitigation.

During the Paris Conference in 2015, (COP21) nations of the world strengthened the United Nations Framework Convention on Climate Change by agreeing to holding the increase in the global average temperature to well below $2°C$ above pre-industrial levels and pursuing efforts to limit the temperature increase to $1.5°C$. According to our projections, crossing either of these thresholds is delayed with respect to the CMIP5/6 MME projections but will eventually happen if strong mitigation is not implemented. To avert a $1.5K$ warming, drastic cuts would have to be made to global greenhouse emissions, similar to that in RCP 2.6 (and SSP 1-26), for which we found <10% (<50%) probability of exceeding $1.5K$ in comparison to the CMIP5

(CMIP6) MME which projects a 67% (>80%). Both the FEBE and CMIP5/6 projections have temperatures surpassing $1.5K$ in scenarios with weak or no mitigation: RCP 4.5/SSP 2-45, and RCP 8.5/SSP 5-85, albeit the FEBE projects this occurring nearly two decades later than the GCMs. The $2K$ threshold is projected to be avoided by both the FEBE and CMIP5/6 MME if we follow low emission scenarios of RCP 2.6 and SSP 1-26. The opposite is true for any other emission scenarios; exceeding the $2K$ threshold will almost occur before 2100. Thus our model reinforces the conclusion that only strong mitigation scenarios

such as RCP 2.6 and SSP 1-26, will avoid exceeding the $1.5K$ and $2K$ thresholds. It remains to be seen whether negative emission technologies are feasible and whether the appropriate policies are implemented.

## 7    Conclusions

Ever since the first climate models at the end of the 1970's, multidecadal projections have had large uncertainties with the wide ECS uncertainty limits of $1.5$-$4.5K$ essentially unchanged. For policy makers, the most deleterious consequence of large

uncertainties is that projections emanating from quite diverse future scenarios have significant overlap. For example, up until 2050, the RCP 2.6 and 8.5 scenarios can both claim to respect the $2K$ threshold - albeit with rather different probabilities (figure 13). Large overlaps imply a disconnection between policies (mitigation scenarios) and outcomes (temperatures). Now that governments have committed themselves to keeping industrial epoch temperature increases to below $2K$ (and aim at $1.5K$), we face an uncertainty crisis (Lovejoy, 2019b).

One way of reducing this uncertainty is by developing complementary types of models. In this paper we directly constructed such a model in the macroweather regime (roughly one month and up) based on the physically principles of energy conservation and scaling: the Fractional Energy Balance Equation (FEBE). Although originally derived phenomenologically, it was recently discovered (Lovejoy, 2021a, b) that the FEBE could be derived as a consequence of classical (Budyko, 1969; Sellers, 1969), Energy Balance Models (EBMs) that have been regularly used to determine the Earth's latitudinal temperature variations,

its stability to perturbations and to study past and future climate states. The key was to introduce a vertical coordinate that allows for the application of the correct conductive-radiative surface boundary condition, needed for correctly determining the energy storage. A surprising consequence is that even the classical (integer ordered) continuum mechanics heat equation used by Budyko and Sellers implies that the surface temperature obeys a fractional ordered energy balance equation. The FEBE's fractional storage terms imply that the system has a long memory so that when calibrated by observational data, its responses

to past forcings are constrained to respect the historical climate.

The FEBE is a parsimonious model with only two shape parameters: an exponent $h$, and relaxation time scale $\tau$; the classical EBE (box model) is the $h = 1$ special case. In order to make FEBE projections, we use a Bayesian parameter estimation approach similar to that used in Hébert et al. (2021). A Climate Response Function was used by Hébert et al. (2021) that at long times ($> \tau$) was close to the corresponding low frequency part of the FEBE Green's function, but that rather than

720 being truncated at high frequencies was a different power law. While the Hébert et al. (2021) CRF was justified only on the basis of scale invariance and linearity, the FEBE has a stronger physical basis since it respects both scaling as well as energy conservation, and the long memory is explicitly situated in energy storage mechanisms. Beyong its stronger theoretical basis,

from the practical (projection) point of view, the main advantage is that the FEBE directly handles the short time scales (down to a month or less). This allows the FEBE to directly take into account the internal variability: a stochastic white noise forcing and the FEBE response. The ability to model the forced response to both external and internal forcing improves the FEBE parameter estimates and contributes to lowering the corresponding projection uncertainties. It also allowed us to demonstrate the projections were reliable (in the technical sense).

Bayesian inference allows for a robust probabilistic parameter characterization. The basic external forcings were those prescribed for the historical part of the CMIP5/6 GCMs and these were constrained by five monthly, global resolution empirical temperature series (since 1880). The internal forcing was assumed to be a Gaussian white noise and, since to a good approximation, the FEBE white noise response is a fractional Gaussian noise (fGn), the latter was taken as the Bayesian inference error model.

In order to estimate the parameters, the forcing series required two adjustments. The most important was the aerosol recalibration parameter $\alpha$ which linearly scales the aerosol forcing to take into account the increasing evidence that the CMIP5 and CMIP6 aerosol cooling was too strong (Padilla et al., 2011; Hébert et al., 2021; Zelinka et al., 2020; Tokarska et al., 2020; Flynn and Mauritsen, 2020). The former aerosol series ($F_{Aer_{RCP}}$) was based both on uncertain data but also on uncertain modelling assumptions, especially about the direct and indirect effects of aerosols. Whereas the latter ($F_{Aer_{SSP}}$) is based on global sulphate production and derived from an alternative model than that in CMIP5.

The forcings and parameters combined with the RCP and SSP scenarios allow us to make projections through to 2100, we did this for RCP 2.6 (SSP 1-26), 4.5 (SSP 2-45), and 8.5 (SSP 5-85). Overall, the observational based FEBE projections had uncertainties that are smaller by more than a factor of two in comparison to the CMIP5/6 MME uncertainties. However, the two modelling approaches have quite different sources of uncertainty. Whereas the CMIP5/6 uncertainty is purely due to differences in the climates of the GCMs ("structural uncertainties"), the FEBE uncertainty is "parametric" and it depends largely on the uncertainty of the historical forcings and temperatures, in particular those associated with aerosols. In fact, a byproduct of the model and Bayesian framework is that we are able to more tightly constrain aerosol forcing, supporting recent literature findings of weaker historical aerosol cooling. As a consequence, the FEBE projections are consistently a little cooler than those of the CMIP5/6 MME but with uncertainties about half of those of the MME, it still lies within the MME uncertainty bounds. By comparing the FEBE with the CMIP5 and CMIP6 MME's we were also able to separately quantify the contribution of changing the RCP to SSP forcing scenarios from that of the difference in the models. The qualitatively different FEBE thus effectively complements the GCMs.

There is a long history - starting with the 4/3 law of turbulent diffusion (Richardson, 1926) – of attempts to directly stochastically model the collective behaviour of huge numbers of interacting structures, processes. Just as thermodynamics and continuum mechanics are themselves high level laws with respect to statistical mechanics, stochastic turbulence laws are "high level" with respect to the usual deterministic laws of fluid mechanics. In both cases, the idea is to ignore most of the "details" that turn out to be irrelevant and model only the relevant ones. The FEBE, based on the basic scale and energy symmetries, is a specific contribution to this tradition. It comes at a moment when there is growing interest in stochastic and other alternative approaches to climate modelling and projections (see e.g. Irrgang et al., 2021).

In addition, whereas the regional (horizontally varying) FEBE has already been derived from the heat equation (Lovejoy, 2021a, b), here it is the greatly simplified "zero-dimensional" (globally averaged) model that is considered (for some early regional FEBE results (see e.g. Del Rio Amador and Lovejoy, 2021b; Procyk, 2021). In addition, for simplicity we restricted the FEBE to the linear case, but it is quite easy to include nonlinear model feedbacks - for example temperature - albedo feedbacks (for glacial-interglacial modelling) or feedbacks such as those from temperature-permafrost emissions relevant to potential tipping points.

In future, time varying parameters may also be considered: for example the climate sensitivity multiplied by the forcing constitutes an "effective forcing" so that time varying sensitivities are trivial to include – they essentially just change the forcing. Less trivial are the inclusion of time varying relaxation times, and at the moment it is not obvious that it is possible to even mathematically define a time varying order of temporal differentiation (i.e. a time varying $h$). However, it is possible to use spatially varying $h$ exponents for regional modelling and this is apparently necessary for regional applications (work in progress and (see Del Rio Amador and Lovejoy, 2021a). Other possible directions for generalizing the FEBE include coupling the temperature field with other fields such as precipitation, ice cover, land use and carbon cycle. Finally, it could also be mentioned that various FEBE foundational issues need to be resolved – such as the relationship between top of the atmosphere and surface radiative fluxes - and there also nontrivial practical issues for estimating regional FEBE parameters.

The FEBE model and projections to 2100 which is an observational-based model physically based on energy and scale symmetries, complementary to GCMs. Future work will explore the full (regional, 2D) FEBE model (Lovejoy, 2021b), which hopefully will constrain and improve future projections. In (Lovejoy et al., 2021; Lovejoy, 2021a), the FEBE is shown to plausibly reproduce the annual cycle at monthly resolution, in particular to explain the lag between the temperature maximum and the maximum in the radiative forcing. We can also calibrate the FEBE on the historical runs of the CMIP models in order to perform a feedback analysis to investigate the differences between how models treat their volcanic and aerosol forcings through the parameters $\nu$ and $\alpha$. Updating our parameter estimates from calibrations on GCMs allows for GCM-FEBE hybrid projections. Extensions to precipitation may also be possible at global and regional scales since the FEBE model is consistent with space-time scaling processes in historical precipitation data (de Lima and Lovejoy, 2015). With tighter constraints on ECS and TCR from the FEBE we can better estimate future warming when bringing together multiple lines of evidence such as that done in Sherwood et al. (2020). The FEBE once expanded spatially provides a flexible framework which can be calibrated directly on observations, providing a direct representation of forcing to response relationships.

*Data availability.* RCP concentrations can be found at https://tntcat.iiasa.ac.at/RcpDb/dsd?Action=htmlpage&page=welcome. SSP radiative forcings are provided at https://doi.org/10.5281/zenodo.3515339. CMIP5/6 model outputs are available at https://esgf-node.llnl.gov.

## Appendix A

**Table A1.** List of CMIP6 Models and model climate parameters.

| Model | ECS ($K$) | TCR ($K$) | TCR-to-ECS ratio |
|---|---|---|---|
| MIROC6 | 2.60 | 1.58 | 0.61 |
| IPSL-CM6A-LR | 4.50 | 2.39 | 0.53 |
| CNRM-CM6-1 | 4.82 | 2.23 | 0.46 |
| BCC-CSM2-MR | 3.07 | 1.60 | 0.52 |
| MRI-ESM2 | 3.11 | 1.67 | 0.54 |
| CanESM5 | 5.58 | 2.75 | 0.49 |
| CESM2 | 5.15 | 1.99 | 0.39 |
| GISS-E2-1-H | 2.99 | 1.81 | 0.61 |
| GISS-E2-1-G | 2.60 | 1.66 | 0.64 |
| SAM0-UNICON | 3.30 | 2.08 | 0.63 |
| E3SM-1-0 | 5.09 | 2.91 | 0.57 |
| UKESM1-0-LL | 5.31 | 2.79 | 0.53 |
| CNRM-ESM2-1 | 4.75 | 1.82 | 0.38 |
| BCC-ESM1 | 3.29 | 1.77 | 0.54 |
| CESM2-WACCM | 4.65 | 1.92 | 0.41 |
| MIROC-ES2L | 2.66 | 1.51 | 0.57 |
| EC-EARTH3-VEG | 3.93 | 2.76 | 0.70 |
| HADGEM3-GC31-LL | 5.46 | 2.47 | 0.45 |
| NORCPM-1 | 2.78 | 1.55 | 0.56 |
| GFDL-CM4 | 3.79 | - | - |
| GFDL-ESM4 | 2.56 | - | - |
| NESM3 | 4.50 | - | - |
| NORESM2-LM | 2.49 | 1.48 | 0.59 |
| NORESM2-LM | 2.49 | 1.48 | 0.59 |
| MPI-ESM1-2-HR | 2.84 | 1.57 | 0.55 |
| INM-CM4-8 | 1.81 | 1.30 | 0.72 |
| Ensemble Mean ± Std: | 3.74±1.11 | 1.98±0.48 | 0.55 ± 0.09 |

**Table A2.** List of CMIP5 Models and climate sensitivity parameters.

| Model | ECS ($K$) | TCR ($K$) | TCR-to-ECS ratio |
|---|---|---|---|
| MPI-ESM-LR | 3.48 | 1.94 | 0.56 |
| MPI-ESM-MR | 3.31 | 1.93 | 0.58 |
| MPI-ESM-P | 3.31 | 1.96 | 0.59 |
| MIROC5 | 2.70 | 1.49 | 0.55 |
| MIROC-ESM | 4.68 | 2.15 | 0.46 |
| IPSL-CM5B-LR | 2.58 | 1.44 | 0.56 |
| IPSL-CM5A-MR | 4.03 | 1.96 | 0.49 |
| IPSL-CM5A-LR | 3.97 | 1.94 | 0.49 |
| ISM-CM4 | 2.01 | 1.22 | 0.61 |
| CSIRO-Mk3.6.0 | 4.05 | 1.76 | 0.43 |
| CNRM-CM5 | 3.21 | 2.04 | 0.64 |
| CNRM-CM5-2 | 3.40 | 1.63 | 0.48 |
| BNU | 3.98 | 2.58 | 0.65 |
| BCC-CSM1.1 | 2.81 | 1.74 | 0.62 |
| BCC-CSM1.1(m) | 2.77 | 2.00 | 0.72 |
| BCC-GCCM3 | 2.65 | 1.58 | 0.60 |
| NORESM1-M | 2.75 | 1.34 | 0.49 |
| ACCESS1.0 | 3.76 | 1.72 | 0.46 |
| CanESM2 | 3.71 | 2.37 | 0.64 |
| GFDL-ESM2M | 2.33 | 1.23 | 0.53 |
| GFDL-CM3 | 3.85 | 1.85 | 0.48 |
| CCSM4 | 2.90 | 1.64 | 0.57 |
| FGOALS-g2 | 3.39 | 1.42 | 0.41 |
| GISS-E2-H | 2.33 | 1.69 | 0.73 |
| GISS-E2-R | 2.06 | 1.41 | 0.68 |
| HADGEM2-ES | 3.96 | 2.38 | 0.60 |
| Esemble Mean ± Std: | 3.20±0.70 | 1.75±0.38 | 0.56 ± 0.09 |

*Author contributions.* SL: conceptualization of the study. RH: design of methods for model calibration. RP: development the model code and prepared the manuscript with contributions from all co-authors.

*Competing interests.* The authors declare that they have no conflict of interest.

*Acknowledgements.* S. Lovejoy acknowledges some support from the National Science and Engineering research Council (Canada). R. Hébert has received funding from the European Research Council (ERC) under the European Union's Horizon 2020 research and innovation programme (grant agreement no. 716092 and grant agreement no. 772852. We thank D. Clarke and L. Del Rio Amador for helpful discussions, and M. Willard-Stepan for help in editing the manuscript. The work profited from discussions at the CVAS working group of the Past 795     Global Changes (PAGES) programme.

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
