# Peer review of "The Fractional Energy Balance Equation for Climate projections through 2100"

_Earth System Dynamics, 2020_

## Referee Comment (RC1) · Anonymous Referee #1 · 11 Sep 2020

This manuscript is a very interesting application of a recent model proposed by some of the authors to the problem of climate projections of global temperature. They are using the fractional energy balance equation (FEBE) with realistic forcing of greenhouse gases, volcanic eruptions and solar variability. Their projections are compared with CMIP5/6 projections. The set of parameters of the model is fitted to the observation data in a Bayesian framework. This work is worth publishing in Earth System Dynamics. I have however a few points that I think should be appropriately addressed by the authors:

Figures 9 to 11 show the current and future evolutions of the global temperature as compared with the observations and the CMIP5 simulations. These projections are provided as ensembles in order to cover the range of uncertainties related to the model

settings. An observational uncertainty is also provided. It is obvious that the spread of the ensemble is much smaller for the FEBE model than for the CMIP5 ensemble in Figure 9, and very often the observation band is not falling into the range of predicted projections of FEBE, while it is the case for the CMIP5. This suggests that the projections are unreliable in the sense that it does not cover all possible situations that can be observed. In weather and climate sciences, this aspect is key when making forecasts, predictions and projections using ensembles. This unreliability should be first acknowledged in the manuscript. Second this unreliability is maybe related to the use of a weak stochastic forcing. Some experiments with larger stochastic forcing would be desirable in order to clarify the under-dispersion of the ensemble. Stronger stochastic forcing could also maybe lead to larger climate sensitivity. This should be checked too.

At page 20, line 421, the authors claim that the hiatus is better represented in the FEBE than in the CMIP5 ensemble. Well to me this is not true as the observations are most of the time out of the range of FEBE. The CMIP5 looks better at capturing the observations. So I suggest to modify these comments and try to be more objective in the comparison, maybe by using measures of reliability.

One key conclusion of the manuscript is the lower sensitivity of the FEBE model on long time scales as shown for instance in Figure 10. What is happening when the FEBE model is fitted with a stronger noise that could maybe help in increasing the spread of the ensemble and its reliability? The results and the conclusions should then be revisited once these experiments are done and the impact of the stochastic noise clarified.

Minor points:

Equation 14: Please clarify what is the variance of gamma(t)

Figure 6: The response of FEBE is compared with an IPCC two-box model. What is this model? Maybe I missed the place where it is described. Please describe this model in more details in the text

[Figure]

Figure 12: Please correct the colors of the curves. I cannot figure out what is plotted

---

## Referee Comment (RC2) · Anonymous Referee #2 · 19 Jan 2021

**Review of: The Fractional Energy Balance Equation for Climate projections through 2100**

**1 Summary of the paper**

The paper presents climate projections with a zero-dimensional energy balance equation and compares the resulting time series, equilibrium climate sensitivity, and transient climate response to CMIP5 and CMIP6 projections. The model parameters for the projections are determined using a Bayesian approach and historical observations. The projections are lower than the CMIP5/6 ensemble means, but mostly within their confidence intervals.

**2 Major comments**

With its focus on how a simple modeling approach can reproduce features from simulations by GCMs, the manuscript is tackling an interesting topic. However, in its present form major revisions are required to bring this manuscript into a form that is suitable for publication.

In particular:

- An assessment of scientific soundness is difficult at this point, in particular since information necessary for the reproduction of the results is not provided. There are errors in some of the figures (listed in detail below) and their quality as well as that of the general sentence structure could be improved for easier comprehension by the reader.

- There are a lot of gaps in the discussion of the model and results. Limitations of FEBE are not described at all, nor whether FEBE can be used for studies other than those concerning modern and future global mean temperature. The potential for future studies or extensions are also skimmed over.

- The reader does not get an idea of what the paper adds to Hébert et al. (2020) aside from replacing the truncated power law in the model equation. Especially, with respect to the results a discussion of differences is needed.

- The research is not sufficiently embedded in other literature on the topic of energy balance modeling in both the introduction and discussion.

**3 Minor comments**

**Abstract**

- L12 "We found that aerosols..." should mention that here aerosols refers to the CMIP aerosol forcings

- L22 It is unclear how FEBE complements (in addition to support) GCMs based on what is said.

- The abstract summarized well what is done in the paper, but fails to address limitations and the gap in literature it aims to fill.

**Introduction**

- L35 reference for the projection uncertainty in response to $CO_2$ doubling missing

- L43 should be motivated why it is desirable to construct models beyond linearity and stochasticity

- L69-72 These explanations require improvement:

  a) The EBE has not been explained enough for anyone who is not familiar with the model to fully comprehend this sentence. The reader has not been made aware that a storage term exists yet.

  b) Maybe include a short discussion of other for the storage term besides the one chosen.

- L118 Please mention why causality is not respected otherwise.

- L121 Please explain why the step response function is "more physical compared to the impulse response". It is not necessarily clear to the reader which response function will be used in which studies. This could be further elaborated in the text to clarify that issue.

- L130 a short explanation of what fractional Relaxation noise is would be helpful

**Methods and Material**

- The implementation of the model is not described at all.

- It should be mentioned explicitly that the model produces annual data (same for the temporal resolution of the forcing).

- L135-138 Would help to start a new paragraph here to clarify that now the text refers back to equations 5 and 6 for the H=1 case and not to the ones discussed in the meantime. It would also be helpful to the reader if it was shown how the function from Hébert et al. with $H_f = -0.5$ is the same as eq. 6 with $H = 0.5$ and why this is the same as "that corresponding to the internal forcing".

- L145 It is unclear why 277ppm is the reference.

- L177 A short elaboration as to why a linear scaling factor is appropriate to describe the uncertainty would be nice.

- Section 2.2.3 For RCO scenarios it is stated that an aerosol forcing smaller than $-1\mathrm{Wm}^{-2}$ is implausible, but not why this is not also the case for the SSP forcing.

- Fig.1:

  a) lower plot has two sets of dashed data unlike the upper plot and unlike the plot description

  b) overlap between data and axis labels in lower plot could be prevented if the axis was at the bottom

  c) It is unclear why the forcing time series differ in length.

  d) The upper plot uses "Aer" and not "Ant" in the legend.

  e) Having the explanations for solid vs dashed vs dotted in the legend, too, would make the plots easier to understand.

  f) The plot sizes are different, would be nice if they were consistent.

- L202 It is unclear why $-27$ was chosen as the conversion factor. With regards to the literature this seems quite high and might also explain why the forcing was overpowered. AR5 recommends $-25$, Schmidt et al. (2011) $-20$ and for AR6 $-18$ is suggested. Might be interesting to look at what the effective AOD would have been that the "damped" volcanic forcing the authors ended up using corresponds to.

- L205 The point of comparison in "comparable effect" is missing.

- L208-210 An expansion of what it means that a quantification in a multi-fractal framework is possible would be helpful. The explanation of why it must be transformed remains unclear to me. (esp. relevant since a multifractality index is introduced in L218)

- L218 Would be helpful if the authors would expand on $C_1$ being the codimension of the mean and what the implications of this are.

- L222 and Eq. 13 This would serve as a good introduction to the whole description of the forcings instead of being hidden away at the end of the description of the volcanic forcing.

- L230-234 This sentence is quite complicated, especially with the choice of punctuation. Consider splitting it. Should be "addresses" instead of "address".

- Sec. 2.3: A plot of the time series would be helpful at this point (historic datasets + CMIP5/6 MME). The discussion is needlessly difficult to follow without it.

- L254 The internal stochastic forcing was not actually introduced in the formulas before so the "Recall that" seems misleading.

- L264 Please expand on why low frequencies only weakly influence the likelihood function.

- L288-291 Please expand on how this five-dimensional parameter space was used (presumably to draw sets of parameters for an ensemble of simulations).

**Results**

- Fig.3 (left) There is no discussion about why the PDFs for the NOAA dataset differ from all others (in particular GISTEMP, which uses the same base).

- Sec 3.1.1 The prior for H is not mentioned.

- L337 Here, the theoretical value is given as 0.32, but in L220 it was given as $\approx 0.3$.

- L341 I would suggest starting a new section for the total forcing series here, instead of keeping the discussion of the full forcing series and Greens functions under the volcanic forcing header.

- The size of the ensembles is not mentioned, nor how large they had to be such that the probability functions converge.

- L347 The notations $G_1$ and $G_2$ were not used before. Please make this consistent throughout the paper.

- L349-353 The wording "singular response" is unclear, as well as why it allows for the accurate reproduction of "the statistics of the internal variability" and makes FEBE "more sensitive to volcanic forcings. It seems more sensitive to everything up to $\approx 300$ years. It is also not explained why this is a better representation of the historical datasets. It is unlcear why 25 years in particular are mentioned when the gap remains similarly large beyond that.

- L354 Please give a reference for this as the standard ramp time. I am unsure whether the implied meaning of multi-year as anything starting from 2 was intended here.

- L355-356 Having the discussion of the projections before they have been shown might not be the best choice.

- end of Sec.3.2: Unclear whether $G_1$ or $G_2$ was used for the projections.

- Fig. 6:

  a) The data in the inset (or the axes) does not seem to fit the data shown in the full plot. This is particularly notable with the IPCC model. The x-axis of the shown inset and the inset area marked in the full plot do not agree.

  b) (top) Even in the inset, small times are barely visible. This is made worse by the fact that the border of the box does not align with the y-axis. The labels for the insets are too small, those for the lower plot are barely legible.

  c) $G_2$ is not discussed in the text, please change this.

- There is a lot of emphasis put on the results with FEBE being less uncertain (i.e. L372) than for CMIP, but this does not necessarily mean that they are a better representation of climate. In addition, I would like to know how much this is due to the CMIP ensemble consisting of different models versus the FEBE ensemble being produced with one model.

- L382 The median for $\lambda$ in CMIP6 is not given.

- L393-399 Unclear how much of the higher RWF in FEBE is due to energy balance models usually responding faster than GCMs. A discussion on how reliable the FEBE estimate is in comparison to the CMIP one is missing.

- Sec. 3.3.1 and 3.3.2 Might be interesting to discuss why both ECS and TCR best estimates are smaller for the SSP than the RCP scenarios with FEBE, but it is the opposite in CMIP.

- Fig. 8:The figure would be easier to read if the single plots were aligned with each other according to their axes and were shown with the same font size.

**Projections**

- L417 The CMIP5 MME is mostly warmer for the given period, but not always (compare i.e. the early 1940s where observed temperatures are higher than the mean). Why the upper limit of 1960 was chosen is unclear, since the CMIP simulations remain warmer most of the time afterwards (and in particular, after 2000).

- L419-422 It is not discussed whether FEBE tracking the hiatus is a sign of FEBE's skill or a results of the fact that the parameters were chosen based on the same observations that include this hiatus. Since EBMs usually have trouble with this hiatus, it would be interesting to discuss whether there exist other conceptual models that reproduce the hiatus or whether this is a standout feature of FEBE.

- L421 There is no lower plot in fig. 9.

- Fig. 9: The observations show strong interannual variability, a discussion on whether a temporally higher resolved FEBE could capture such short-term variability (or even whether a higher resolution in time is feasible with this approach). Equally interesting would be to discuss whether other response functions could be designed to capture variability on arbitrary timescales. A bit unclear whether FEBE is useful for discussions of variability at all or should be applied only to discussions of the mean.

- L461-462 According to the section on aerosol forcings (fig. 1), there is a future aerosol forcing in the RCP and SSP scenarios. It is unclear whether here the removal of the volcanic forcing is meant. Since the forcings in FEBE and SSP should be consistent (with the exception to the dampening parameters for the aerosols mentioned in the first part of the sentence) uncler why this would produce the temperature difference.

- The effect of the lack of volcanic forcing in the future projections is not discussed, nor whether (and if so, how) the authors suggest its inclusion in future studies.

- Fig. 10: The sine-like oscillations in the future projections by FEBE (esp. RCP2.6 and RCP4.5) should be mentioned in the text. I assume they reflect the solar forcing cycle, but remain uncertain whether the different magnitude of this cycle in the scenarios is due to a difference in overall magnitudes of forcings between scenarios.

- Fig. 11: It is not discussed why these FEBE projections are notably smoother than the ones in fig. 9 and whether this is only a reflection of the difference in forcing.

- A discussion about what could lead FEBE to overestimate future warming would be interesting and in particular whether it can be argued based on the projections in Fig. 10 & 11 that FEBE provides a baseline of minimal warming.

- L478-479 It seems to me that this phrasing is slightly misleading. It reads as if extremely likely refers to the 15 years. However, it seems like 15 years later is just the point when it becomes extremely likely that the 1.5° threshold is exceeded.

- Fig.12 bottom: It is not mentioned that the solid RCP scenario is at 0. Would be good to change the plot so that this becomes visible. Would also help to mark the "extremely likely" threshold at 0.95 discussed in the text.

- Fig.13 top: It is not discussed why the probability for the SSP126 scenario decreases so strongly after 2070 in the FEBE ensemble.

**Conclusions**

- L482-487: This seems more like a motivation suitable for the introduction than a conclusion.

- It is not discussed for what kind of studies the authors suggest the usage of FEBE, nor what the gaps are in studies with more complex models that FEBE might fill.

- L568 The authors say that their goal is to improve future projections, but do not expand on how they think that will be possible with FEBE.

- L569 It is unclear how FEBE can be used to understand the generational differences between CMIP models better.

**Code and data availability**

- Neither the model code nor the simulations with the FEBE are included in a code/data availability section and made available to the reader.

**4   Technical corrections**

**General comments**

- citation style hinders readability (brackets even when the citation is part of the sentence and double brackets when listing several references)

- The whole text needs to be checked for correct comma placement, especially with respect to introductory phrases and interrupters. The punctuation could also be improved with respect to colon and semicolon usage.

- in this state of the manuscript it did not seem to make sense to give detailed feedback on specific grammatical errors, so these are not included in this review

- the way "FEBE" is used in the text is inconsistent, sometimes it is "the FEBE", at other times just "FEBE" without an article

- inconsistent usage of "figure X" or "fig. X" in the text

- There are quite a lot of unnecessarily run longing sentences that hamper readability.

- In general, the plots could be cleaner, i.e. for multi-panel plots the plot sizes as well as font sizes differ.

- The linkage between sentences, paragraphs and sections could be improved to make the reader's life easier. Also the internal structure of sections, paragraphs and even sentences could be more coherent. For example, it's not always easy to identify the topic and stress in a sentence. This could be facilitated by an improved structure of sentences. Sometimes, the connection between subject and verb gets lost due to a sentence structures that is too complicated. In general, reducing long sentences and technical language would improve the reader's understanding.

**Abstract**

- L20 MME used without introducing the abbreviation

- L21 should be "the FEBE projections were...", also comma missing after the introductory phrase

**Introduction**

- L40 different dashes are used at the beginning and end of the insertion, "that" after the insertion should be discarded (whole sentence could benefit from being reformulated for better readability, though)

- L63 MME as an abbreviation is only introduced in L94

- L83 should be Hébert et al.'s truncated power law or at least it has not been introduced so far as having been developed by only one person

- L89 article missing in "In the methods and materials section"

**Methods and Material**

- L138 should have a reference to where that discussion happens

- L165 "While RCP8.5 and SSP585..." subject of the sentence is plural, rest of sentence is singular

- L171 "and" between "Kyoto protocol" and "ozone depleting substances" would make this sentence easier to read

- L197 should be "two 11-year solar cycles"

**Results**

- L327 the last part of the sentence lacks a verb

- L229 should be "find in/for both cases" or similar

- L348 no apostrophe for " its' "

- L349 "where" or similar instead of "whereas"

- L362 "$CO_2$ levels were" or "$CO_2$ was increased"

**Projections**

- L415 rogue bracket at the end of the sentence

- L416 either "Between 1915-1960" or "In the 1915-1960 period" or similar

- Fig.9 There are white artefacts in the grid lines, box borders, and axes of the plot as well as in the upper limit of the CMIP5 MME. It is unclear why the x-axis of the inset unneccessarily includes negative values. The plot would appear cleaner if the borders of the inset were aligned with the axes. CMIP5 MME is in gray not black (same in later plots). Referring to the inset as the top is confusing. The left border of the inset in the whole plot is placed too late.

- L431 scratch "In comparison" at the beginning of the sentence (at this point it is unclear with respect to what the comparison is supposed to be and it does not match the remainder of the sentence)

- L448 forcings should be plural in "the other forcing are practically"

- L454 doubled "the"

- L461 drop "by" in "are by nearly 65% warmer"

**Conclusions**

- L547 "to" missing in "purely due differences"

- L556 I would suggest replacing "be passed" with "happen" in the context of this sentence

**References**

- Held et al. citation not in correct place in alphabetical order

**References**

Hébert, R., Lovejoy, S., and Tremblay, B.: An observation-based scaling model for climate sensitivity estimates and global projections to 2100, Climate Dynamics, https://doi.org/10.1007/s00382-020-05521-x, URL https://doi.org/10.1007/s00382-020-05521-x, 2020.

Schmidt, G. A., Jungclaus, J. H., Ammann, C. M., Bard, E., Braconnot, P., Crowley, T. J., Delaygue, G., Joos, F., Krivova, N. A., Muscheler, R., Otto-Bliesner, B. L., Pongratz, J., Shindell, D. T., Solanki, S. K., Steinhilber, F., and Vieira, L. E. A.: Climate forcing reconstructions for use in PMIP simulations of the last millennium (v1.0), Geosci. Model Dev., 4, 33–45, https://doi.org/10.5194/gmd-4-33-2011, 2011.

---

## Referee Comment (RC3) · Anonymous Referee #3 · 20 Jan 2021

The authors apply a fractional energy balance equation to model temperature variations. They formulate the model in a Bayesian framework. This approach is used to perform temperature projections, and estimate the equilibrium climate sensitivity and the transient climate response. These are compared with the CMIP5 and CMIP6 projections. The paper is mostly well written and the topic fits the scope of the journal. I find the paper worthy of publication in ESD. However, I would recommend some major revisions before the paper is ready to be published.

The manuscript provides, at times, incomplete details on how the results were obtained and how the approach was designed, making reproducibility of the results difficult. I would suggest sufficient details be added. This would also help make potential errors in the approach more detectable.

[Figure]

I would also like to see some discussion on approximating the joint prior probability density functions with a multivariate Gaussian process, including what motivated this decision, how accurate this approximation is and if this could have any repercussions for the final results. How the covariance between the parameters used in the joint prior distribution is determined should also be included.

Other than this I have some minor comments and suggestions:

- Physical units and chemical formulas should not be typeset in italics.

- Ranges should be denoted using en dashes (–), instead of hyphens (-), e.g. 1998–2015.

- Top panel in figure 6: The horizontal axis of the smaller window ranges from 0 to 25 years, but the dotted lines suggest it is taken from the range of 0 to 50 years from the larger figure. Please correct/clarify.

- Line 274: The likelihood function is said to be a posterior probability. This is incorrect, as the likelihood is a distinct probability distribution. Also in line 274: the terms "posterior" and "a priori" seem to have been switched.

- Being in the Bayesian modeling framework, where parameters are treated as stochastic variables instead of fixed and true values, I think it would be more appropriate to use the term "credible intervals" instead of "confidence intervals". The former refers to an interval within which an unobserved (stochastic) parameter value falls with a particular probability, whilst the latter is an interval that we are, to a certain degree, confident include the true (deterministic) parameter value.

---

## Author Comment (AC1) · 18 Mar 2021

Roman Procyk et. al.

**Figures 9 to 11 show the current and future evolutions of the global temperature as compared with the observations and the CMIP5 simulations. These projections are provided as ensembles in order to cover the range of uncertainties related to the model settings. An observational uncertainty is also provided. It is obvious that the spread of the ensemble is much smaller for the FEBE model than for the CMIP5 ensemble in Figure 9, and very often the observation band is not falling into the range of predicted projections of FEBE, while it is the case for the CMIP5. This suggests that the projections are unreliable in the sense that it does not cover all possible situations that can be observed. In weather and climate sciences, this aspect is key when making forecasts, predictions and projections using ensembles. This unreliability should be first acknowledged in the manuscript. Second this unreliability is maybe related to the use of a weak stochastic forcing. Some experiments with larger stochastic forcing would be desirable in order to clarify the under-dispersion of the ensemble. Stronger stochastic forcing could also maybe lead to larger climate sensitivity. This should be checked too.**

Author: The basic purpose of Figures 9 to 11 was to compare the FEBE and GCM projections. In both cases, the projections are deterministic but with uncertainty limits due to their respective model uncertainties. Both yielded an estimate of the forced response but with qualitatively different uncertainty bounds. In the case of GCMs, the uncertainty is termed "structural" while for the FEBE it is parametric uncertainty. Unlike a probabilistic forecast, the results cannot be interpreted with the help of stochastic forecast notions such as reliability.

We can see why the referee may have misunderstood, since we compared the forced and internal components with the observed temperature series. This was intended as a quick visual validation of the forced component but since it contains the (stochastic) internal variability, strictly speaking, it should not be directly compared to the forced component.

In order to make a proper comparison with data, we first removed the observations from the figures so as to only compare the forced component (new figs. 10, 11 to replace the original figs. 10, 11). All the forced components are deterministic (the internal variability has been averaged out), for the GCMs the uncertainty is structural uncertainty, while for the FEBE it is parametric uncertainty. We then replaced the original figure 9 with one that represents the ensemble average over all the parametric and internal variability of the FEBE and the mean observational temperature (shown in new figs. 9a, 9b). This was achieved since the statistical dependence of the internal forcing and the parametric uncertainty are independent: the errors therefore add in quadrature. For this, it is sufficient to take the globally averaged yearly temperature anomaly ($\approx \pm 0.11C$) and combined it with the annual resolution parametric forced component from figs. 10, 11 over the historical period (1880-2020). The new figs. 9a,9b shows this result. This can also be done at monthly resolution following the same procedure but using the globally averaged monthly temperature anomaly ($\approx \pm 0.14C$), shown in figs. 9c, 9d.

The temperature observations do indeed fall within the 90% confidence limits of the FEBE historical reconstruction (i.e. the ensemble average of the response to both internal and external forcing). In both figures at annual resolution shown below (figs. 9a,9b), the historical mean temperature (red) is within the 90% CI of the FEBE forced response (with internal variability added) 92% of the years using the RCP scenario, and 94% using the SSP scenario. At the monthly resolution shown in figs. 9c, 9d, the historical mean temperature (red) is within the 90% CI of the FEBE forced response (with internal variability added) 90% of the months using the RCP scenario or the SSP scenario. The uncertainty is therefore compatible with the data. This is not the same as the reliability but it is an analogous validation of probabilistic aspects of the projection.

[Figure]

Fig. 9a: The historical reconstruction (forced temperature response and internal variability) of the FEBE, with parameters calibrated using $F_{Aer_{RCP}}$ (blue) alongside mean of 5 observational temperature series (red) at yearly resolution; 90% CI (due to parametric uncertainty and internal variability) are indicated (shaded).

[Figure]

Fig. 9b: Same as fig. 9a except using $F_{Aer_{SSP}}$ parameters and forcing.

[Figure]

Fig. 9c: The historical reconstruction (forced temperature response and internal variability) of the FEBE, with parameters calibrated using $F_{Aer_{RCP}}$ (blue) alongside mean of 5 observational temperature series (red) at monthly resolution; 90% CI (due to parametric uncertainty and internal variability) are indicated (shaded).

[Figure]

Fig. 9d: Same as fig. 9c except using $F_{Aer_{SSP}}$ parameters and forcing.

[Figure]

Fig. 10: The deterministic forced temperature response projected using the FEBE, with parameters calibrated using $F_{Aer_{RCP}}$ (blue) compared with the CMIP5 MME projection (black); 90% CI from the parametric uncertainty are indicated (shaded). The projections until 2100, for RCP 2.6 (top), RCP 4.5 (middle) and RCP 8.5 (bottom), are shown.

[Figure]

Fig. 11: The deterministic forced temperature response projected using the FEBE, with parameters calibrated using $F_{Aer_{SSP}}$ (blue) compared with the CMIP6 MME projection (black); 90% CI from the parametric uncertainty are indicated (shaded). The projections until 2100, for SSP 126 (top), SSP 245 (middle) and SSP 585 (bottom), are shown.

**At page 20, line 421, the authors claim that the hiatus is better represented in the FEBE than in the CMIP5 ensemble. Well to me this is not true as the observations are most of the time out of the range of FEBE. The CMIP5 looks better at capturing the observations. So I suggest to modify these comments and try to be more objective in the comparison, maybe by using measures of reliability.**

Author: We discussed the reliability in the previous responses, by showing the observed temperature series along with the projection of the forced response plus the internal variability response. With the internal variability we expect the data to lie within the 90% CI, 90% of the time. We also discussed how historical reconstructions were made (figs. 9a, 9b, 9c, 9d). We now discuss the latter over the hiatus period. In the insets of figs. 1a, 1b, we show the comparison of the forced median FEBE projection (blue) (using $F_{Aer_{RCP}}$ in fig. 1a, and $F_{Aer_{SSP}}$ in fig. 1b), the CMIP5/6 MME median (black) and the mean of 5 observational temperature series (red) with the 90% CI over the historical period with the inset showing the hiatus period (a blow-up of 1998-2015). We see that indeed, the FEBE median forced component in both cases captures the hiatus rather accurately (see Lovejoy (2015) for a stochastic forecast with a similar high frequency limit).

A quantitative comparison between the amount of time the FEBE median response is within the bounds of the observational temperature series 90% CI and the same for the CMIP5/6 MME was performed at the annual resolution data. The amount of time the median FEBE forced component using $F_{Aer_{RCP}}$ is within the 90% CI of the observational temperature series over the whole historic period is 47% and over the hiatus is 70% in comparison to the CMIP5 MME median which is within the whole historic period 39% and over the hiatus is 17%. When using the median FEBE forced component using $F_{Aer_{SSP}}$ similar results are found, over the whole period: 45% and over the hiatus: 35%, in comparison to the CMIP6 MME median which is within the whole historic period 39% and over the hiatus is 30%. In can be seen in both cases that the CMIP MME is generally warmer than the FEBE forced component notably over the period of the hiatus.

Lovejoy, S. (2015), Using scaling for macroweather forecasting including the pause, Geophys. Res. Lett., 42, 7148– 7155, doi:10.1002/2015GL065665.

[Figure]

Fig. 1a: The median historical forced component of the FEBE, with parameters calibrated using $F_{Aer_{RCP}}$ (blue), and the median of the CMIP5 MME (black) alongside mean of 5 observational temperature series (red) with their 90% CI indicated (shaded).

[Figure]

Fig. 1b: The median historical forced component of the FEBE, with parameters calibrated using $F_{Aer_{SSP}}$ (blue), and the median of the CMIP6 MME (black) alongside mean of 5 observational temperature series (red) with the 90% CI indicated (shaded).

One key conclusion of the manuscript is the lower sensitivity of the FEBE model on long time scales as shown for instance in Figure 10. What is happening when the FEBE model is fitted with a stronger noise that could maybe help in increasing the spread of the ensemble and its reliability? The results and the conclusions should then be revisited once these experiments are done and the impact of the stochastic noise clarified.

Author: The the amplitude of the noise is not an adjustable parameter, it is determined from the data. Although the empirical amplitude of the internal variability does not alter the projections of the forced temperature response, it does affect the uncertainties of the parameters (once these have been estimated, the resulting projection is purely deterministic).

**Equation 14: Please clarify what is the variance of gamma(t)**

Author: The variance of $\gamma(t)$ is the amplitude of the internal forcing assumed to be a Gaussian white noise. The internal variability of the observational temperature is equal to the observed series with the forced temperature response removed. If we take the global annually averaged monthly temperature anomaly to be $\sigma_{T,\tau_r} \approx \pm 0.14 C$, we can determine the variance of $\gamma(t)$ from Lovejoy et. al (2021):

$$K_h = \sqrt{\frac{\pi}{2cos(\pi\left(h - \frac{1}{2}\right))\Gamma(-1-2h)}},$$

$$\sigma_{f,\tau_r} = \frac{\sigma_{T,\tau_r} K_h}{s}\left(\frac{\tau}{\tau_r}\right)^h.$$

Where $K_h$ is a standard normalization constant chosen for convenience, $\tau$ is the relaxation time, $\tau_r$ is the resolution (taken to be monthly in this case), $s$ is the climate sensitivity parameter, $\sigma_{T,\tau_r}$ is standard deviation of the globally averaged monthly temperature anomaly at resolution $\tau_r$, $h$ the scaling exponent of the temperature fluctuations, and $\sigma_{f,\tau_r} = \gamma(t)$ is the standard deviation of the internal forcing. Using our $F_{RCP}$ (and $F_{SSP}$) parameter estimates, we find a mean estimate of the variance of $\gamma(t)$ to be 1.15 $Wm^{-2}$ (1.29 $Wm^{-2}$) and 90% CI of $[0.89, 1.42]$ $Wm^{-2}$ ($[0.99, 1.65]$ $Wm^{-2}$). If we introduce a white noise forcing, $\gamma(t)$, with the variance calculated above to the FEBE we will be able to recreate the the amplitude of the internal temperature variability response. This will be included in the revision.

Harries et. al. (2010) sets out to examine the net energy flux balance at the top of atmosphere (TOA) measured using observations from polar-orbiting spacecraft. The early observations, using the Nimbus experiments, show an internal variability of the $4.1 \pm 4.0 Wm^{-2}$, while more modern measurements (CERES) in the 2000s show variability of between $\pm 2$ and $\pm 4$ $Wm^{-2}$ generally laying a few $Wm^{-2}$ of zero. Thus our estimate of the internal forcing variability

is within estimates of the TOA net energy flux balance.

Lovejoy S, Procyk R, Hébert R, Rio Amador L. The fractional energy balance equation. QJR Meteorol Soc. 2021;1–25. https://doi.org/10.1002/qj.4005

**Figure 6: The response of FEBE is compared with an IPCC two-box model. What is this model? Maybe I missed the place where it is described. Please describe this model in more details in the text**

Author: The two-box model we are referring to is the classical linear two-layer energy-balance model described in Held et al. (2010) and found in IPCC AR5 (2013, section 8.SM.11.2):

$$C\frac{dT}{dt} = F - \lambda T - \gamma(T - T_0),$$
$$C_0\frac{dT}{dt} = \gamma(T - T_0).$$

We graphically showed the comparison in fig. 6. The parameters for the two-box model are the best estimates from Geoffroy et al. (2013).

Held, I. M., M. Winton, K. Takahashi, T. Delworth, F. Zeng, and G. K. Vallis, (2010): Probing the fast and slow components of global warming by returning abruptly to preindustrial forcing. J. Climate, 23, 2418–2427. https://doi.org/10.1175/2009JCLI3466.1

Geoffroy O, Saint-Martin D, Olivié DJ, Voldoire A, Bellon G, Tytéca S (2013) Transient climate response in a two-layer energy-balance model. part I: analytical solution and parameter calibration using CMIP5 AOGCM experiments. J Clim 26:1841–1857. https://doi.org/10.1002/env.2140

**Figure 12: Please correct the colors of the curves. I cannot figure out what is plotted**

Author: Thank you for pointing this out, the colours will be changed as to be easily readable.

---

## Author Comment (AC2) · 18 Mar 2021

Roman Procyk et. al.

**An assessment of scientific soundness is difficult at this point, in particular since information necessary for the reproduction of the results is not provided. There are errors in some of the figures (listed in detail below) and their quality as well as that of the general sentence structure could be improved for easier comprehension by the reader.**

Author: The FEBE projections are deterministic i.e. for a given set of parameters, our projection represents the average over an infinite ensemble. However, the parameters are uncertain. Therefore, the projections have uncertainty even though each is deterministic. In order to numerically estimate the ensemble mean projection and the uncertainty in the projections it is convenient to use Monte Carlo methods. However it should be clear that this is simply a convenient numerical technique that does not imply any stochasticity in the projections. However, using Monte Carlo methods to determine the projections introduces standard Monte Carlo numerical uncertainty, but this is made quite small by using large numbers of Monte Carlo realizations.

For the FEBE, an ensemble of realizations is used throughout the paper. The ensemble for the FEBE calibrated using either the CMIP5/6 is included in the revision to allow for the reproduction of results; the full FEBE code will be publicly released upon publication and can be shared immediately with the reviewer.

**There are a lot of gaps in the discussion of the model and results. Limitations of FEBE are not described at all, nor whether FEBE can be used for studies other than those concerning modern and future global mean temperature. The potential for future studies or extensions are also skimmed over.**

Author: The fundamental limitations of the FEBE are "structural" (is the model a good model?), and parametric (how well have we estimated the parameters?). As usual, the first is difficult to answer, we have now replaced the original fig. 9 with figs. 9a, 9b which show a comparison of the ensemble averaged hindcast compared with past data. In both figures shown below (figs.

9a,9b) at annual resolution, the historical mean temperature (red) is within the 90% CI of the FEBE forced response (with internal variability added) 92% of the years using the RCP scenario, and 94% using the SSP scenario. At the monthly resolution shown in figs. 9c, 9d, the historical mean temperature (red) is within the 90% CI of the FEBE forced response (with internal variability added) 90% of the months using the RCP scenario or the SSP scenario. At the monthly resolution shown in figs. 9c, 9d, the historical mean temperature (red) is within the 90% CI of the FEBE forced response (with internal variability added) 90% of the months using the RCP scenario or the SSP scenario. The uncertainty is therefore compatible with the data and shows a high degree of internal consistency in the model and in the Bayesian parameter estimates. We also note in the revision that the end of war (1945) temperature spike which lays out of the FEBE 90% CI may be explained due to biases explained in Chan and Huybers (2021). Physically, the FEBE agrees with two dynamical conservation principles: scale invariance of the storage and energy conservation. The parametric uncertainty is the uncertainty that we can quantify, it arises essentially due to poor knowledge (uncertainty) in the past forcing. The FEBE has already been generalized to a regional model (Lovejoy 2021) and we are currently preparing regional projections using it. In the future, it will be extended to nonlinear feedbacks (e.g. temperature albedo feedbacks). We have added some material in the discussion and conclusions sections on this.

Lovejoy, S.: The Half-order Energy Balance Equation, Part 2: The inhomogeneous HEBE and 2D energy balance models, Earth Syst. Dynam. Discuss. https://doi.org/10.5194/esd-2020-13, 2021.

Chan, Duo, and Peter Huybers. " Correcting Observational Biases in Sea-Surface Temperature Observations Removes Anomalous Warmth during World War II". Journal of Climate (2021): 1-44. https://doi.org/10.1175/JCLI-D-20-0907.1.

[Figure]

Fig. 9a: The historical reconstruction (forced temperature response and internal variability) of the FEBE, with parameters calibrated using $F_{Aer_{RCP}}$ (blue) alongside mean of 5 observational temperature series (red) at monthly resolution; 90% CI (due to parametric uncertainty and internal variability) are indicated (shaded).

[Figure]

Fig. 9b: Same as fig. 9a except using $F_{Aer_{SSP}}$ parameters and forcing.

[Figure]

Fig. 9c: The historical reconstruction (forced temperature response and internal variability) of the FEBE, with parameters calibrated using $F_{Aer_{RCP}}$ (blue) alongside mean of 5 observational temperature series (red) at monthly resolution; 90% CI (due to parametric uncertainty and internal variability) are indicated (shaded).

[Figure]

Fig. 9d: Same as fig. 9c except using $F_{Aer_{SSP}}$ parameters and forcing.

**The reader does not get an idea of what the paper adds to Hebert et al. (2020) aside from replacing the truncated power law in the model equation. Especially, with respect to the results a discussion of differences is needed.**

Author: The main differences are at high frequencies as a consequence the FEBE models both the responses to the internal and external forcings and it can better take into account volcanic forcing. Whereas Hebert et. al. (2020) makes an ad hoc model for the internal variability, in the FEBE, the internal variability determines in the order h of the FEBE. A paragraph will be added exploring the differences between the model and results in Hebert et al. (2020).

The FEBE is able to estimate the forcing amplitude to reproduce the internal variability of the temperature (see Lovejoy et. al (2021)). This is discussed below and will be included in the text, highlighting the new aspect of the FEBE which allow us to physically model the internal variability compared to Hebert et al. (2020). A further discussion on the variance of $\gamma(t)$ along with estimates are in the response to referee 1, and will be included in the revision.

Lovejoy S, Procyk R, Hébert R, Rio Amador L. The fractional energy balance equation. QJR Meteorol Soc. 2021;1–25. https://doi.org/10.1002/qj.4005

**The research is not sufficiently embedded in other literature on the topic of energy balance modeling in both the introduction and discussion.**

Author: More background references and connections to standard literature will be included in the revision.

Trenberth, Kevin E., John T. Fasullo, and Jeffrey Kiehl. " Earth's Global Energy Budget". Bulletin of the American Meteorological Society 90.3 (2009): 311-324. https://doi.org/10.1175/2008BAMS2634.1

Rypdal, Martin, and Kristoffer Rypdal. " Long-Memory Effects in Linear Response Models of Earth's Temperature and Implications for Future Global Warming". Journal of Climate 27.14 (2014): 5240-5258. https://doi.org/10.1175/JCLI-D-13-00296.1

North, G. R., Kim, K. Y. (2017). Energy Balance Climate Models. John Wiley Sons.

Proistosescu, C., Donohoe, A., Armour, K. C., Roe, G. H., Stuecker, M. F., Bitz, C. M. (2018). Radiative feedbacks from stochastic variability in surface temperature and radiative imbalance. Geophysical Research Letters, 45, 5082–5094. https://doi.org/10.1029/2018GL077678

Ziegler, E. and Rehfeld, K.: TransEBM v. 1.0: Description, tuning, and

validation of a transient model of the Earth's energy balance in two dimensions, Geosci. Model Dev. Discuss. https://doi.org/10.5194/gmd-2020-237, 2020.

**L12 "We found that aerosols..." should mention that here aerosols refers to the CMIP aerosol forcings**

Author: Thank you, this will be clarified.

**L22 It is unclear how FEBE complements (in addition to support) GCMs based on what is said.**

Author: The FEBE projections to 2100 are entirely independent of the GCMs. Still, they are within the uncertainty bounds of the latter, effectively providing an independent confirmation of the GCM projections. This eliminates one of the key climate skeptic arguments: projections are not reliable since they are solely GCM-based.

It also has the advantage that the uncertainties are significantly reduced. This is important for policy (see the conclusions).

Another advantage is that it gives us information about the past forcings, especially the aerosols (also to a lesser extent about volcanic forcing). It supports numerous papers that find that the historic aerosol forcing in CMIP models was too strong (Stevens (2015), Zhou and Penner, (2017), Sato et al., (2018), Bellouin et al., (2020)). We could also diagnose what is wrong with models with respect to too strong aerosol and volcanic forcings by calibrating the FEBE on the historical temperature reconstruction of the model.

Stevens, B. " Rethinking the Lower Bound on Aerosol Radiative Forcing". Journal of Climate 28.12 (2015): 4794-4819. https://doi.org/10.1175/JCLI-D-14-00656.1

Zhou, C. and Penner, J. E.: Why do general circulation models overestimate the aerosol cloud lifetime effect? A case study comparing CAM5 and a CRM, Atmospheric Chemistry and Physics, 17, 21–29, https://doi.org/10.5194/acp-17-21-2017, https://www.atmos-chem-phys.net/ 17/21/2017/, 2017.

Sato, Y., Goto, D., Michibata, T., Suzuki, K., Takemura, T., Tomita, H., and Nakajima, T.: Aerosol effects on cloud water amounts were successfully simulated by a global cloud-system resolving model, Nature Communications, 9, 985, https://doi.org/10.1038/s41467-018- 03379-6, https://doi.org/10.1038/s41467-018-03379-6, 2018

Bellouin, N., Quaas, J., Gryspeerdt, E., Kinne, S., Stier, P., Watson-Parris, D., Boucher, O., Carslaw, K. S., Christensen, M., Daniau, A.- L., Dufresne, J.-L., Feingold, G., Fiedler, S., Forster, P., Gettelman, A., Haywood, J. M., Lohmann, U., Malavelle, F., Mauritsen, T., McCoy, D. T., Myhre, G., Mülmenstädt,

J., Neubauer, D., Possner, A., Rugenstein, M., Sato, Y., Schulz, M., Schwartz, S. E., Sourde- val, O., Storelvmo, T., Toll, V., Winker, D., and Stevens, B.: Bounding Global Aerosol Radiative Forcing of Climate Change, Reviews of Geophysics, 58, e2019RG000 660, https://doi.org/10.1029/2019RG000660, 2020.

**The abstract summarized well what is done in the paper, but fails to address limitations and the gap in literature it aims to fill.**

Author: Ok this will be revisited to better address the FEBE with respect to the literature and area of EBMs.

**L35 reference for the projection uncertainty in response to CO2 doubling missing**

Author: Thank you, the reference will now be included.

IPCC (2013), Climate Change 2013: The Physical Science Basis (AR5), edited by T. F. Stocker, D. Qin, G.-K. Plattner, M. Tignor, S. K. Allen, J. Boschung, A. Nauels, Y. Xia, V. Bex, and P. M. Midgley, Cambridge University Press, Cambridge, U. K.

**L43 should be motivated why it is desirable to construct models beyond linearity and stochasticity**

Author: By themselves, these constraints are insufficient to specify any particular model, they remain too general. We have added a comment on this.

**L69-72 These explanations require improvement: a) The EBE has not been explained enough for anyone who is not familiar with the model to fully comprehend this sentence. The reader has not been made aware that a storage term exists yet. b) Maybe include a short discussion of other for the storage term besides the one chosen.**

Author: We have added a sentence explaining that the earth energy balance is not exact: at any instant there is a difference between incoming and outgoing radiation, this difference is stored (or comes out of storage). The usual storage is assumed to be in thermodynamic (molecular) degrees of freedom with one or two slabs or "boxes" instantaneously changing their temperatures to account for any radiative imbalances. (This is a property of global models such as the model discussed here: for regional models horizontal divergence of heat fluxes can also be important).

**L118 Please mention why causality is not respected otherwise.**

Author: The equation expresses causal antecedence: cause must precede effect. This is a necessary part of the causal principle, If this property is not held,

then the future forcings can affect the past temperatures (i.e. the convolution for any given time would integrate both past and future forcing).

**L121 Please explain why the step response function is "more physical compared to the impulse response". It is not necessarily clear to the reader which response function will be used in which studies. This could be further elaborated in the text to clarify that issue.**

Author: Dirac and Step response functions are mathematically equivalent with the latter corresponding to the more usual experiments (e.g. CO2 doubling). The step function response shows how the model approaches energy balance whereas the Dirac response shows the effect of a short perturbation. We will replace "more physical" with "easier to physically interpret" and make clear in the text that they are equivalent.

**L130 a short explanation of what fractional Relaxation noise is would be helpful**

Author: We will add a short explanation: it is the response of the FEBE to a white noise forcing.

**The implementation of the model is not described at all.**

Author: The model is calibrated over the historical temperature observations using a Bayesian scheme. This will be better introduced and explained throughout sec 2.3 in the revision.

**It should be mentioned explicitly that the model produces annual data (same for the temporal resolution of the forcing).**

Author: The model produces data at the same resolution as the forcing, be it annual or monthly (see figs. 9a, 9b for annual resolution, and figs. 9c, 9d for monthly resolution). This will be explicitly mentioned in the revision. In fact, monthly forcing and response was used to fully exploit the volcanic data and better constrain the model parameters. For clarity we showed annual resolution for all projection plots, this will be clearly indicated.

**L135-138 Would help to start a new paragraph here to clarify that now the text refers back to equations 5 and 6 for the H=1 case and not to the ones discussed in the meantime. It would also be helpful to the reader if it was shown how the function from Hebert et al. With Hf =0.5 is the same as eq. 6 with H=0.5 and why this is the same as "that corresponding to the internal forcing".**

Author: Thank you for mentioning this point. $H_f$ is the asymptotic exponent determining the convergence of FEBE to a step forcing, mathematically it

is given by the asymptotic expansion of the step Green's function that shows that $H_f = -h$ (note that we changed the FEBE order "H" to "h" to avoid confusion with numerous exponents "H" used in similar circumstances). This will be further explained in the revision.

**L145 It is unclear why 277ppm is the reference.**

Author: It is a standard reference value, the reference value is the pre-industrial CO2 concentration taken from page 140. We added the reference to the text.

Forster, P., V. Ramaswamy, P. Artaxo, T. Berntsen, R. Betts, D.W. Fahey, J. Haywood, J. Lean, D.C. Lowe, G. Myhre, J. Nganga, R. Prinn, G. Raga, M. Schulz and R. Van Dorland, 2007: Changes in Atmospheric Constituents and in Radiative Forcing. In: Climate Change 2007: The Physical Science Basis. Contribution of Working Group I to the Fourth Assessment Report of the Intergovernmental Panel on Climate Change [Solomon, S., D. Qin, M. Manning, Z. Chen, M. Marquis, K.B. Averyt, M.Tignor and H.L. Miller (eds.)]. Cambridge University Press, Cambridge, United Kingdom and New York, NY, USA.

**L177 A short elaboration as to why a linear scaling factor is appropriate to describe the uncertainty would be nice.**

Author: The primary uncertainty in the radiative forcing is the total uncertainty in anthropogenic aerosol forcing that arises from the uncertainty in aerosol radiative properties and cloud effects as well as in their concentrations over the industrial period (Penner et al. (2001), Ramaswamy et al. (2001)). We suppose aerosols are known only within a multiplicative-scale factor $\alpha$, which is a unity-mean, normally distributed random variable. This scaling of the magnitude of aerosol forcing is an approach that has been adopted previously by Forest et al. (2002), Harvey and Kaufmann (2002), Forest et al. (2006), and Padilla et al. (2011).

J. E. Penner et al., in Climate Change 2001, The Scientific Basis, J. T. Houghton et al., Eds. (Cambridge Univ. Press, Cambridge, 2001), pp. 289-348.

V. Ramaswamy et al., in Climate Change 2001, The Scientific Basis, J. T. Houghton et al., Eds. (Cambridge Univ. Press, Cambridge, 2001), pp. 349-416.

Forest CE, Stone PH, Sokolov AP, Allen MR, Webster MD. Quantifying uncertainties in climate system properties with the use of recent climate observations. Science. 2002 Jan 4;295(5552):113-7. doi: 10.1126/science.1064419

Harvey, L. D. Danny, and Robert K. Kaufmann. Simultaneously Constraining Climate Sensitivity and Aerosol Radiative Forcing. Journal of Climate 15.20

(2002): 2837-2861. ¡ https://doi.org/10.1175/1520-0442(2002)015¡2837:SCCSAA¿2.0.CO;2.

Forest, C. E., Stone, P. H., and Sokolov, A. P. (2006), Estimated PDFs of climate system properties including natural and anthropogenic forcings, Geophys. Res. Lett., 33, L01705, doi:10.1029/2005GL023977.

Padilla, Lauren E., Geoffrey K. Vallis, and Clarence W. Rowley. "Probabilistic Estimates of Transient Climate Sensitivity Subject to Uncertainty in Forcing and Natural Variability". Journal of Climate 24.21 (2011): 5521-5537. https://doi.org/10.1175/2011JCLI3989.1

**Section 2.2.3 For RCP scenarios it is stated that an aerosol forcing smaller than 1Wm2 is implausible, but not why this is not also the case for the SSP forcing.**

Author: The forcing value itself is implausible, of course this applies to any scenario. We do not say that it applies particularly to the RCP, we only have quoted the AR5 values.

**Fig.1: a) lower plot has two sets of dashed data unlike the upper plot and unlike the plot description b) overlap between data and axis labels in lower plot could be prevented if the axis was at the bottom c) It is unclear why the forcing time series differ in length. d) The upper plot uses "Aer" and not "Ant" in the legend. e) Having the explanations for solid vs dashed vs dotted in the legend, too, would make the plots easier to understand. f) The plot sizes are different, would be nice if they were consistent.**

Author: We have made the following changes to fig. 1, shown below.

a)We have now included the SSP 245 scenario into our updated results. Both plots will have two sets of 3 different dashed lines representing RCP 2.6/SSP 126, RCP 4.5/SSP 245, and RCP 8.5/SSP 585.
b) Ok, this will be improved.
c) The series are equal of length - this is a graphical error and misunderstanding. The plots will be better aligned in the revision. All series run from 1765–2100, the historical (solid) section for the RCP scenarios (blue) goes to 2005, while for the SSP scenarios (black) it extends to 2015.
d) The aerosol forcing of the top plot will be removed, so as to just compare the greenhouse gas forcing of the various scenarios.
e) In the revised image we will have the solid lines be the historical forcing (historic forcings for each scenario group (RCP or SSP) are the same) and RCP 2.6/SSP 126, dashed will be for RCP 4.5/SSP 245, and dotted will be for RCP 8.5/SSP 585.
f) Thank you for pointing this out, it will be fixed.

[Figure]

Fig. 1: (top) The total GHG forcing series, are shown over the historical period and projection period until 2100 for RCP 2.6/SSP 126 (solid), RCP 4.5/SSP 245 (dashed), and RCP 8.5/SSP 585 (dotted). (bottom) The anthropogenic aerosol forcing series used, $F_{Aer_{RCP}}$ (blue) and $F_{Aer_{SSP}}$ (black). Updated from (Hébert et al., 2020). All series run from 1765–2100, the historical (solid) section for the RCP scenarios (blue) goes to 2005, while for the SSP scenarios (black) it extends to 2015.

**L202 It is unclear why 27 was chosen as the conversion factor. With regards to the literature this seems quite high and might also explain why the forcing was overpowered. AR5 recommends 25, Schmidt et al. (2011) 20 and for AR6 18 is suggested. Might be interesting to look at what the effective AOD would have been that the "damped" volcanic forcing the authors ended up using corresponds to.**

Author: The -27 value was taken from Sato (2012), lower values could be used but this is something which the non-linear volcanic parameter $\nu$ takes into account. Performing a linear regression between the non-linear damped volcanic forcing using with best parameter estimate, $\nu = 0.28$ and the AOD we find a conversion factor $\approx -4$, thus even if lower values were used the forcing would still be overpowered.

Sato M (2012) Forcings in GISS climate model : stratospheric aerosol optical thickness. https://doi.org/10.1038/nature04237

**L205 The point of comparison in "comparable effect" is missing.**

Author: This sentence should be rewritten; the volcanic forcings are scaled down (typically using a linear factor by other authors) so to best fit the global temperature response in climate models as found by Lewis and Curry, 2015. See also:

Gregory, J.M., Andrews, T., Good, P. et al. Small global-mean cooling due to volcanic radiative forcing. Clim Dyn 47, 3979–3991 (2016). https://doi.org/10.1007/s00382-016-3055-1. which mention volcanism may be overpowered in a model simulation.

Tomassini L, Reichert P, Knutti R, Stocker TF, Borsuk ME (2007) Robust Bayesian uncertainty analysis of climate system properties using Markov chain Monte Carlo methods. J Clim 20:1239–1254

Ring MJ, Lindner D, Cross EF, Schlesinger ME (2012) Causes of the global warming observed since the 19th century. Atmos Clim Sci 2:401–415

Meinshausen M, Smith SJ, Calvin K, Daniel JS, Kainuma MLT, Lamarque J-F, Matsumoto K, Montzka SA, Raper SCB, Riahi K, Thomson A, Velders GJM, van Vuuren DPP (2011) The RCP greenhouse gas concentrations and their extensions from 1765 to 2300. Clim Change 109(1–2):213–241

**L208-210 An expansion of what it means that a quantification in a multi-fractal framework is possible would be helpful. The explanation of why it must be transformed remains unclear to me. (esp. relevant since a multifractality index is introduced in L218)**

Author: This will be clarified in the revision, a brief explanation can be found in the response to the next comment.

**L218 Would be helpful if the authors would expand on C1 being the codimension of the mean and what the implications of this are.**

Author: Volcanic forcing is peculiar as it is strong and highly intermittent. The intermittency can be quantified by the parameter C1 which corresponds to the fractal codimension (i.e. for a 1-D (time series), it is 1 minus the fractal dimension) characterizing the sparseness of volcanic "spikes" of mean amplitude. (see Lovejoy and Varotsos 2016).

The volcanic response appears to be non-linear as the intermittency ("spikiness", sparseness of the spikes) parameter $C_1$ changes from about $C_{1,F_V} \approx 0.16$ for the input volcanic forcing to $C_{1,T} \approx 0.03$ for the temperature response: the latter is therefore much less intermittent than the former although it is possible that the estimated $C_1$ changes slightly due to finite size effects and internal variability (Lovejoy and Varotsos 2016). When two processes have different intermittencies, they cannot be linearly related. However, by taking appropriate powers, they can be. Therefore to put the volcanic forcing into the linear forcing framework it must first be nonlinearly transformed to that it has the same intermittency as the temperature response.

Lovejoy S., Varotsos C. (2016) Scaling regimes and linear/nonlinear responses of last millennium climate to volcanic and solar forcings. Earth Syst Dyn 7:133–150. https://doi.org/10.1002/wcc.397

**L222 and Eq. 13 This would serve as a good introduction to the whole description of the forcings instead of being hidden away at the end of the description of the volcanic forcing.**

Author: Ok, thanks, will be moved to earlier on.

**L230-234 This sentence is quite complicated, especially with the choice of punctuation. Consider splitting it. Should be "addresses" instead of "address".**

Author: Ok.

**Sec. 2.3: A plot of the time series would be helpful at this point (historic datasets + CMIP5/6 MME). The discussion is needlessly difficult to follow without it.**

Author: A figure will be introduced here of the historical datasets and CMIP5/6 MME.

**L254 The internal stochastic forcing was not actually introduced in the formulas before so the "Recall that" seems misleading.**

Author: Ok, thanks, this will be rewritten.

**L264 Please expand on why low frequencies only weakly influence the likelihood function.**

Author: The calibration of the FEBE is done on historical temperature observation residuals, calculated by removing the forced temperature response (anthropogenic and natural). The residuals contain mainly high frequency variability as most of the low frequency variability has been removed by the forced response (all forcings but volcanic contain primarily low frequency information).

**L288-291 Please expand on how this five-dimensional parameter space was used (presumably to draw sets of parameters for an ensemble of simulations).**

Author: The five-dimensional parameter space was indeed used to draw sets of parameters for an ensemble of simulations - this will be made explicit in the revision. It was the basis of the Monte Carlo technique used to determine the projection mean and uncertainty. Our goal was to estimate the ensemble average over an infinite ensemble, but Monte Carlo methods were used only as a convenient numerical technique.

**Fig.3 (left) There is no discussion about why the PDFs for the NOAA dataset differ from all others (in particular GISTEMP, which uses the same base).**

Author: The difference in PDFs for the NOAA and GISTEMP dataset is most prominent in the PDFs of the scaling exponent H and the volcanic intermittency correction exponent $\nu$. The exact cause of the difference is not clear although it arises from the MLOST dataset's use of a complex frequency algorithm with low-frequency tuning. This low-frequency tuning along with the spatio-temporal smoothing applied in the MLOST dataset is likely the cause of a lower $\nu$ (i.e. a smoother volcanic forcing) along with a slightly higher H (i.e. a smoother temperature response).

Smith, Thomas M., Richard W. Reynolds, Thomas C. Peterson, and Jay Lawrimore. " Improvements to NOAA's Historical Merged Land–Ocean Surface Temperature Analysis (1880–2006)". Journal of Climate 21.10 (2008): 2283-2296. https://doi.org/10.1175/2007JCLI2100.1.

**Sec 3.1.1 The prior for H is not mentioned.**

Author: It was mentioned earlier in the text in Sec. 2.3; we will include the

prior here also.

**L337 Here, the theoretical value is given as 0.32, but in L220 it was given as $\approx 0.3$.**

Author: Thank you for pointing this out, we will make it consistent as $\nu \approx 0.3$.

**L341 I would suggest starting a new section for the total forcing series here, instead of keeping the discussion of the full forcing series and Greens functions under the volcanic forcing header.**

Author: Ok, good idea, thanks.

**The size of the ensembles is not mentioned, nor how large they had to be such that the probability functions converge.**

Author: For the FEBE, the spread of the forced projections is the spread of an infinite ensemble as it is only deterministic so the uncertainty comes from the parametric uncertainty. The probability distributions for the parameters are derived by calibrating FEBE using either the RCP or SSP historical forcing on the historical observed temperature series using a Bayesian scheme explained in Sec. 2.3.

**L347 The notations G1 and G2 were not used before. Please make this consistent throughout the paper.**

Author: Thank you for pointing this out, we will correct this so the notation used is consistently $G_0, G_1, G_2$ instead of $G_\delta, G_\Theta$.

**L349-353 The wording "singular response" is unclear, as well as why it allows for the accurate reproduction of "the statistics of the internal variability" and makes FEBE "more sensitive to volcanic forcings. It seems more sensitive to everything up to $\approx 300$ years. It is also not explained why this is a better representation of the historical datasets. It is unclear why 25 years in particular are mentioned when the gap remains similarly large beyond that.**

Author: For $h < 1$, the impulse (Dirac) response Green's function is mathematically singular at the origin: for small t it behaves as $t^{h-1}$. This makes it particularly sensitive to high frequency forcing and this is mostly from volcanoes.

**L354 Please give a reference for this as the standard ramp time. I am unsure whether the implied meaning of multi-year as anything starting from 2 was intended here.**

Author: The transient climate response (TCR) has traditionally been defined as the change in global mean temperature at the time of a doubling of atmospheric CO2 concentration increasing at a rate of 1% per year, which takes 69 years, (approximately) 70 years (Collins et al. (2013). The implied meaning of multi-year is the scale where high frequency variability unimportant - this will be rewritten in the revision.

Collins M, Knutti R, Arblaster J, Dufresne J-L, Fichefet T, Friedlingstein P, Gao X, Gutowski WJ, Johns T, Krinner G, Shongwe M, Tebaldi C, Weaver AJ, Wehner M (2013) Long-term climate change: projections, commitments and irreversibility. In: Stocker TF, Qin D, Plattner G-K, Tignor M, Allen SK, Boschung J, Nauels A, Xia Y, Bex V, Midgley PM (eds) Climate Change 2013: The Physical Science Basis. Contribution of Working Group I to the Fifth Assessment Report of the Intergovernmental Panel on Climate Change. Cambridge University Press, Cambridge

**L355-356 Having the discussion of the projections before they have been shown might not be the best choice.**

Author: The plot position will be moved.

**end of Sec.3.2: Unclear whether G1 or G2 was used for the projections.**

Author: Either can be used with respect to the linear framework shown in eq. 3 with $G_0$ - the impulse response function. In fact $G_1$ was used since it is smoother (at small time scales, it varies as $t^h$ i.e. it is not singular at $t = 0$), and this makes it numerically attractive. This will be explicitly clarified in the revision in an appendix.

Recall eq. 3, putting $s = 1$:

$$T(t) = \int_0^t G_0(t - u)F(u)du,$$

at small $t$, $G_0(t) \approx t^{h-1}$, i.e. it is singular. To avoid the corresponding numerical issues, we instead first calculate the integral of T:

$$\zeta(t) = \int_0^t T(v)dv.$$

This is given by:

$$\zeta(t) = \int_0^t \left( \int_0^v G_0(v - u)F(u)du \right) dv = \int_0^t G_1(t - v)F(v)dv$$

Where we have used:

$$G_1(t) = \int_0^t G_0(v)dv.$$

At small $t$, $G_1(t) \sim t^h$ which is regular since $h > 0$. Thus using $\zeta(t)$ we determine $T(t)$ using finite differences. To calculate the forced response for the projections we use an efficient numerical convolution algorithm.

**Fig. 6: a) The data in the inset (or the axes) does not seem to fit the data shown in the full plot. This is particularly notable with the IPCC model. The x-axis of the shown inset and the inset area marked in the full plot do not agree. b) (top) Even in the inset, small times are barely visible. This is made worse by the fact that the border of the box does not align with the y-axis. The labels for the insets are too small, those for the lower plot are barely legible. c) G2 is not discussed in the text, please change this.**

Author:

a)The inset area marked on the full plot will be changed to agree with the x-axis range of the inset.
b) Ok, this will be fixed.
c) $G_2(t)$ will be included in a discussion in Sec. 2.1.

**There is a lot of emphasis put on the results with FEBE being less uncertain (i.e. L372) than for CMIP, but this does not necessarily mean that they are a better representation of climate. In addition, I would like to know how much this is due to the CMIP ensemble consisting of different models versus the FEBE ensemble being produced with one model.**

Author: In the case of GCMs, the uncertainty is termed "structural". The multi-model ensemble is treated as a single "super" process. Each GCM is considered to be a different stochastic realization of this super process. It may indeed be difficult to justify this assumption, but that is what the community has used. Questions surrounding the validity of this assumption help motivate and justify the development of other approaches such as the approach discussed here. In comparison, for the FEBE it is the more conventional parametric uncertainty, that is, uncertainty about the values that should be assigned to a climate model's parameters.

In both cases, the projections were deterministic but with uncertainty limits due to their respective model uncertainties and yielded an estimate of the forced component with qualitatively different uncertainty bounds.

**L382 The median for $\lambda$ in CMIP6 is not given.**

Author: The median for $\lambda$ ($s$) is 0.92 $K(Wm^{-2})$ in CMIP6 - this will be updated in the revision, also, we have changed the notation, using the more standard symbol "$s$" for the climate sensitivity.

**L393-399 Unclear how much of the higher RWF in FEBE is due to energy balance models usually responding faster than GCMs. A discussion on how reliable the FEBE estimate is in comparison to the CMIP one is missing.**

Author: The Realized Warming Fraction (RWF) will be further explored in the following revision. The rapidity depends on the response function $G$, and model parameters $\tau, H$. The reliability of the FEBE is discussed in the response to referee 1.

**Sec. 3.3.1 and 3.3.2 Might be interesting to discuss why both ECS and TCR best estimates are smaller for the SSP than the RCP scenarios with FEBE, but it is the opposite in CMIP.**

Author: The ECS and TCR estimates using the SSP scenarios with the FEBE are lower than those using RCP due to the overly strong aerosols over the historical period in the SSPs which require a lower aerosol linear factor along with lower ECS to best match the historical temperature record. The difference between the shape of the RCP and SSP aerosol forcing can also account for this. We will expand upon this in the revision.

**Fig. 8: The figure would be easier to read if the single plots were aligned with each other according to their axes and were shown with the same font size.**

Author: Ok.

**L417 The CMIP5 MME is mostly warmer for the given period, but not always (compare i.e. the early 1940s where observed temperatures are higher than the mean). Why the upper limit of 1960 was chosen is unclear, since the CMIP simulations remain warmer most of the time afterwards (and in particular, after 2000).**

Author: This statement will be changed to entail the whole historic temperature sets in the revision.

**L419-422 It is not discussed whether FEBE tracking the hiatus is a sign of FEBE's skill or a result of the fact that the parameters were chosen based on the same observations that include this hiatus. Since EBMs usually have trouble with this hiatus, it would be interesting to**

**discuss whether there exist other conceptual models that reproduce the hiatus or whether this is a standout feature of FEBE.**

Author: One of the reasons why the FEBE tracks the hiatus well is due to the fact that the parameters are calibrated over the historical temperature observations. We show that the FEBE tracks the entire period since 1880 well, not only the hiatus. We calrify this in the revision.

**L421 There is no lower plot in fig. 9.**

Author: The main plot (rather than the inset) was meant, this will be corrected.

**Fig. 9: The observations show strong interannual variability, a discussion on whether a temporally higher resolved FEBE could capture such short-term variability (or even whether a higher resolution in time is feasible with this approach). Equally interesting would be to discuss whether other response functions could be designed to capture variability on arbitrary timescales. A bit unclear whether FEBE is useful for discussions of variability at all or should be applied only to discussions of the mean.**

Author: In Lovejoy et. al. (2021) and Lovejoy (2021), the FEBE is shown to plausibly reproduce the annual cycle at monthly resolution, in particular to explain the lag between the temperature maximum and the maximum in the radiative forcing. The small scale FEBE limit is not known, although it is likely to be 1 month. Justification comes from the success of the high frequency FEBE limit that successfully forecasts monthly and seasonal temperatures (see Del Rio Amador et al. (2019)). The variance of $\gamma(t)$ is the amplitude of the internal forcing assumed to be a Gaussian white noise is estimated in the response to referee 1 using Lovejoy et al. (2021). The following plot (fig. 1) shows the mean observational temperatures series along with one realization of the FEBE (with optimal parameters based on $F_{Aer_{RCP}}$) using both the internal and external forced variability. This work needs more work to be fully convincing, but it is promising at the moment.

[Figure]

Fig. 1: The mean observational temperature series (red) shown alongside a single realization of the FEBE with internal and externally forced variability (blue, shifted up 1 for clarity.)

Del Rio Amador, L., and Lovejoy, S., Predicting the global temperature with the Stochastic Seasonal to Interannual Prediction System (StocSIPS) Clim. Dyn. doi: org/10.1007/s00382-019-04791-4., 2019.

Lovejoy, S., Procyk, R., Hébert, R. and del Rio Amador, L. (2021), The Fractional Energy Balance Equation. Q J R Meteorol Soc. https://doi.org/10.1002/qj.4005

Lovejoy, S.: The Half-order Energy Balance Equation, Part 1: The homogeneous HEBE and long memories, Earth Syst. Dynam. Discuss., https://doi.org/10.5194/esd-2020-12, 2021.

**L461-462 According to the section on aerosol forcings (fig. 1), there is a future aerosol forcing in the RCP and SSP scenarios. It is unclear whether here the removal of the volcanic forcing is meant. Since the forcings in FEBE and SSP should be consistent (with the exception to the dampening parameters for the aerosols mentioned in the first part of the sentence) uncler why this would produce the temperature difference.**

Author: Thank you for pointing out how this line is unclear - the higher ECS found in CMIP6 models is related to compensating for the too strong aerosols in the historical part, thus when the concentration of aerosols begins to drop to zero (as in the SSP scenarios) the CMIP6 models heat much quicker than the FEBE. This will be better clarified in the revision.

**The effect of the lack of volcanic forcing in the future projections is not discussed, nor whether (and if so, how) the authors suggest its inclusion in future studies.**

Author: Future volcanism was not prescribed by the CMIP experiments so it was set to null in the future. Since we may be confident that volcanoes will continue to erupt, a mean rate of volcanic cooling could be prescribed set to the time average of past volcanism assuming that future volcanism is on average the same as the past. It could be interesting to include stochastic volcanic forcing in the future for ensemble realizations of projections, in a future paper.

**Fig. 10: The sine-like oscillations in the future projections by FEBE (esp. RCP2.6 and RCP4.5) should be mentioned in the text. I assume they reflect the solar forcing cycle, but remain uncertain whether the different magnitude of this cycle in the scenarios is due to a difference in overall magnitudes of forcings between scenarios.**

Author: The sine-like oscillations are from the solar forcing cycle which prescribed the reproduction of the last 11-year cycle for the future - their magnitude is the same for each scenario.

**Fig. 11: It is not discussed why these FEBE projections are notably smoother than the ones in fig. 9 and whether this is only a reflection of the difference in forcing.**

Author: These projections are smoother due to the scale shown in the plots - this will be addressed in the revision.

**A discussion about what could lead FEBE to overestimate future warming would be interesting and in particular whether it can be argued based on the projections in Fig. 10  11 that FEBE provides a baseline of minimal warming.**

Author: The FEBE constrains the strength of historic aerosol cooling to best match the observed temperature record, this results in less aerosol cooling than that modelled in the CMIP experiments. So if a large future aerosol forcing is introduced the resulting warming may be underestimated by FEBE due to the reduced strength of the aerosol forcing in the model compared to the CMIP models.

In general if the FEBE underestimates the memory of the system to radiative forcing, then it may underestimate the warming, but would at least be a robust lower bound. The converse could be said about overestimating the memory, and thus overestimating the warming as the forcing decreases, but this looks less likely.

**L478-479 It seems to me that this phrasing is slightly misleading. It reads as if extremely likely refers to the 15 years. However, it seems like 15 years later is just the point when it becomes extremely likely that the 1.5○ threshold is exceeded.**

Author: Thank you for noticing this - it is indeed poor phrasing, it will be fixed in the revision.

**Fig.12 bottom: It is not mentioned that the solid RCP scenario is at 0. Would be good to change the plot so that this becomes visible. Would also help to mark the "extremely likely" threshold at 0.95 discussed in the text.**

Author: It will be mentioned that the solid RCP scenario is at 0 and reflected in an improved plot.

**Fig.13 top: It is not discussed why the probability for the SSP126 scenario decreases so strongly after 2070 in the FEBE ensemble.**

Author: The probability for SSP126 scenario (blue, dotted) decreasing so strongly after 2070 was partially a visual artefact, this has been corrected in the updated fig.13. The probability still decreases strongly, but with a probability higher than before, this decrease is due to the combination of a lower FEBE ECS with regards to the CMIP6 MME and the strong decrease of $CO_2$ emissions in the scenario. We have also included the CMIP6 MME for scenario SSP245 in the revision (dashed lines).

**L482-487: This seems more like a motivation suitable for the introduction than a conclusion.**

Author: These lines are to reiterate why complementary models should be developed, and to explain the motivation for the FEBE.

**It is not discussed for what kind of studies the authors suggest the usage of FEBE, nor what the gaps are in studies with more complex models that FEBE might fill.**

Author: EBMs are useful as they are able to produce individual climate projections with reduced computational resources. The FEBE is based on the real climate not model climates, it is therefore complementary to the GCMs. An extended discussion on the benefits and uses of this will be included in the revision.

**L568 The authors say that their goal is to improve future projections, but do not expand on how they think that will be possible with FEBE.**

[Figure]

Fig. 13: The probability for the global mean surface temperature of exceeding a 1.5K threshold (top), and a 2K (bottom) are given as a function of years for the FEBE, using $F_{Aer_{RCP}}$ (blue) and for the CMIP6 MME (black). The three SSP scenarios are considered for each case: SSP 126 (solid), SSP 245 (dashed), and SSP 585 (circles).

Author: With tighter constraints on ECS and TCR from the FEBE we can better estimate future warming when bringing together multiple lines of evidence such as that done is Sherwood et al. (2020). The FEBE once expanded spatially provides a flexible framework which can be calibrated directly on observations, providing a direct representation of forcing to response relationships. We can also calibrate the FEBE on the historical runs of the CMIP models, updating our parameter estimates, allowing for GCM-FEBE hybrid projections.

Sherwood, S. C., Webb, M. J., Annan, J. D., Armour, K. C., Forster, P. M., Hargreaves, J. C., et al. (2020). An assessment of Earth's climate sensitivity using multiple lines of evidence. Reviews of Geophysics, 58, e2019RG000678. https://doi.org/10.1029/2019RG000678

**L569 It is unclear how FEBE can be used to understand the generational differences between CMIP models better.**

Author: We plan on calibrating the FEBE on the historical runs of the CMIP models in order to perform a feedback analysis to investigate the differences between how models treat their volcanic and aerosol forcings through the parameters $\nu$ and $\alpha$ in a future work.

**Neither the model code nor the simulations with the FEBE are**

**included in a code/data availability section and made available to the reader.**

Author: The forcing, and temperature data along with realizations of the FEBE will be included in an open source format, and the code of the model will be made available upon publication, but can be made directly available to the reviewer.

**citation style hinders readability (brackets even when the citation is part of the sentence and double brackets when listing several references)**

Author: Ok, this will be cleaned up.

**The whole text needs to be checked for correct comma placement, especially with respect to introductory phrases and interrupters. The punctuation could also be improved with respect to colon and semicolon usage.**

Author: A thorough proof reading will be done to improve the grammar and readability.

**in this state of the manuscript it did not seem to make sense to give detailed feedback on specific grammatical errors, so these are not included in this review**

Author: The grammar will be improved.

**the way "FEBE" is used in the text is inconsistent, sometimes it is "the FEBE", at other times just "FEBE" without an article**

Author: "The FEBE" will be made consistent throughout the text.

**inconsistent usage of "figure X" or "fig. X" in the text**

Author: We will revise using ESD's publishing style:

- The abbreviation "Fig." should be used when it appears in running text and should be followed by a number unless it comes at the beginning of a sentence, e.g.: "The results are depicted in Fig. 5. Figure 9 reveals that...".

**There are quite a lot of unnecessarily run longing sentences that hamper readability.**

Author: These will be shortened or split into two to improve readability.

In general, the plots could be cleaner, i.e. for multi-panel plots the plot sizes as well as font sizes differ.

Author: Thank you for pointing this out, they will be made cleaner.

The linkage between sentences, paragraphs and sections could be improved to make the reader's life easier. Also the internal structure of sections, paragraphs and even sentences could be more coherent. For example, it's not always easy to identify the topic and stress in a sentence. This could be facilitated by an improved structure of sentences. Sometimes, the connection between subject and verb gets lost due to a sentence structures that is too complicated. In general, reducing long sentences and technical language would improve the reader's understanding.

Author: Ok.

**L20 MME used without introducing the abbreviation**

Author: Multi-Model Ensemble will be used first before introducing the abbreviation.

**L21 should be "the FEBE projections were...", also comma missing after the introductory phrase**

Author: Ok, corrected.

**L40 different dashes are used at the beginning and end of the insertion, "that" after the insertion should be discarded (whole sentence could benefit from being re- formulated for better readability, though)**

Author: Ok.

**L63 MME as an abbreviation is only introduced in L94**

Author: Ok.

**L83 should be Hebert et al.'s truncated power law or at least it has not been introduced so far as having been developed by only one person**

Author: Ok.

**L89 article missing in "In the methods and materials section"**

Author: This will be corrected, thanks.

**L138 should have a reference to where that discussion happens**

Author: Ok.

**L165 "While RCP8.5 and SSP585..." subject of the sentence is plural, rest of sentence is singular**

Author: Ok.

**L171 "and" between "Kyoto protocol" and "ozone depleting substances" would make this sentence easier to read**

Author: Ok.

**L197 should be "two 11-year solar cycles"**

Author: Ok.

**L327 the last part of the sentence lacks a verb**

Author: Ok.

**L229 should be "find in/for both cases" or similar**

Author: Ok.

**L348 no apostrophe for " its' "**

Author: Ok.

**L349 "where" or similar instead of "whereas"**

Author: Ok.

**L362 "CO2 levels were" or "CO2 was increased"**

Author: Ok.

**L415 rogue bracket at the end of the sentence**

Author: Thank you for noticing this.

**L416 either "Between 1915-1960" or "In the 1915-1960 period" or similar**

Author: Ok.

**Fig.9 There are white artefacts in the grid lines, box borders, and axes of the plot as well as in the upper limit of the CMIP5 MME. It is unclear why the x-axis of the inset unneccessarily includes negative values. The plot would appear cleaner if the borders of the inset were aligned with the axes. CMIP5 MME is in gray not black (same in later plots). Referring to the inset as the top is confusing. The left border of the inset in the whole plot is placed too late.**

Author: Ok.

**L431 scratch "In comparison" at the beginning of the sentence (at this point it is unclear with respect to what the comparison is supposed to be and it does not match the remainder of the sentence)**

Author: Ok.

**L448 forcings should be plural in "the other forcing are practically"**

Author: Ok.

**L454 doubled "the"**

Author: Ok.

**L461 drop "by" in "are by nearly 65% warmer"**

Author: Ok.

**L547 "to" missing in "purely due differences"**

Author: Ok.

**L556 I would suggest replacing "be passed" with "happen" in the context of this sentence**

Author: Ok.

**Held et al. citation not in correct place in alphabetical order**

Author: This will be corrected, thanks.

---

## Author Comment (AC3) · 18 Mar 2021

Roman Procyk et. al.

**The manuscript provides, at times, incomplete details on how the results were obtained and how the approach was designed, making reproducibility of the results difficult. I would suggest sufficient details be added. This would also help make potential errors in the approach more detectable.**

Author: A revised discussion will be given to how the results were obtained in the revision. The forcing, and temperature data will be included in an open source format, and the code of the model will be made available in the future. Extra information about what is to be included was given in the response to reviewer 1 and 2.

**I would also like to see some discussion on approximating the joint prior probability density functions with a multivariate Gaussian process, including what motivated this decision, how accurate this approximation is and if this could have any repercussions for the final results. How the covariance between the parameters used in the joint prior distribution is determined should also be included.**

Author: The results shown in Figures 3,4 and 7 are the marginal distributions for each of our five parameters - their Gaussianity was checked and thus we were motivated to approximate the joint posterior probability density functions with a multivariate Gaussian for our future projections otherwise this process can be computationally expensive (prior experience of a coauthors). The way it is written in the paper, we understand the confusion - the prior probability functions are not approximated by a multivariate Gaussian but the posterior probability was with the correlations between all parameters taken into account, this will be clarified further in the revision.

**Physical units and chemical formulas should not be typeset in italics.**

Author: Ok, this will be corrected.

**Ranges should be denoted using en dashes (–), instead of hyphens (-), e.g. 1998–2015.**

Author: Ok, this will be corrected.

**Top panel in figure 6: The horizontal axis of the smaller window ranges from 0 to 25 years, but the dotted lines suggest it is taken from the range of 0 to 50 years from the larger figure. Please correct/clarify.**

Author: The inset is for 0 to 25 years, this will be cleaned up in a revised figure 6 so that the dotted lines reflect this.

**Line 274: The likelihood function is said to be a posterior probability. This is incorrect, as the likelihood is a distinct probability distribution. Also in line 274: the terms "posterior" and "a priori" seem to have been switched.**

Author: This is correct, a change will be made in the revision.

**Being in the Bayesian modeling framework, where parameters are treated as stochastic variables instead of fixed and true values, I think it would be more appropriate to use the term "credible intervals" instead of "confidence intervals". The former refers to an interval within which an unobserved (stochastic) parameter value falls with a particular probability, whilst the latter is an interval that we are, to a certain degree, confident include the true (deterministic) parameter value.**

Author: Thank you for pointing out this useful distinction, it will be changed in the revision.

---

## Author Response (AR2)

**ESD-2020-48 Minor Revision**

**Reviewer 1:**

- This paper addresses the same problems as Hebert et al. (2021), but uses the FEBE model instead of a truncated power law. I would like to see some more comments on the strengths and weaknesses between these two models, and some discussion on what this paper adds to Hébert et al. (2021) to justify this work as a standalone paper.

As mentioned in the paper, Hébert et al. (2021) tamed the divergences by cutting off the power law CRFs at small scales. The caveat was that the CRF model truncation was somewhat ad hoc, and therefore only useful at decadal or longer scales, while the FEBE and its Green's function covers all ranges of scales. While the low frequency Green's function can be very close to Hébert et al. (2021)'s truncated power law CRF, the high frequency regime is able to produce internal variability coherent with the observed scaling and fractional Gaussian noise used for skillfully forecasting the stochastic (internal) variability at monthly, seasonal, interannual (macroweather) scales (Lovejoy et al. 2015; Del Rio Amador and Lovejoy 2019, 2020). In addition, it is much more sensitive to the volcanic forcing and the parameters are more strongly constrained. A significant consequence and improvement over Hébert et al. (2021) is the error model was not ad hoc, rather predicted by the model itself: the internal variability response to white noise internal forcing. The differences in the two models are more thoroughly covered in line numbers 65-70, 87-92, 365-375, 691-700 of the first revision.

Lovejoy, S., Del Rio Amador, L., and Hebert, R.: The ScaLIng Macroweather Model (SLIMM): using scaling to forecast global-scale macroweather from months to decades, Earth System Dynamics, 6, 637, 2015.

Del Rio Amador, L. and Lovejoy, S.: Predicting the global temperature with the Stochastic Seasonal to Interannual Prediction System (Stoc-SIPS), Climate Dynamics, 53, 4373–4411, https://doi.org/10.1007/s00382-019-04791-4, https://doi.org/10.1007/s00382-019-04791-4, 2019.

Del Rio Amador, L. and Lovejoy, S.: Using scaling for seasonal global surface temperature forecasts: StocSIPS, Climate Dynamics, 2020.

- In line 298 the paper argues that the fractional Gaussian noise approximation of the residuals allows the model to take into account the strong power law correlations, but it's accuracy is weaker on low frequencies which only weakly influence the likelihood function. I would like some clarification on what motivates the FEBE (on the applications considered in this study) instead of simply using a power law.

Described in the comment above, the FEBE is a model that can be derived rather than being ad hoc as the case would be when using a pure power law. To make more realistic models, the key issue is energy storage. Storage is a consequence of imbalances in incoming short wave and

outgoing long wave radiation and it must be accounted for in applications of the energy balance principle (Trenberth et al., 2009). As pointed out in Lovejoy (2019, 2021a) and developed in Lovejoy et al. (2021) it is sufficient that the scaling principle not be applied to the Greens (Climate Response) Function, but rather to the storage term in the EBE. In lieu of the energy being stored by uniformly heating a box, energy is instead stored in a hierarchy of structures from small to large, each with time constants that are power laws of their sizes. This conceptual shift can be implemented simply by changing the integer order of the storage (derivative) term in the EBE to a fractional value: the Fractional Energy Balance Equation (FEBE). While Lovejoy et al. (2021) derived the FEBE in a phenomenological manner, Lovejoy (2021b, 2021c) showed how it could instead be derived from the continuum mechanics heat equation used in the Budyko-Sellers models. Indeed, by extending Budyko-Sellers models from 2D to 3D (i.e. to include the vertical) and imposing the (correct) conductive – radiative surface boundary conditions, one immediately obtains fractional order equations for the surface temperature. In other words, nonclassical fractional equations and long memories turn out to be necessary consequences of the standard Budyko-Sellers approach.

Trenberth, K. E., Fasullo, J. T., & Kiehl, J. (2009). Earth's global energy budget. Bulletin of the American Meteorological Society, 90 (3), 311–324. https://doi.org/10.1175/2008BAMS2634.1

Lovejoy, S. (2019). Weather, macroweather and climate: Our random yet predictable atmosphere. Oxford U. Press.

Lovejoy, S. (2021a). Fractional relaxation noises, motions and the fractional energy balance equation. Nonlinear Processes in Geophysics, 2021. https://doi.org/10.5194/npg-2019-39

Lovejoy, S., Procyk, R., Hébert, R., & Del Rio Amador, L. (2021). The fractional energy balance equation. Quarterly Journal of the Royal Meteorological Society, n/a(n/a). https://doi.org/https://doi.org/10.1002/qj.4005

Lovejoy, S. (2021b). The half-order energy balance equation, part 1: The homogeneous hebe and long memories. Earth System Dynamics Discussions, 1–36. https://doi.org/10.5194/esd-2020-12

Lovejoy, S. (2021c). The half-order energy balance equation, part 2:the inhomogeneous hebe and 2d energy balance models. Earth System Dynamics Discussions, 1–44. https://doi.org/10.5194/esd-2020-13

- There is currently very little description on the limitations of the FEBE. The manuscript would benefit from further discussion on where the model is suitable and where it is not.

The FEBE is in fact a regional (horizontal space) – time model that here is integrated over space to yield a "zero-dimensional" model similar to the standard "Box model" except for a different order of differentiation.  At the moment it is linear, but nonlinearities such as temperature-albedo feedbacks are easy to introduce.  Other temperature- forcing feedbacks such as those responsible for tipping point phenomena could also be easily incorporated.  Since the FEBE can be generalized in many ways, its limitations are not in fact clear.  A practical difficulty (but not a limitation) is that some of the parameters- especially in the regional FEBE are difficult to empirically estimate (especially the regional relaxation times).  These issues will be explored in further publications.

 - In the estimation of the model parameters the forced response is subtracted from the data before the residuals are fitted using Mathematica. In my understanding this is done by first sampling parameters from the prior distribution, then by removing the corresponding forced response (based on the simulated parameters) before fitting a fractional Gaussian noise process to the data. However, since the removed forced response also depends on the same parameters which are to be estimated, this component is not fitted to the data before it is removed and hence its shape is determined entirely by the priors. This could make the model more sensitive to the choice of priors.

The parameters are not initially sampled from the prior distributions, but rather sampled from broad uninformative uniform distributions to generate a wide array of possible forced responses. Then we obtain many residual temperature series by removing the generated forced responses from the global temperature series which is a fraction Gaussian noise process as predicted by the FEBE model itself. From these residual series, we calculate the likelihood of being a fractional Gaussian noise with parameter h, giving us our multidimensional likelihood function wholly independent to the prior distributions which are applied later using Bayes.

In my opinion, it would be better if both the forced response and the residuals were to be fitted simultaneously. This can be achieved by e.g. a hierarchical Bayesian modeling approach. This framework is also able to incorporate non-Gaussian priors, which could possibly remove the need for approximating the joint posterior.

We understand what the referee is suggesting, but this is an unnecessary complication as the error model for each possible forced response is given theoretically by the FEBE itself (when the FEBE is forced by a white noise, as is the case in Eq. 1, the stochastic portion is a fractional Gaussian noise – the residual series). The joint posterior is only approximated by a Gaussian for computationally efficiency when generating many realizations of projections.

- I would like to see a comment added to the text which ensures that the Gaussian approximation of the joint posterior (Eq. (21)) is indeed accurate.

The Gaussian approximation was only used for projections, rather than creating a net over the large five-dimensional parameter space to draw parameters (as was done in Hebert et al. 2021 – a computationally expensive process). The parameter distributions for all five parameters shown in the results (Section 3) are all from the marginal probabilities of the joint posterior – their Gaussian appearance and computationally considerations are why a Gaussian approximation was chosen.

- Gaussian priors imply a non-zero probability of negative values, which could cause e.g. scaling parameters to be negative. I would like a comment that addresses if/how the authors have constrained the model parameters.

For the model parameters, h and $\tau$, we restrict them to being greater than zero – in the case of the scaling exponent this is justified as the theory of fractional Relaxation noises (fRn, the generalization of fGn) is only for h>0, and a negative relaxation time has no physical meaning (breaks causality).

- In line 467 the authors state that they have performed 500 Monte Carlo simulations of the projections. Has it been verified that the accuracy is sufficient? A comment clarifying this would be welcome. Furthermore, would it be computationally feasible to increase this number, if needed?

The 500 Monte Carlo simulations chosen is already far more than needed. Included is a table showing the mean and standard deviation of the RCP parameters depending on the amount of simulations drawn from the Monte Carlo simulation of the approximated posterior distribution – it is clear that at already 100 simulations we can reproduce the values presented in Table 1 (shown in the first row of this table) in the paper.

| Parameters | h | $\tau$ | $\alpha$ | $\nu$ | s |
|---|---|---|---|---|---|
| MCMC Simulations | 0.38 (0.05) | 4.7 (2.3) | 0.60 (0.40) | 0.28 (0.13) | 0.56 (0.11) |
| 10 | 0.41 (0.05) | 5.1 (2.0) | 0.44 (0.27) | 0.32 (0.10) | 0.48 (0.09) |
| 100 | 0.38 (0.04) | 4.5 (1.9) | 0.58 (0.36) | 0.31 (0.13) | 0.55 (0.13) |
| 250 | 0.38 (0.05) | 4.6 (2.0) | 0.62 (0.39) | 0.31 (0.12) | 0.56 (0.10) |

| | | | | | |
|---|---|---|---|---|---|
| 500 | 0.38 (0.05) | 4.6 (2.4) | 0.63 (0.41) | 0.29 (0.12) | 0.55 (0.11) |
| 1000 | 0.38 (0.04) | 4.6 (2.2) | 0.64 (0.42) | 0.28 (0.11) | 0.56 (0.10) |

Other than this I have some minor/technical comments and suggestions:

- In the Bayesian framework one should use "credible intervals" instead of "confidence intervals".

- Appropriate punctuation after equations:

(3), (4), (11), (20), (22)

- Figure 13: Caption states that CMIP5/6 MME is represented by black, but in the figure the color is gray.

- Line 693: "Latter" is used when the preceding sentence only has one object

- Line 712: "Projections through to 2100"

- Line 726: "The FEBE could be also" to "The FEBE could also be"

- Line 731: citation should be parenthetical

These will be corrected in the revision.

Reviewer 2:

1. I would suggest to take care of some typos, to close some parentheses, correct figure captions when describing lines or symbols, and to fix some issue as capital letters without any punctuation before. This especially occurs after equations.

Ok.

2. Line 238: "We consider the standard assumption about internal variability that it is forced by a Gaussian "delta correlated" white noise". Could the authors add a reference to this? Why not to use a red noise spectrum that is also generally related to the internal noise?

The internal variability produced by FEBE is forced by a white noise, but does not result in a white noise (the result is red). The internal variability generated is a fractional Relaxation noise, which is a type of red noise. This is the same concept as in Hasselman (1976) where a white noise was used to produce a red noise, but with an exponential function rather than the FEBE. The physical concept is the same: the white noise corresponds to the high-frequency atmospheric

forcing, and the response function (FEBE in our case, exponential in Hasselman 1976) corresponds mainly to the mixed-layer of the ocean which integrates those fast variations to produce a red noise spectrum.

Hasselmann, K., 1976: Stochastic climate models. Part I. Theory. *Tellus*, **28**, 473–485, https://doi.org/10.3402/tellusa.v28i6.11316.

3. Figure 10: I would like to suggest the authors to comment more on this interesting result. In particular:
what is the range of scales over which the scaling exponent is evaluated?
if, as expected, in the macroweather regime temperature fluctuations decrease, I was wondering why the authors do not use the full range of scales belonging to this regime?
If using the (full) accessible macroweather regime to compute h, how to explain the difference with observations? I would expect h~0 for observations.

We do not use the full range of scales to calculate the scaling exponent because the data is a superposition of natural variability and forced response , the latter is not scaling (and in fact, the former is not perfectly scaling either once we approach  the relaxation time (here ≈ 5 years). Both effects break the scaling as indicated. The straight line shown in Figure 10 is not a regression, but a reference line to show the theoretical high frequency result when forcing by a white noise internal forcing using the empirical h≈ 0.4 value; over the macroweather regime the observations are as the referee says h~0.

4. Figure 12: it seems to me that FEBE lowered projection uncertainties but also directly projections. How the authors explain this? Could be due to the parametric uncertainty that differs from structural one of MME?

The key reason for the lower temperature projections as compared to the CMIP MMEs, is the lower ECS of the FEBE which is a result of the long memory storage and inclusion of the aerosol scaling factor that accounts for the overly strong historical aerosol forcings. Due to these two main factors, the FEBE has a lower median (and more constrained) estimate of ECS in comparison to the CMIP MMEs, thus leading to lower projection uncertainties and lower projected temperatures.

5. Figure 13: how the authors explain the oscillations observed for the RCP 2.6/SSP 1-26 scenario for FEBE? It seems to me that they are 10-yr period, is there any relation with the transition for scaling laws?

The oscillations observed in the RCP 2.6/SSP 1-26 are caused by the 11-year solar cycle, which can be seen in the same scenario projections in Figure 12. These oscillations also occur in the projections using the higher emission scenarios, although due to the scale of temperature change due to anthropogenic warming they are not visible.

6. Figure 13: it seems to me that there is a good agreement with RCP 8.5/SSP 5-85 scenario for FEBE and MME (although shifted, why?), while a different slope of the probability is found (steeper for FEBE than MME). Could the authors argument on this? Is it related to the intrinsic parametric uncertainty?

In the case of RCP 8.5/SSP 5-85 (along with other scenarios), the FEBE probability of crossing thresholds 1.5K/2.0K (if they are to be crossed) later as compared to the CMIP MMEs. This can be understood from Figure 12, which shows that the warming projected by FEBE is less than the CMIP MMEs, and in the cases where emissions continue to rise (all but scenarios RCP 2.6/SSP 1-26) this will result in the crossing of said thresholds in the inevitable future. The slope may be slightly steeper in the FEBE probabilities as the referee notes but only very tangentially as the RCP8.5/SSP 1-26 probabilities (circles – also the probability curve furthest to the present in all cases) are nearly parallel. The marginal difference in the slopes of the probability curves may be that future aerosols never are reduced to zero (see Figure 1 bottom); so that when the FEBE reduces aerosol forcing by the aerosol linear scaling factor they have a much weaker (nearly negligible effect) in the far future in comparison to CMIP models which maintain a constant cooling forcing into the future.